# Learning Encoding-Decoding Direction Pairs to Unveil Concepts of Influence in Deep Vision Networks

**Alexandros Doumanoglou**                                          *aldoum@iti.gr*
*Department of Advanced Computing Sciences (DACS), University of Maastricht (UM)*
*Information Technologies Institute (ITI), Centre for Research and Technology Hellas (CERTH)*

**Kurt Driessens**                          *kurt.driessens@maastrichtuniversity.nl*
*Department of Advanced Computing Sciences (DACS), University of Maastricht (UM)*

**Dimitrios Zarpalas**                                          *zarpalas@iti.gr*
*Information Technologies Institute (ITI), Centre for Research and Technology Hellas (CERTH)*

**Reviewed on OpenReview:** *https://openreview.net/forum?id=lIeyZpPEJn*

## Abstract

Empirical evidence shows that deep vision networks often represent concepts as directions in latent space with concept information written along directional components in the vector representation of the input. However, the mechanism to encode (write) and decode (read) concept information to and from vector representations is not directly accessible as it constitutes a latent mechanism that naturally emerges from the training process of the network. Recovering this mechanism unlocks significant potential to open the black-box nature of deep networks, enabling understanding, debugging, and improving deep learning models.

In this work, we propose an unsupervised method to recover this mechanism. For each concept, we explain that under the hypothesis of linear concept representations, this mechanism can be implemented with the help of two directions: the first facilitating encoding of concept information and the second facilitating decoding. Compared to previous matrix decomposition, autoencoder, and dictionary learning approaches which rely on the reconstruction of feature activations, we propose a different perspective to learn these encoding-decoding direction pairs. We base identifying the decoding directions on directional clustering of feature activations and introduce signal vectors to estimate encoding directions under a probabilistic perspective. Unlike most other works, we also take advantage of the network's instructions encoded in its weights to guide our direction search. For this, we illustrate that a novel technique called *Uncertainty Region Alignment* can exploit these instructions to reveal the encoding-decoding mechanism of interpretable concepts that influence the network's predictions.

Our thorough and multifaceted analysis shows that, in controlled, toy settings with synthetic data, our approach can recover the ground-truth encoding-decoding direction pairs. In real-world settings, our method effectively reveals the encoding-decoding mechanism of interpretable concepts, often scoring substantially better in interpretability metrics than other unsupervised baselines, such as PCA and NMF. Finally, we provide concrete applications of how the learned directions can help open the black box and understand global model behavior, explain individual sample predictions in terms of local, spatially-aware, concept contributions and intervene on the network's prediction strategy to provide either counterfactual explanations or correct erroneous model behavior.

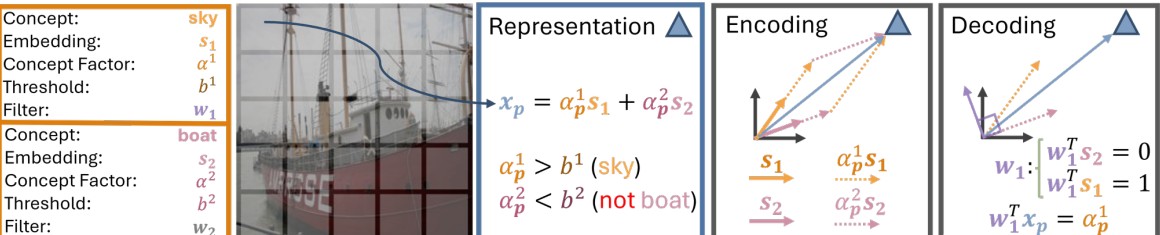

Figure 1: Concept Encoding - Decoding under the Linear Representation Hypothesis: Deep networks encode high-level concepts, such as *sky* or *boat*, in distinct directions of their latent space, respectively $s_1$ and $s_2$. The illustration shows the encoding of two concept latent factors (i.e., the degree of concept presence) $\alpha^1, \alpha^2$ within a patch's representation $x_p$ by utilizing the concept *embedding* directions $s$. Additionally, it demonstrates how a *filter* $w_1$ can be employed to extract one of these latent factors from the representation. The illustration omits depicting the latent space bias for brevity. We use the terms {*encoding* direction, concept *embedding*, *signal* direction} and the terms {*filter*, *decoding* direction} interchangeably throughout this article.

## 1 Introduction

The linear representation hypothesis suggests that deep neural networks encode high-level concepts in directions of their latent space (Bereska & Gavves, 2024). This hypothesis is sufficiently supported by empirical evidence, mostly by the effectiveness of linear probing. In the latter, a single linear layer can be trained on top of feature representations originating from the upper part of deep neural networks to solve semantic tasks with great success (Szegedy et al., 2014; Alain & Bengio, 2018; Zhou et al., 2018; Kim et al., 2018; Elhage et al., 2022; Nanda et al., 2023). In the concept encoding mechanism suggested by the linear representation hypothesis, the concept content of an input patch is written to its embedding as a linear combination of *concept embeddings*: directions in the latent space represented as vectors, each one associated with a concept. The corresponding scalar coefficient in the linear combination constitutes the concept's latent factor that encodes the degree of concept presence in the input. Due to its utility in encoding the latent factor to a vector representation, we also refer to the concept embedding as the concept's *encoding* or *signal* direction. Retrieving the latent factor back can be accomplished by taking the inner product between the patch embedding and a *filter*, another vector which we term a *decoding* direction. (Fig. 1) An encoding-decoding direction pair is *interpretable* whenever it is related to a concept that is aligned with human intuition; and is *influential* whenever it is related to a concept that the network consistently uses to make predictions (Kim et al., 2018; Pfau et al., 2020).

The encoding - decoding mechanism of concepts for a pre-trained network is not directly accessible. Instead, it is a latent mechanism emerged from network training and needs to be inferred by any means of reverse engineering. Recovering this mechanism is of significant importance, as accessing it enables understanding representations (Zhou et al., 2018; Kim et al., 2018), assessing influence of concepts on the network's predictions (Kim et al., 2018; Pfau et al., 2020; Fel et al., 2023b; Pahde et al., 2024), debugging neural networks (Anders et al., 2020), and correcting erroneous model behavior (Anders et al., 2022; Pahde et al., 2023; Neuhaus et al., 2023; Dreyer et al., 2024a).

Previous works that identify concept directions rely either on concept annotations (Zhou et al., 2018; Kim et al., 2018; Pahde et al., 2024) or on a decompose-then-reconstruct regime of feature activations (Zhang et al., 2021; Graziani et al., 2023b; Bricken et al., 2023; Lim et al., 2024; Cunningham et al., 2024; Bussmann et al., 2025). The former suffer from the need for concept speculation and the access to annotations while the latter, face the difficulty of overcoming unstructured noise that may be present in feature activations in the form of distractors (Haufe et al., 2014; Kindermans et al., 2017) or dark matter (Engels et al., 2025).

Unlike previous approaches, our method relies on neither annotations nor the reconstruction of activations, thereby sidestepping these earlier limitations. Instead, we identify decoding directions via **directional clustering** of feature activations and estimate encoding directions from a probabilistic perspective. Moreover, in contrast to most prior work, we guide our search for directions by **exploiting the network's instructions, i.e. the prediction strategy baked into its weights** through a process we call *Uncertainty*

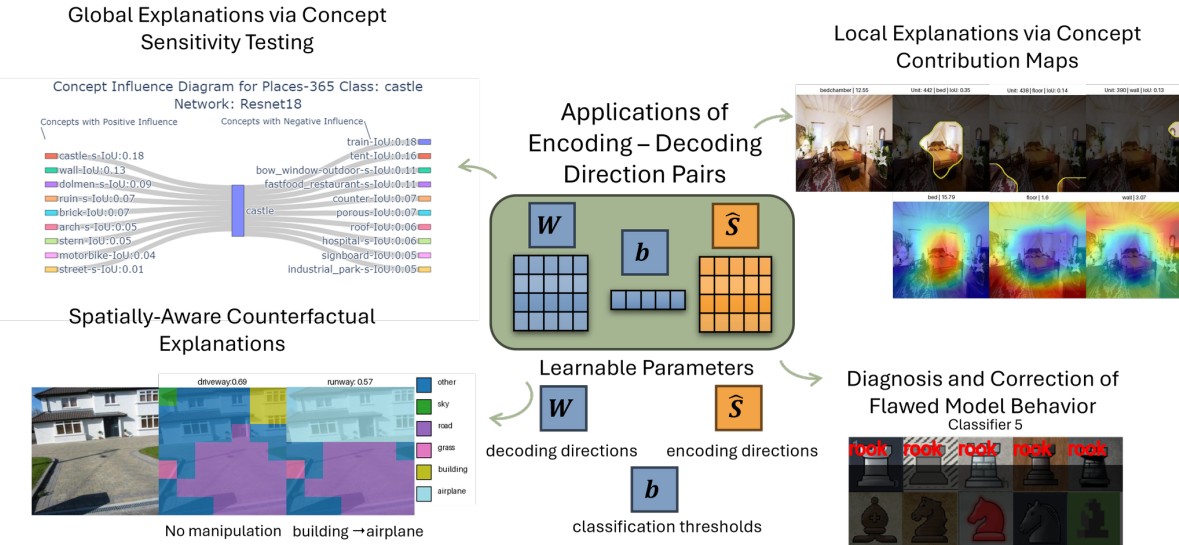

Figure 2: Our Encoding-Decoding Direction Pairs (EDDP) powers a range of applications, highlighting both the generality and the precision of our approach. The figure summarizes applications that we selected to discuss in this work.

*Region Alignment.* We conduct a thorough and multifaceted analysis that highlights the effectiveness of the approach in revealing the encoding-decoding mechanism of (a) the ground-truth concepts in controlled and synthetic toy settings, and (b) interpretable monosemantic concepts in real-world studies of state-of-the-art deep vision architectures. We also demonstrate the utility of our learned directions in obtaining global, local, and counter-factual explanations, as well as identifying and correcting flawed model behavior in a toy setting (Fig. 2).

Our comparative analysis primarily considers established matrix decomposition baselines, specifically Principal Component Analysis (PCA) and Non-Negative Matrix Factorization (NMF). We further extend the comparisons in Appendix Section A.16, where we evaluate our approach against the methods on which we build, namely Unsupervised Interpretable Basis Extraction (UIBE) (Doumanoglou et al., 2023) and Concept Basis Extraction (CBE) (Doumanoglou et al., 2024). Throughout, we focus on settings in which the concept space does not exceed the dimensionality of the embedding space, and we leave a systematic comparison with Sparse AutoEncoders (SAEs) (Sharkey et al., 2022) to future work.

# 2 Related Work

## 2.1 Direction Learning

We categorize related work into supervised and unsupervised direction learning approaches. In each category, we go through the previous methods, describing the limitations, differences, and similarities with our approach.

### Supervised Concept Direction Learning

Concept Activation Vectors (CAVs) (Kim et al., 2018; Zhou et al., 2018) use supervised linear classifiers trained on annotated concept datasets to separate representations of samples with a target concept from those without. To solve the task, the resulting classifier weights (filters) must extract (decode) the concept factors from these representations; thus, they approximate concept decoding directions, although they can be inexact due to distractor-noise in the features (Haufe et al., 2014; Kindermans et al., 2017; Pahde et al., 2024). To estimate concept encoding directions, PCAVs (Pahde et al., 2024) take the difference of means between representations of samples with and without the concept. These methods require both labeled data and prior specification of the concept name. In contrast to these probing approaches, the proposed method

is unsupervised, revealing concepts by analyzing the structure of the latent space and thereby avoiding both annotation costs and upfront concept specification.

**Unsupervised Concept Direction Learning** Existing unsupervised direction-learning methods that rely on matrix decomposition (PCA (Graziani et al., 2023a), SVD (Graziani et al., 2023b), NMF (Zhang et al., 2021; Fel et al., 2023b)) can discover concept directions without annotations, but they have key limitations. PCA and SVD are constrained by orthogonality and miss concepts that do not influence variance (Fel et al., 2023a). NMF, while non-negative, lacks latent space bias and, thus, expressivity, and it has no straightforward decoding directions. To alleviate these shortcomings, at least to some extent, Graziani et al. (2023a;b) combined direction learning with gradient re-ranking, and both Zhang et al. (2021) and Graziani et al. (2023a;b) worked on subgroups of features belonging to the same class category, establishing a class-conditioned protocol for direction learning. In contrast, the proposed method uses a class-agnostic protocol without the need for sample class labels. Additionally, our approach considers non-orthogonal concept directions and accounts for feature space bias, overcoming the limitations of previous methods.

Dictionary Learning (Bricken et al., 2023; Yun et al., 2023) and Sparse Autoencoders (SAEs) (Sharkey et al., 2022; Cunningham et al., 2024; Bussmann et al., 2025) identify decoding-encoding directions driven by the reconstruction of representations after decomposing them into sparse latent factors. They typically consider a latent space larger than the embedding space and may suffer from the difficulty of overcoming noise in the representations in the form of dark matter (Engels et al., 2025). In contrast, our method is independent of feature reconstruction, unrestricted by the relative sizes between the concept and the embedding space, and unlike these prior works, it implicitly links the learned directions to how the model actually utilizes them. However, in this work, we restrict our experiments to cases where the concept space is no larger than the embedding space and leave experimental comparisons to SAEs with the opposite assumption for future work. Further discussion regarding SAEs is provided in Section A.23.

Moving away from reconstructing representations, the methods of Doumanoglou et al. (2023; 2024) perform directional clustering of feature activations, a process that often unveils the decoding mechanism of interpretable, monosemantic concepts. We build on these previous approaches to estimate decoding directions, removing orthogonality and feature-standardization constraints and introducing new loss terms that preserve or improve concept interpretability while reducing the impact of distractor noise in the filter weights. Compared to the previous attempt (Doumanoglou et al., 2024) of linking directions to their utilization by the network, our *Uncertainty Region Alignment* approach achieves up to 22.56% relative improvement in interpretability metrics in 3 out of 4 cases. We also address the estimation of concept encoding directions, which these prior works did not address. Further details on this comparison are provided in Section A.3.

## 2.2 Applications of Directions

Our learned direction pairs enable a series of applications, from global model understanding and detailed spatially-aware local explanations down to model intervention, which allows for counterfactual explanations and model correction. Typical previous enablers of such applications are the Concept Bottleneck Models (CBMs) (Koh et al., 2020). Classic CBMs consist of a backbone feature extractor, a concept bottleneck predicting concept content from features, and a linear head mapping concepts to classes. What makes this architecture interpretable is the fact that the concept vector, i.e, the output of the concept bottleneck, is trained to have each dimension aligned with an interpretable concept. Recent post-hoc CBM variants (Yuksekgonul et al., 2023; Oikarinen et al., 2023) retrofit this structure onto existing black-box models by freezing the original backbone, discarding its head, inserting a concept bottleneck projection into concept space, and learning a task-specific sparse linear head over concepts. To learn concepts without annotations, these works exploit CLIP Radford et al. (2021), and further advances Benou & Raviv (2025) use visual prompting (Shtedritski et al., 2023) to make the bottleneck spatially aware, allowing localization of concepts at the patch level rather than only at the image level. Most recently (Rao et al., 2024), SAEs (Cunningham et al., 2024) have been used to build CBMs directly for CLIP.

To access the advantages of CBMs, these previous approaches require training the last linear head using the model's training data and, sometimes, additional concept-annotated datasets or external models like CLIP. In contrast, the proposed work is non-intrusive: it doesn't need to train new model components and does

not require concept labels. It shows that one can capture many benefits of concept bottleneck models by identifying directions corresponding to concepts the model already encodes and then using these directions in applications of mechanistic interpretability (Saphra & Wiegreffe, 2024; Kästner & Crook, 2024; Bereska & Gavves, 2024).

## 3 Background

In our formulation, which follows the linear representation hypothesis (Elhage et al., 2022; Nanda et al., 2023), the latent factor of a concept is a scalar denoting the concept's presence in the input, and is embedded in the latent space via multiplication with the concept's **encoding** direction, also called the concept's **embedding** or **signal** direction (Kindermans et al., 2017). For this reason, we also refer to this latent factor as the **signal value**. Feature activations are considered as linear combinations of signals and noisy directional components called **distractors**. In the proposed approach, a **filter**, through the inner product with an input representation, can extract the signal's value and for this reason we term it a **decoding** direction. Below we provide a more formal explanation of these terms and provide details essential to understand our contributions.

### 3.1 Preliminaries

Let $\boldsymbol{X} \in \mathbb{R}^{H \times W \times D}$ denote the feature representation of an image in an intermediate layer of a deep neural network with spatial dimensions $H, W \in \mathbb{N}^+$ and latent space dimensionality $D \in \mathbb{N}^+$. Let also $\boldsymbol{x_p} \in \mathbb{R}^D$ denote a **pixel** element of this representation at the spatial location $\boldsymbol{p} = (w, h)$, $w \in \{0, 1, ..., W-1\}, h \in \{0, 1, ..., H-1\}$. Since $\boldsymbol{x_p}$ is related to a patch within the input image of the network, we also refer to it as **patch embedding**.

### 3.2 Signals, Distractors, Filters, Concept-Detectors

In encoding a single concept $i$, Kindermans et al. (2017); Pahde et al. (2024) proposed a model for the data generation process of feature representations: $\boldsymbol{x_p} = \alpha_{\boldsymbol{p}} \boldsymbol{s}_i + \beta_{\boldsymbol{p}} \boldsymbol{d}$, $\boldsymbol{s}_i, \boldsymbol{d} \in \mathbb{R}^D, \alpha_{\boldsymbol{p}}, \beta_{\boldsymbol{p}} \in \mathbb{R}$. Here, $\boldsymbol{s}_i$ is the **signal** direction that carries the information of whether $\boldsymbol{x_p}$ is part of concept $i$. The concept information lies within the **signal value** $\alpha_{\boldsymbol{p}}$. Larger $\alpha_{\boldsymbol{p}}$ suggests greater confidence that $\boldsymbol{x_p}$ belongs to concept $i$. $\boldsymbol{d}$ is the **distractor** direction, modeling noise, or information not related to the concept. $\beta_{\boldsymbol{p}}$ follows a random distribution, typically the gaussian or uniform distribution, and is **independent** of whether $\boldsymbol{x_p}$ belongs to concept $i$. According to Kindermans et al. (2017), the value of the signal $\alpha_{\boldsymbol{p}}$ can be extracted using a **filter** $\boldsymbol{w}_i$ and the inner product: $z_{\boldsymbol{p},i} = \boldsymbol{w}_i^T \boldsymbol{x_p} = \alpha_{\boldsymbol{p}} \boldsymbol{w}_i^T \boldsymbol{s}_i + \beta_{\boldsymbol{p}} \boldsymbol{w}_i^T \boldsymbol{d}$, if we choose $\boldsymbol{w}_i : \boldsymbol{w}_i \perp \boldsymbol{d}$, and $\boldsymbol{w}_i^T \boldsymbol{s}_i = 1$. Let $\sigma$ denote the sigmoid function. Since stronger values of $\alpha_{\boldsymbol{p}}$ indicate more confidence in concept presence, when combined with a threshold $b_i \in \mathbb{R}$ that can be learned from data, this filter can be turned into a **concept detector**: $y_{\boldsymbol{p},i} = \sigma(z_{\boldsymbol{p},i} - b_i)$, essentially a binary classifier that can answer the question of whether $\boldsymbol{p}$ belongs to concept $i$.

With access to the signal value, Haufe et al. (2014); Kindermans et al. (2017) offer a formula to estimate the concept's signal direction:

$$\hat{\boldsymbol{s}}_i = \frac{\text{cov}[\boldsymbol{x_p}, \alpha_{\boldsymbol{p}}]}{\text{var}[\alpha_{\boldsymbol{p}}]} = \frac{\text{cov}[\boldsymbol{x_p}, z_{\boldsymbol{p},i}]}{\text{var}[z_{\boldsymbol{p},i}]} \tag{1}$$

### 3.3 Unsupervised Interpretable Direction Learning

Recent research (Doumanoglou et al., 2023) introduced an unsupervised method to identify concepts from the structure of the latent space. Motivated by the directional encoding of concepts, the method partitions the latent space into linear regions, each represented by a hyperplane and a normal vector, forming clusters. Feature activations from an unlabeled *concept dataset*, possibly activations of images from the network's training set, are assigned to these clusters. The method learns $\boldsymbol{W}$ and $\boldsymbol{b}$ of a feature-to-cluster membership function, a mapping to the semantic space, with $\boldsymbol{y_p} = \sigma(\boldsymbol{W}^T \boldsymbol{x_p} - \boldsymbol{b}) \in [0,1]^I, \boldsymbol{W} \in \mathbb{R}^{D \times I}, \boldsymbol{b} \in \mathbb{R}^I$, and $I$ as the cluster count. By softly assigning features to a small number of clusters, the interpretability of the

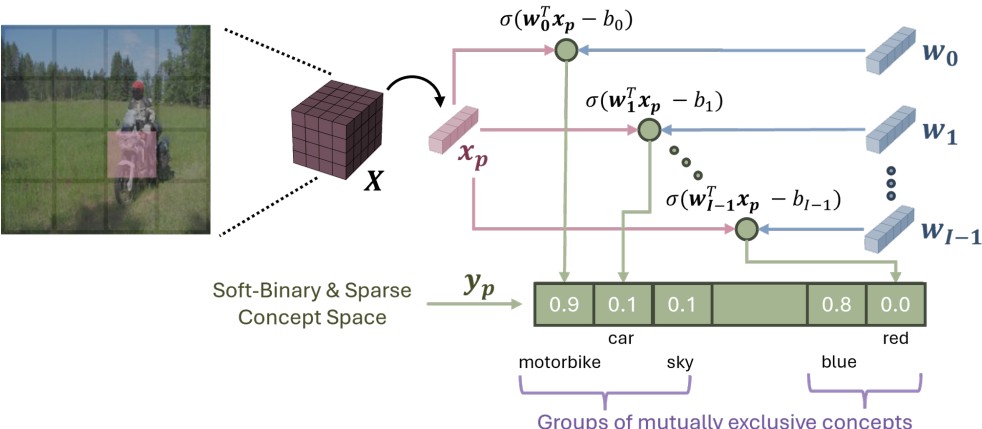

Figure 3: The core concept of Unsupervised Interpretable Basis Extraction (UIBE Doumanoglou et al. (2023)) is to learn a set of concept detectors, which are essentially binary linear classifiers with learnable filters and biases. These detectors aim to transform feature representations to the soft-binary vector space of concepts in which the newly transformed representations are sparse. In this procedure the input to the method consists of image representations coming from an unlabeled concept dataset. Identifying the concept name behind each detector is done in a post-processing step with a procedure we refer to as *Direction Labeling.*

clustering is improved. This is grounded in the idea that an image patch generally holds only a few semantic labels from a larger set, reflecting sparsity in the semantic space. Sparsity in the assignments is achieved using two loss terms: the first is *Sparsity Loss* ($\mathcal{L}^s$), and the second is *Maximum Activation Loss* ($\mathcal{L}^{ma}$), which ensures binary cluster membership:

$$\mathcal{L}^s = \mathbb{E}_{\boldsymbol{p}}\big[\mathcal{L}^s_{\boldsymbol{p}}\big], \quad \mathcal{L}^{ma} = -\mathbb{E}_{\boldsymbol{p}}\big[\boldsymbol{q}^T_{\boldsymbol{p}} \log_2(\boldsymbol{y_p})\big],$$
$$\mathcal{L}^s_{\boldsymbol{p}} = \mathcal{H}(\boldsymbol{q_p}), \quad \boldsymbol{q_p} = \frac{\boldsymbol{y_p}}{||\boldsymbol{y_p}||_1} \tag{2}$$

with $\mathcal{H}$ denoting entropy. Under a different interpretation perspective, a column of $\boldsymbol{W}$ together with a corresponding element of $\boldsymbol{b}$ (i.e., $\boldsymbol{w}_i, b_i$) forms a linear classifier or concept detector $y_{\boldsymbol{p},i} = \sigma(\boldsymbol{w}^T_i \boldsymbol{x_p} - b_i)$, which means that $\boldsymbol{w}_i$ is a filter and thus acting as a decoding direction. This method also optimizes linear separability by minimizing the inverse of the classification margin $M_i = \frac{1}{||\boldsymbol{w}_i||_2}$ (*Maximum Margin Loss - $\mathcal{L}^{mm}$*) and penalizes clusters with few assignments using the *Inactive Classifier Loss - $\mathcal{L}^{ic}$* (Doumanoglou et al., 2024) (See Fig. 3 and more details in Section A.1).

### 3.4 Direction Labeling

When learning directions in an unsupervised manner, the name of the concept represented by each encoding-decoding direction pair is unknown. In a real-world setting, the concept name could be identified by manually inspecting the samples that (maximally) activate the decoding direction. However, in a benchmark setting, where interpretability needs to be quantified in terms of metrics and without human involvement, it is necessary to automate the assignment of a concept label to each of the direction pairs. We refer to this process as *direction labeling* and employ Network Dissection Bau et al. (2017) for its implementation. In essence, Network Dissection assigns a semantic label to each of the concept detectors based on their segmentation performance in a dataset with annotated concepts. Although it was originally proposed as a method to assign labels to each of the basis vectors in the natural latent space basis, Network Dissection is capable of assigning labels to any set of directions after basis change. Despite possible biases against unsupervised learning due to annotation limitations, we adopt and expand on this labeling protocol as a best-effort approach to evaluate the interpretability of our concept detectors and the rest of the unsupervised baselines.

### 3.5 Concept Sensitivity Testing

Given the intermediate representation of an image belonging to class $k$ and the direction of concept $c$ in the latent space, RCAV (Pfau et al., 2020) measures the sensitivity of the model to concept $c$ when predicting class $k$. This is accomplished by perturbing the representation towards the direction of the concept with strength $\alpha$, and subsequently comparing the output probability of the network for the same class before and after the perturbation. Subsequently, an overall dataset score in the range $[-1, 1]$ is computed, where zero means inconsistent use of the concept by the model, while extremes indicate consistent positive or negative concept contributions to predicting class $k$. A statistical test compares concept sensitivity against sensitivity towards random directions to ensure significance. In our evaluations, we refer to *directions of significant influence* in cases where the directions meet the criteria of this statistical significance test. Although the initial work used CAVs to test concept sensitivity, a recent study (Pahde et al., 2024) suggests that the appropriate direction for this purpose is the concept's encoding direction, an aspect that we consider in our evaluations.

## 4 Proposed Method

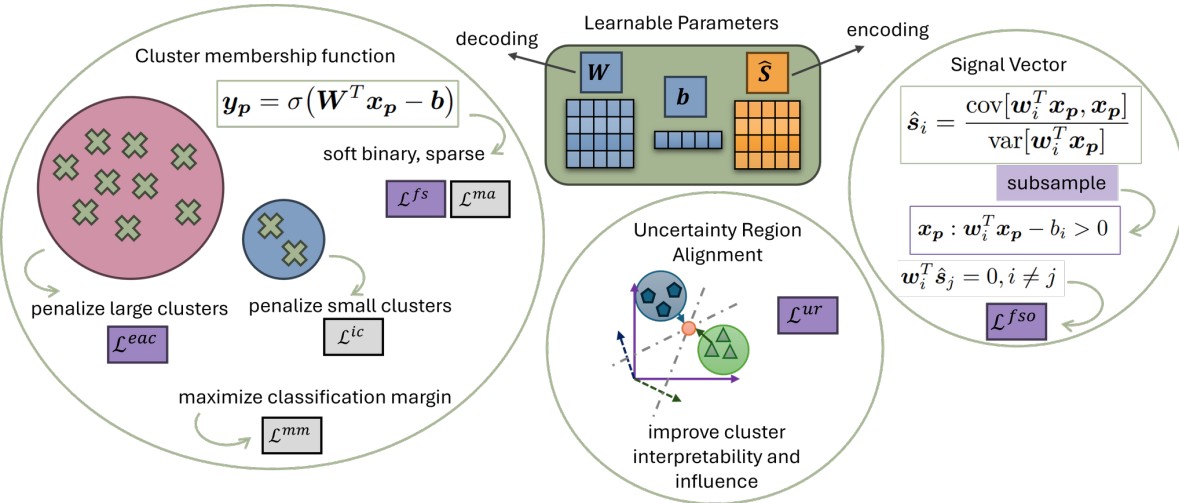

Figure 4: The proposed method analyzes the latent space to uncover its directional structure. Because many concepts are naturally encoded as specific directions, this process often reveals the encoding-decoding mechanism of meaningful, monosemantic, and highly interpretable concepts. The figure depicts an overview of the method's components. $\mathcal{L}$ denotes loss terms. **Purple** indicates contributions of this work, while **light gray** indicates loss terms from Doumanoglou et al. (2023; 2024).

**Method Overview**

The aim of our unsupervised approach is to reveal the encoding-decoding mechanism of meaningful concepts that influence the network's predictions. Our method learns encoding-decoding direction pairs that explain the structure of a deep vision network's latent space and takes as input feature representations from an unlabeled concept dataset. The latent space, from which feature representations come, corresponds to a network layer at a given depth. We jointly learn the **decoding directions (concept detectors)** $\{W, b\}$ and the **encoding directions (signal vectors)** $\hat{S}$ in an end-to-end process.

Our formulation is built upon several key contributions:

1. **Flexible and Sparser Clustering:** We lift previous constraints on the concept detectors (such as orthogonality between filters and feature space standardization) in Doumanoglou et al. (2023), enabling more flexible and less redundant (i.e., sparser) clustering. To preserve or even improve interpretability, we incorporate **two novel loss terms**: Focal Sparsity Loss and Excessively Active Classifier Loss (Section 4.2).

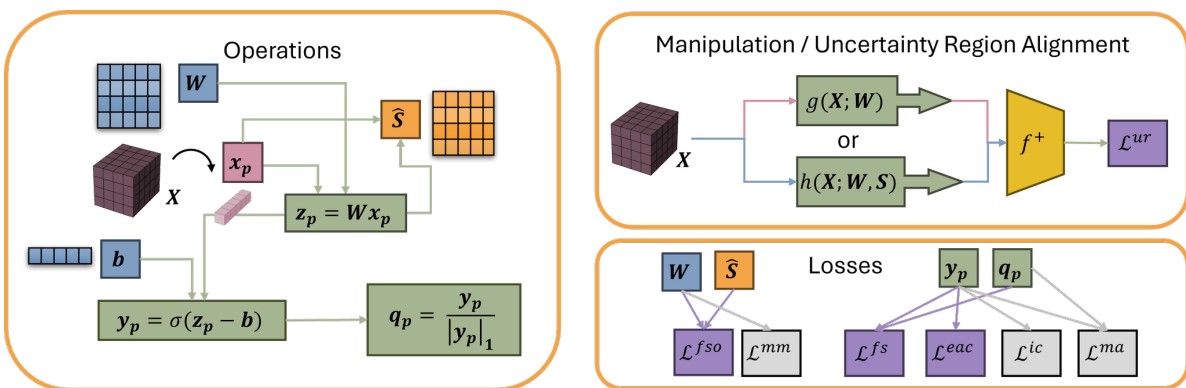

Figure 5: **Left**: The learnable parameters of the method $\hat{\boldsymbol{S}}, \boldsymbol{W}, \boldsymbol{b}$ and intermediate variables $\mathbf{z_p}, \boldsymbol{y_p}, \boldsymbol{q_p}$. **Top Right**: Feature manipulation and Uncertainty Region Alignment. **Bottom Right**: Loss terms $\mathcal{L}$ with their dependencies. **Purple** indicates loss contributions of this work, while **light gray** indicates loss terms from Doumanoglou et al. (2023; 2024).

2. **Encoding Direction Estimation:** We incorporate an **extended multi-concept signal-distractor model** (Section 4.1) and define **signal vectors** ($\hat{\mathbf{S}}$) as estimators for the concept encoding directions based on a probabilistic framework (Section 4.3).

3. **Network Instruction Driven Search:** We introduce **Uncertainty Region Alignment** ($\mathcal{L}^{ur}$) (Section 4.4), a novel loss that can improve concept interpretability and influence by aligning concept detector ambiguity with model prediction uncertainty. For the latter, the method makes use of the upper part of the network, after the layer of study, that produces output class probabilities, denoted as $f^+$.

4. **Effective Optimization:** The entire set of learning parameters is optimized jointly using an $\epsilon$-constrained optimization scheme based on the **Augmented Lagrangian Loss** (Section 4.5), which allows direct control of clustering quality with easier to tune, interpretable hyper-parameters.

Our method overview and component interconnections are shown in Figures 4 and 5. The used loss terms are described in Table 1, and the learning algorithm is outlined in Section A.2

## 4.1 Multi-Concept Signal-Distractor Data Model

We introduce an extended signal-distractor data model for the latent space, which models the encoding of **multiple** concepts. Each patch embedding $\boldsymbol{x_p}$ is considered as a linear combination of latent concept signals $\boldsymbol{S} \in \mathbb{R}^{D \times I}$ and distractors $\boldsymbol{D} \in \mathbb{R}^{D \times F}$. We also consider a latent space bias $\boldsymbol{c} \in \mathbb{R}^D$, common for all $\boldsymbol{x_p}$.

$$\boldsymbol{x_p} = \boldsymbol{S}\boldsymbol{\alpha_p} + \boldsymbol{D}\boldsymbol{\beta_p} + \boldsymbol{c} \tag{3}$$

with $\boldsymbol{\alpha_p} \in \mathbb{R}^I$ and $\boldsymbol{\beta_p} \in \mathbb{R}^F$. $\boldsymbol{S}$ is a matrix of $I \in \mathbb{N}^+$, $D$-dimensional, unit-norm concept signal directions and $\boldsymbol{D}$ a matrix denoting a basis for distractor components. Each signal direction encodes the presence of a distinct concept. We apply the same assumptions for individual signal values $\alpha_{\boldsymbol{p},i}$ (the $i$-th element of $\boldsymbol{\alpha_p}$) and distractor coefficients $\beta_{\boldsymbol{p},f}$ as in Section 3.2.

## 4.2 Losses to Preserve or Improve Interpretability

We propose **Self-Weighted Reduction** ($\mathcal{R}_{SW}$) as an aggregation method to estimate the maximum element in a set. Consider the set of elements $\{\zeta_k\}$, $\zeta \in \mathbb{R}^+, k \in \mathbb{N}$. The Self-Weighted Reduction is defined as:

$$\mathcal{R}_{SW}(\{\zeta_k\}) = \frac{\sum_k \zeta_k^{\nu+1}}{\sum_k \zeta_k^{\nu}} \tag{4}$$

which is equal to the weighted average of elements in $\{\zeta_k\}$ with each element being weighted by $\zeta_k^\nu$, $\nu \geq 1, \nu \in \mathbb{R}^+$ a sharpening factor. This aggregation may be seen as a soft differentiable version of the max operation, since the largest value in the set $\{\zeta_k\}$ is weighted with the largest weight.

**Excessively Active Classifier Loss ($\mathcal{L}^{eac}$)** This loss penalizes excessively large clusters to prevent trivial solutions in which all inputs are assigned to a single cluster. It relies on a hyper-parameter $\rho \in (0, 1)$, similar to sparse autoencoders Ng et al. (2011), which sets a proportional bound on cluster size. With $y_{\boldsymbol{p},i} \in [0, 1]$ denoting the membership of patch $\boldsymbol{p}$ for cluster $i$ (Section 3.3), the unreduced loss formula for the $i$-th cluster is below, with $\gamma > 1, \gamma \in \mathbb{R}^+$ as a sharpening factor, $1 - \rho$ normalizing the loss in the range $[0, 1]$, and $\boldsymbol{p}$ varying across all pixels and image representations in the concept dataset:

$$\mathcal{L}_i^{eac} = \frac{1}{1-\rho}\text{ReLU}(\mathbb{E}_{\boldsymbol{p}}[y_{\boldsymbol{p},i}^\gamma] - \rho) \tag{5}$$

The final loss uses $\mathcal{R}_{SW}$: $\mathcal{L}^{eac} = \mathcal{R}_{SW}(\{\mathcal{L}_i^{eac}, i \in \{0, 1, ..., I-1\}\})$

**Focal Sparsity Loss ($\mathcal{L}^{fs}$)** Inspired by Focal Loss (Lin et al. (2017)), we introduce **Focal Sparsity Loss**, which places more emphasis on sparsifying the feature-to-cluster assignments of the most challenging patch embeddings. If $\mathcal{L}_{\boldsymbol{p}}^s$ (2) denotes the *Sparsity Loss* for pixel $\boldsymbol{p}$, the *Focal Sparsity Loss* is defined as:

$$\mathcal{L}^{fs} = \frac{\sum_{\boldsymbol{p}} \theta_{\boldsymbol{p}} \mathcal{L}_{\boldsymbol{p}}^s}{\sum_{\boldsymbol{p}} \theta_{\boldsymbol{p}}} \tag{6}$$

i.e. based on a weighted scheme defined by coefficients $\theta_{\boldsymbol{p}} \in [0, 1]$. We choose $\theta_{\boldsymbol{p}}$ to be proportional to the patch's number of cluster assignments. In detail:

$$\theta_{\boldsymbol{p}} = 1 - (\mathcal{R}_{SW}(\{\boldsymbol{q}_{\boldsymbol{p},i}, i \in \{0, 1, ..., I-1\}\}))^\mu, \mu \in \mathbb{R}^+ \tag{7}$$

with $\mu$ a sharpening factor and $\boldsymbol{q}_{\boldsymbol{p}} = \frac{\boldsymbol{y}_{\boldsymbol{p}}}{||\boldsymbol{y}_{\boldsymbol{p}}||_1}$, calculated as in (2). $\mathcal{R}_{SW}\{\boldsymbol{q}_{\boldsymbol{p},i}\} \in [0, 1]$ is approximately equal to $1/k$, with $k$ denoting the patch's number of cluster assignments. Similar to the previous case, in (6) $\boldsymbol{p}$ varies across all spatial elements and image representations in the concept dataset. When we use *Focal Sparsity Loss* we use it as a replacement for the *Sparsity Loss* $\mathcal{L}^s$ of Doumanoglou et al. (2023).

### 4.3 Signal Vectors as Concept Signal Estimators

In this paragraph, we consider the new data model that was outlined in Section 4.1 and try to validate whether (1) can be used to estimate the signal direction of a concept. Suppose that our objective is to estimate the signal direction of concept $i$ when we have access to a collection of patch embeddings $\{\boldsymbol{x}_{\boldsymbol{p}}\}$ and their signal values $\{\alpha_{\boldsymbol{p},i}\}$. Starting from the approach of Kindermans et al. (2017), it is easy to prove that whenever $a_{\boldsymbol{p},i}$ is independent of all $a_{\boldsymbol{p},j}, j \neq i$ and $\beta_{\boldsymbol{p},f}$ the following property holds: $\text{cov}[\boldsymbol{x}_{\boldsymbol{p}} - a_{\boldsymbol{p},i}\boldsymbol{s}_i, a_{\boldsymbol{p},i}] = 0$, which, according to Kindermans et al. (2017) directly implies that (1) can be utilized to estimate the direction of the signal.

However, while $a_{\boldsymbol{p},i}$ can be considered independent of distractor coefficients $\beta_{\boldsymbol{p},f}$ that represent noise, the independence assumption between $a_{\boldsymbol{p},i}$ and $a_{\boldsymbol{p},j}, j \neq i$ is easily violated in practice. Some pairs of variables $a_{\boldsymbol{p},i}$ and $a_{\boldsymbol{p},j}, j \neq i$ may indeed be independent and this would be the case when concepts $i$ and $j$ belong to different groups of mutually-exclusive concepts, for example when concept $i$ belongs to the group of *objects* and concept $j$ to the group of *colors* and the concepts of the first group are independent of those of the second. However, when the concepts $i$ and $j$ belong to the same group of mutually exclusive concepts (for instance, when concept $i$ corresponds to *car* and concept $j$ to *tree* and both concepts belong to the group of *objects*), there may be an anti-correlated relationship between the variables $a_{\boldsymbol{p},j}$ and $a_{\boldsymbol{p},i}, j \neq i$ due to the fact that whenever $a_{\boldsymbol{p},i} > b_i$, $a_{\boldsymbol{p},j} < b_j$. However, the latter bias can be eliminated if we consider only samples with the concept instead of both positive and negative samples. In that case, among that subset of the data, the signal values $a_{\boldsymbol{p},i}$ and $a_{\boldsymbol{p},j}$ can be considered independent by assumption, as we now removed the biases $b_i, b_j$ due to sub-sampling. This allows us to still consider (1) as a signal estimator, even under the extended data model of multiple concepts, provided that in the computation of the covariance and variance terms of (1) we subsample the patch embeddings based on their concept label, i.e. keep only samples with

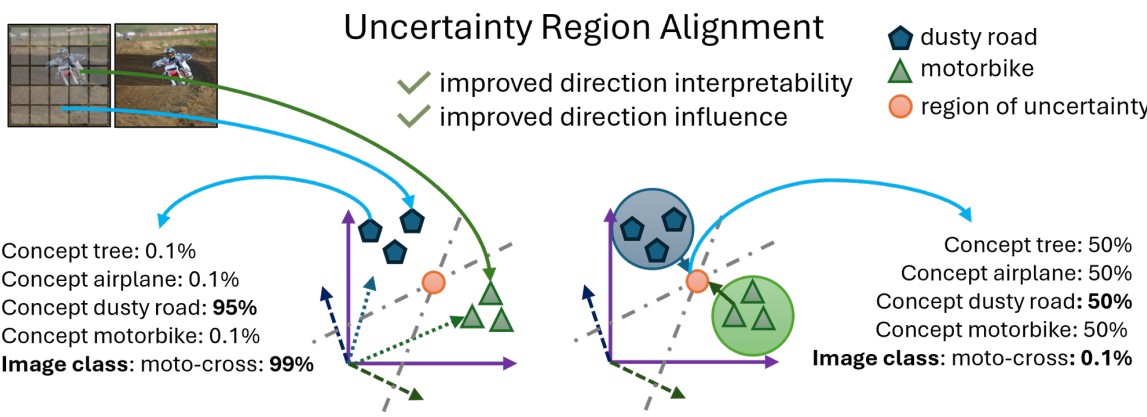

Figure 6: The uncertainty region of the network is defined as the subspace where all network's predictions are maximally uncertain. The uncertainty region of the concept detectors is defined as the intersection of all their decision hyperplanes, i.e. the subspace of ambiguous concept predictions. Aligning these two through feature manipulation may improve direction pair interpretability or serve as a balance between the interpretability and influence of the learned direction pairs.

the concept, instead of additionally considering samples without it. The latter can be easily accomplished when employing the respective concept detector.

We refer to the signal estimator of concept $i$ that is obtained under these conditions as the **signal vector** $\hat{\boldsymbol{s}}_i$. However, we still require access to the signal values. As explained in Section 3.2, estimating signal values can be attributed to the filters of the concept detectors. They can serve this purpose if the weight vector $\boldsymbol{w}_i$ is orthogonal to all $\boldsymbol{s}_j$ where $j \neq i$, as well as the distractor subspace $\boldsymbol{D}$.

Thus, we employ the following **Filter-Signal Vector Orthogonality Loss** when learning the directions:

$$\mathcal{L}^{fso} = \sqrt{\mathbb{E}_{i,j}\left[((1 - \delta_{i,j})\bar{\boldsymbol{w}}_i^T \bar{\boldsymbol{s}}_j)^2\right]} \tag{8}$$

with $\delta_{i,j}$ the kronecker delta and $\bar{\boldsymbol{w}}, \bar{\boldsymbol{s}}$ denoting the L2-normalized filter weights and signal vectors.

To achieve accurate signal value extraction, $\boldsymbol{w}_i$ should additionally be orthogonal to the distractor basis; however, we do not explicitly estimate the distractors. Instead, we use the Uncertainty Region Alignment loss from Section 4.4 to ensure alignment of the directions with utilization by the network.

## 4.4 Uncertainty Region Alignment to Discover Meaningful Concepts of Influence

The presence or absence of a concept in a representation can provide neutral, supportive, or opposing evidence against the prediction of a class. Since the concept-class pair association is unknown when learning concept directions, a straightforward strategy to perform concept arithmetic on the features in order to find their utility by the network lacks ground-truth information on how concepts affect class predictions. To overcome this difficulty, we can make a simple but more elegant hypothesis that uncertain network predictions occur when the representation has ambiguous concept information. We propose improving the direction search by aligning the uncertainty regions of the network and the concept detectors. The *uncertainty region of the network* is the subspace where its predictions are most uncertain, and the *uncertainty region of the concept detectors* is the subspace where their decision hyperplanes intersect. Figure 6 illustrates the concept of Uncertainty Region Alignment.

To accomplish the alignment, **all** the patch embeddings $\boldsymbol{x_p}$ in an image are manipulated towards the direction $-\boldsymbol{dx_p}$ to arrive at $\boldsymbol{x'_p} = \boldsymbol{x_p} - \boldsymbol{dx_p}$. Based on our estimates of $\boldsymbol{w}_i$, $b_i$, and $\hat{\boldsymbol{s}}_i$, we select the direction $\boldsymbol{dx_p}$ so that the shifted $\boldsymbol{x'_p}$ lies at the intersection of the concept detectors' decision hyperplanes, i.e. $\boldsymbol{w}_i^T(\boldsymbol{x_p} - \boldsymbol{dx_p}) - b_i = 0$, $\forall i$. Then, we ensure the network's prediction for the resulting manipulated image representation is highly

Table 1: Encoding-Decoding Direction Pairs Loss Descriptions. **Purple** indicates loss contributions of this work, while **light gray** indicates loss terms from Doumanoglou et al. (2023; 2024)

| Symbol | Loss Name | Loss Description |
|---|---|---|
| $\mathcal{L}^{fs}$ | **Focal Sparsity** | Ensure sparse patch-to-cluster assignments, emphasizing challenging cases |
| $\mathcal{L}^{ma}$ | **Maximum Activation** | Push towards binary patch-to-cluster memberships |
| $\mathcal{L}^{ic}$ | **Inactive Classifier** | Ensure all clusters have a sufficient number of members |
| $\mathcal{L}^{mm}$ | **Maximum Margin** | Ensure cluster members are linearly separated from non-cluster members by a sufficient margin |
| $\mathcal{L}^{eac}$ | **Excessively Active Classifier** | Penalize large clusters to avoid cases such as assigning all patches to a single cluster |
| $\mathcal{L}^{fso}$ | **Filter - Signal Orthogonality** | More accurate signal value extraction, enabling a more precise estimation of encoding directions |
| $\mathcal{L}^{ur}$ ($\mathcal{L}^{uur}$ or $\mathcal{L}^{cur}$) | **Uncertainty Region Alignment** | Discover interpretable concepts of influence exploiting network instructions. |

uncertain, effectively aligning both uncertainty regions. More specifically, we define two types of feature manipulation for this purpose:

**Unconstrained Feature Manipulation (UFM)** in which $\boldsymbol{x'_p}$ corresponds to the closest point of $\boldsymbol{x_p}$ in the uncertainty region of the concept detectors, i.e.: $\boldsymbol{x'_p} = g_{\boldsymbol{p}}(\boldsymbol{x_p}; \boldsymbol{W}) = \boldsymbol{x_p} - \boldsymbol{dx_p}$ and $\boldsymbol{dx_p}$ given by:

$$\boldsymbol{dx_p} = (\boldsymbol{W}^T)^+(\boldsymbol{W}^T\boldsymbol{x_p} - \boldsymbol{b}) \tag{9}$$

with $\boldsymbol{A}^+$ denoting the pseudo-inverse of $\boldsymbol{A}$ and a detailed derivation provided in Section A.4.

**Constrained Feature Manipulation (CFM)** in which we additionally restrict feature manipulation to occur within the span of the signal vectors, i.e. $\boldsymbol{dx_p} = \hat{\boldsymbol{S}}\boldsymbol{v_p}$, $\boldsymbol{v_p} \in \mathbb{R}^I$, with $\hat{\boldsymbol{S}} \in \mathbb{R}^{D \times I}$ denoting the matrix whose columns correspond to the learned signal vectors $\hat{\boldsymbol{s}}_i, i \in \{0, 1, ..., I-1\}$. The feature manipulation formula for a patch embedding $\boldsymbol{x_p}$ is $h_{\boldsymbol{p}}(\boldsymbol{x_p}; \boldsymbol{W}, \hat{\boldsymbol{S}}) = \boldsymbol{x_p} - \boldsymbol{dx_p}$, with $\boldsymbol{v_p}$ and $\boldsymbol{dx_p}$ given by:

$$\boldsymbol{v_p} = (\boldsymbol{W}^T\hat{\boldsymbol{S}})^+(\boldsymbol{W}^T\boldsymbol{x_p} - \boldsymbol{b}) \Rightarrow \tag{10}$$

$$\boldsymbol{dx_p} = \hat{\boldsymbol{S}}\boldsymbol{v_p} = \hat{\boldsymbol{S}}(\boldsymbol{W}^T\hat{\boldsymbol{S}})^+(\boldsymbol{W}^T\boldsymbol{x_p} - \boldsymbol{b}) \tag{11}$$

and again, a detailed derivation is provided in Section A.4. These manipulations are carried out for all $\boldsymbol{p}$ in the image representation $\boldsymbol{X}$ simultaneously, leading to a manipulated image representation $\boldsymbol{X'} = g(\boldsymbol{X}; \boldsymbol{W})$ or $\boldsymbol{X'} = h(\boldsymbol{X}; \boldsymbol{W}, \hat{\boldsymbol{S}})$, based on the manipulation type (Fig. 5).

Finally, the **Uncertainty Region Alignment Loss ($\mathcal{L}^{ur}$)** is:

$$\mathcal{L}^{ur} = -\mathbb{E}_{\boldsymbol{X'}}\big[\mathcal{H}(f^+(\boldsymbol{X'})\big] \tag{12}$$

with $\mathcal{H}$ denoting entropy. When we manipulate the representations via either UFM or CFM, we use the layer's activation function (typically a ReLU) to keep the features in the input domain of the next layer. Throughout the experiments, we will denote the Uncertainty Region Alignment loss with UFM as $\mathcal{L}^{uur}$ and with CFM as $\mathcal{L}^{cur}$. We consider the use of $\mathcal{L}^{uur}$ and $\mathcal{L}^{cur}$ to be mutually-exclusive. We either use the first or the second type of manipulation when computing the Uncertainty Region Alignment Loss and never both types at the same time.

## 4.5 Augmented Lagrangian Loss for Effective Direction Learning

The loss terms of the proposed method often conflict with each other. For instance, increasing confidence in the predictions of concept detectors ($\mathcal{L}^{ma}$) conflicts with maximizing the separation margin ($\mathcal{L}^{mm}$), and a

large margin may also conflict with sparsity losses ($\mathcal{L}^s$ or $\mathcal{L}^{fs}$). Simply using a weighted sum of these losses requires careful architecture and dataset specific tuning of loss weights. The latter can be laborious and only **indirectly controls** clustering quality, which is mostly determined by the final loss values. To avoid this, we adopt an $\epsilon$-constrained optimization approach and reformulate our objective as a minimization problem with inequality constraints rather than linearly combining all loss terms. In this formulation, hyper-parameter tuning primarily pertains to target loss values, providing **direct control** over the clustering quality. We use the Augmented Lagrangian formulation (Hestenes, 1969; Bertsekas, 2014) to solve the following optimization problem:

$$\min \quad \lambda^{fs}\mathcal{L}^{fs} + \lambda^{ur}\mathcal{L}^{ur} \quad s.t. \quad \begin{cases} \mathcal{L}^{ma} \leq \tau^{ma} \\ \mathcal{L}^{ic} \leq \tau^{ic} \\ \mathcal{L}^{mm} \leq \tau^{mm} \\ \mathcal{L}^{eac} \leq \tau^{eac} \\ \mathcal{L}^{fso} \leq \tau^{fso} \end{cases} \tag{13}$$

with $\lambda^{fs}, \lambda^{ur} \in \mathbb{R}^+$ the only loss weight coefficients. In this formulation, we assign a target value $\tau$ to each individual loss term participating in the inequalities and, as shown in the experiments, subsequently optimize for interpretability ($\mathcal{L}^{fs}, \mathcal{L}^{uur}$), or a balance between interpretability $\mathcal{L}^{fs}$ and influence $\mathcal{L}^{cur}$.

## 5 Encoding-Decoding Direction Pairs in Applications

Moving forward to the application level, in this Section, we discuss how to read and intervene on concept information (Section 5.1) and how to capitalize on the learned direction pairs to obtain local, detailed, and spatially-aware concept explanations (Section 5.2).

### 5.1 Reading Concept Information and Intervening on their Encoding

**Reading Concept Information from Patch Embeddings:** Reading concept information from a patch embedding is equivalent to estimating the signal value encoded in the representation. Filter directions can extract this value; however, this is done with a constant offset derived from the latent space bias $\boldsymbol{c}$. This offset can be neutralized by calculating the difference between two projected embeddings (in the direction of the filter), which directly equals the difference in their signal values. Thus, reading concept information is helpful to be done with respect to the concept information in a point of reference. Two points of reference that we find useful are:

- **The Average Embedding:** Estimating the signal value relative to a collection's mean, for instance a collection of samples with or without the concept.

- **Point of Uncertainty:** A point on the concept detector's decision hyperplane that represents maximum concept ambiguity. Centering around this point ensures that positive values indicate concept presence and negative values indicate concept absence.

**Intervening on Concept Encoding:** Intervention modifies an embedding's concept content by overwriting its signal value with a target value derived from a reference embedding. To encode the presence or absence of a concept in a patch embedding, we select target values based on:

- **The Average Signal Value** in representative positive or negative concept samples.

- **Top or Bottom Quantiles of Signal Values** in a collection of positive or negative concept samples.

Concrete implementation details for reading and intervening on concept information are provided in Section A.5.

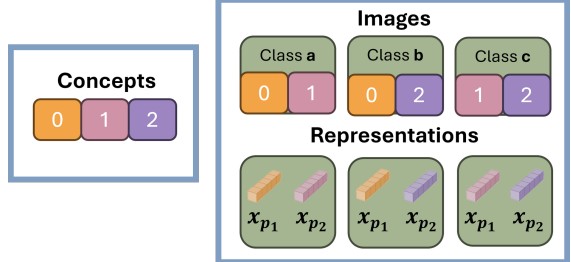

Figure 7: Description of the synthetic dataset that we use in the experiments: a) Concept set: $\{\mathbf{0,1,2}\}$ b) Image Classes: $\{\mathbf{a,b,c}\}$ with each class corresponding to a unique pair of concepts. c) Each image is comprised of two patch representations $\boldsymbol{x_{p_1}}, \boldsymbol{x_{p_2}}$.

## 5.2 Using Encoding-Decoding Direction Pairs and the Regions of Uncertainty to Provide Concept-Based Local Explanations and Detailed Spatially-Aware Concept Contribution Maps

In this Section, we provide details on how the learned Encoding-Decoding Direction Pairs can be leveraged to provide **concept-based local explanations** for a network's prediction. The core contribution is the introduction of an **explanation logit** ($l_e = l_c - l_b^m$), which is the difference between the network's class logit for the input image ($l_c$) and a class logit corresponding to a **baseline artificial representation** derived from the **uncertainty region** of the model ($l_b^m$). We then **decompose** this $l_e$ into local, concept-specific components.

In Section A.6, we show that, for networks with a Global Average Pooling (GAP) penultimate layer, the explanation logit can be **decomposed** into four key components: the **sample concept** contributions, which are input dependent, the **baseline concept** contributions from an uncertain prediction, a **correction** factor compensating for imperfect learning convergence, and a final, unexplained **residual**.

Grouping the first two components in the previous decomposition and exploiting the linearity of GAP, the explanation logit enables the derivation of **Concept Contribution Maps (CCMs)**, which provide a **spatially-aware** breakdown of the explanation logit in terms of its constituent concept contributions. CCMs differ from previous approaches (Zhou et al., 2018; Graziani et al., 2023a) in that they **do not** simply localize the concepts in the input image, but they **quantify the contributions of each concept across patches to the explanation logit**, including **both positive and negative** contributions.

The contribution of a specific concept $i$ from an image patch $\boldsymbol{p}$ to the explanation logit is calculated as the product of two factors:

- A **Global Contribution Factor**, referred to as **Concept-Class Relation Coefficient (CCRC)** ($\boldsymbol{w}_c^T \hat{\boldsymbol{s}}_i$), independent of the individual image input, with $\boldsymbol{w}_c$ denoting the network's class vector.

- A **Local Contribution Factor** ($\hat{v}_{\boldsymbol{p},i} - \hat{v}_{\boldsymbol{p},i}^b$): A local, sample dependent, spatially-aware factor representing the difference in concept information between the actual image patch ($\hat{v}_{\boldsymbol{p},i}$) and the baseline patch ($\hat{v}_{\boldsymbol{p},i}^b$).

The CCM is a heatmap that visualizes the multiplication of these two factors across the image, precisely showing which regions and concepts positively or negatively drove the final classification decision. Further details are provided in Section A.6.

## 6 Experiment on Synthetic Data

In this Section, we test our method on synthetic data which follow the data generation process detailed in Section 4.1. We validate three aspects: a) the effectiveness of signal vectors in estimating the concept's encoding direction **given ground-truth signal values**, b) the efficacy of our **unsupervised** approach to reliably identify the ground-truth concept encoding-decoding direction pairs, under challenging conditions

Table 2: Evaluating the performance of the concept detectors in classifying patch embeddings in the experiment on synthetic data. The metric is Intersection over Union (IoU). Rows correspond to concept detectors and columns to ground-truth concept classes. Clearly, each detector is aligned with one distinct ground-truth concept.

|  |  | Concept | | |
|---|---|---|---|---|
|  |  | #0 | #1 | #2 |
| Detector | #0 | 0 | 1.0 | 0 |
|  | #1 | 1.0 | 0 | 0 |
|  | #2 | 0 | 0 | 1.0 |

for conventional techniques, c) the necessity of Filter-Signal Orthogonality loss $\mathcal{L}^{fso}$ for reliable estimation of encoding directions.

We consider synthetic **image representations** with two spatial elements $\boldsymbol{p}_1, \boldsymbol{p}_2$, i.e. $W = 2$ and $H = 1$. Every pixel $\boldsymbol{p}$ is presumed to be associated with just one concept from a set of $I = 3$ concepts. Let $c(\boldsymbol{p}) \in \{0, 1, 2\}$ represent the concept label of $\boldsymbol{p}$, and $k \in \{a, b, c\}$ denote an image class. We construct **image representations** as follows: for $k = a$, $c(\boldsymbol{p}_1) = 0$ and $c(\boldsymbol{p}_2) = 1$; for $k = b$, $c(\boldsymbol{p}_1) = 0$ and $c(\boldsymbol{p}_2) = 2$; and for $k = c$, $c(\boldsymbol{p}_1) = 1$ and $c(\boldsymbol{p}_2) = 2$ (Fig. 7). We set the embedding space dimensionality to $D = 8$, the size of the distractor basis to $F = 2$ and randomly create unit-norm vectors to construct the matrices $\boldsymbol{S}$ and $\boldsymbol{D}$. Using those principles and hyper-parameters we generate **patch embeddings** $\boldsymbol{x}_{\boldsymbol{p}_1}, \boldsymbol{x}_{\boldsymbol{p}_2}$ for each image according to (3). The latent signal values and distractor coefficients follow the uniform distribution: $\alpha_{\boldsymbol{p},i} \sim \mathcal{U}(0.0, 2.25)$ if $\boldsymbol{p}$ is not part of concept $i$ and $\alpha_{\boldsymbol{p},i} \sim \mathcal{U}(2.75, 5.0)$, otherwise, while $\beta_{\boldsymbol{p},f} \sim \mathcal{U}(0, 5.0)$ is independent of the patch's concept label. We introduce a bias of $\boldsymbol{c} = 10$ across all dimensions of the representations to maintain them in the positive quartile, similar to the impact of a ReLU layer. We generate a balanced dataset with each class being represented by 1000 images.

Using this dataset, we train a network to predict the image class based on the concept content of its patches. The network we use is composed of just two layers (corresponding to the top part of a potentially larger deep network). The first is an average-pooling layer, and the second is a linear layer with $K = 3$ output classes. After training, the network attains 96.33% accuracy on a test set, randomly generated based on the previous principles. More details regarding the setup of this experiment are provided in Section A.8.

### 6.1 Evaluation of the Signal-Vector Estimator

Based on the synthetic pixel dataset $\{\boldsymbol{x}_{\boldsymbol{p}_1}\} \cup \{\boldsymbol{x}_{\boldsymbol{p}_2}\}$ and the **ground-truth signal values** $\alpha_{\boldsymbol{p},i}$, we put the estimator of Kindermans et al. (2017) and our signal vectors under test. The difference between the two estimators is the sub-sampling procedure that we proposed in Section 4.3. Given the ground-truth matrix of signal directions $\boldsymbol{S}$ we are able to calculate cosine similarities between the estimated signal directions and the respective ground-truth. The results of this experiment are depicted in Fig. 8 (in **blue** and **green**). Since we use the ground-truth signal values, this experiment evaluates the efficacy of signal vectors under perfect input conditions. The figure showcases the effectiveness of our proposed sub-sampling procedure when using (1), as in all cases, signal vectors demonstrate perfect alignment with the ground-truth directions, while in the absence of sub-sampling, the estimation is less reliable. Conclusively, the sub-sampling technique that we proposed in Section 4.3 adapts more precisely to our extended data model.

### 6.2 Evaluation of the Encoding-Decoding Direction Pairs

In this experiment, signal vectors are jointly learned with the concept detectors, employing every proposed facet of the method presented in Section 4. Concept detectors are evaluated for their ability to detect each one of the ground-truth concepts, while signal vectors are evaluated for their alignment with the ground-truth concept encoding directions.

**Decoding Directions (Concept Detectors):** Using the synthetic pixel dataset $\{\boldsymbol{x}_{\boldsymbol{p}_1}\} \cup \{\boldsymbol{x}_{\boldsymbol{p}_2}\}$, we assess the ability of each learned concept detector to detect each of the ground-truth concepts. Since the method is unsupervised, the name of the concept that each detector identifies is unknown. For this reason, we need

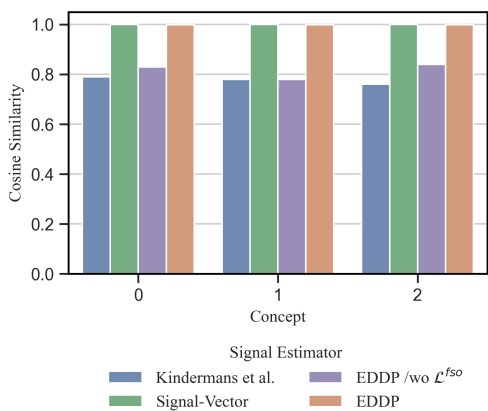

Figure 8: **1)** Cosine similarity between the ground-truth concept encoding directions and their estimation via Kindermans et al (1) calculated **without** our proposed feature sub-sampling. In this experiment, the estimation of signal directions **uses ground-truth signal values**. **2)** Cosine similarity between the ground-truth concept encoding directions and their estimation via Signal-Vectors, i.e. using (1) calculated **with** our proposed feature sub-sampling. Similarly to the previous experiment this one uses **ground-truth signal values**. Under ideal input conditions, Signal-Vectors are proven to be faithful signal estimators. **3)** Cosine similarity between ground-truth concept encoding directions and their estimation via our proposed Encoding-Decoding Direction Pairs (EDDP). In constrast to the previous two experiments, in this one Signal Vectors are learned in an **unsupervised** manner using all the aspects of the proposed method. Without access to ground-truth signal values, our approach is capable of recovering the ground-truth concept encoding directions. **4)** Cosine similarity between the ground-truth concept encoding directions and their estimation via our proposed Encoding-Decoding Direction Pairs (EDDP) but **without** using the Filter-Signal Orthogonality Loss $\mathcal{L}^{fso}$. In that case, the encoding direction estimation is inferior compared to (3).

to implement direction labeling (Section 3.4). In Table 2, we present the Intersection over Union scores for each detector against actual concept classes. Zero values indicate complete purity and no mixing of the concepts, while scores of 1.0 indicate perfect concept detection. As an additional step, we also examine how well the learned filters extract signal values from representations. Since distractors and the latent space bias $\boldsymbol{c}$ are not directly estimated by our method, we can only quantify the extracted signal values as deviations from the dataset's average value (See also Section 5.1, (29) and Section A.9). The Root Mean Squared Error (RMSE) between these extracted values and the ground truth, after subtracting the mean signal value, is noted as 0.06. We also verified that in this experiment, where the distractor components are **independent** of the concept content contained in the features, the learned filters were orthogonal to all the vectors in the distractor basis $\boldsymbol{D}$, **without** explicitly enforcing this to occur.

**Encoding Directions:** Fig. 8 - **orange** indicates perfect alignment between the learned signal vectors and the ground-truth concept encoding directions. We stress the fact that, in this experiment, signal vectors are driven by signal values extracted by the concept detectors, and this is different from Section 6.1, where signal vectors were tested under ideal input conditions.

## 6.3 Ablation Study

We also consider a case study where we learn the direction pairs by omitting $\mathcal{L}^{fso}$ from the objective function, i.e., learning the direction pairs without (/wo) it. In that case, the concept detectors still discovered the ground-truth directional clusters, as in Table 2. However, the decoding directions were less reliable in extracting the ground-truth signal values. The RMSE between the extracted values and the ground truth, after subtracting the mean signal value, was found to be 0.31, which is substantially inferior to the standard case discussed above. The latter has an impact on the fidelity of the signal vectors. Fig. 8 **purple** visualizes the cosine similarity between the learned signal vectors and the ground-truth encoding directions for this case. The experiment shows that the signal direction estimation is less effective compared to when $\mathcal{L}^{fso}$ is taken into account.

### 6.4 Evaluation against other Baselines

The ground-truth encoding directions of this example (Table 15 in Section A.8) contain both positive and negative components, and the relationship among the encoding directions (and the distractors) is, in general, not orthogonal. In theory, and without the need for practical experiments, this example cannot be addressed by NMF, K-Means, or PCA. NMF would produce a signal basis of non-negative components. Similarly, the cluster centers of K-Means would point toward the positive quartile where the centroids reside. Finally, the non-orthogonal nature of the ground truth encoding directions implies that the PCA's solution space is insufficient.

## 7 Experiments on Deep Image Classifiers

In this Section, we apply our method in real-world scenarios. In the following sections, we a) assess the faithfulness of the learned encoding-decoding direction pairs, b) provide qualitative segmentation results obtained by utilizing our learned concept detectors, c) compare with unsupervised baselines in interpretability and influence terms, d) conduct an ablation study to quantify the contribution of each proposed component, e) compare with supervised direction learning, aiming to provide more insights on the interpretability and influence of the directions, f) provide application use cases in which the learned direction pairs assist in providing global, local, and counterfactual model explanations, g) provide an example application in model correction.

### 7.1 Direction Learning and Evaluation Protocol

We apply our Encoding-Decoding Direction Pairs (EDDP) on the last convolution layer of five different networks: ResNet18 (He et al., 2016) trained on Places365 (Zhou et al., 2017), ResNet50 (He et al., 2016) trained on Moments in Time (MiT) (Monfort et al., 2019), as well as EfficientNet (b0) (Tan & Le, 2019), Inception-v3 (Szegedy et al., 2016), and VGG16 (Simonyan & Zisserman, 2014) trained on ImageNet (Deng et al., 2009). For all the networks that we study, when learning the direction pairs, we use the Broden (Bau et al., 2017) concept dataset, except for ResNet50, for which we use Broden-Action Ramakrishnan et al. (2019). The concept datasets feature dense pixel annotations; Broden includes 1197 concepts across $63K$ images in 5 concept categories (*object, part, material, texture, color*), while Broden-Action incorporates an additional *action* category with 210 labels and $23K$ more images. We emphasize that when learning the direction pairs, we do not make use of the annotations that complement the concept datasets.

For baselines, we consider directional clusterings based on either PCA, NMF, or the Natural latent space basis (referred to as *Natural*). For our EDDP, we examine two variants that are differentiated based on the choice of feature manipulation strategy used in Uncertainty Region Alignment. We use EDDP-U to refer to UFM and EDDP-C to CFM. Unless stated otherwise, our Encoding-Decoding Direction Pairs use the Augmented Lagrangian Loss from Section 4.5, all losses from Sections 4.2 and 3.3 (except $\mathcal{L}^s$), along with either $\mathcal{L}^{uur}$ (EDDP-U) or $\mathcal{L}^{cur}$ and $\mathcal{L}^{fso}$ (EDDP-C) from Sections 4.3 and 4.4.

After learning the directions with any unsupervised method, we use Network Dissection (Bau et al., 2017) and the annotations available in the concept datasets to label them. As described in Section 3.4, we make a basis change by projecting the network feature activations onto the learned decoding directions (for PCA and EDPP) or by directly utilizing the concept factors learned by NMF. More details on the experimental setup for all methods, including baselines, are provided in Section A.13, while in Section A.10, we discuss how our evaluation pipeline implicitly captures human judgment.

### 7.2 Faithfulness Assessment of the Encoding-Decoding Direction Pairs

In this Section, we aim to evaluate the faithfulness of the encoding-decoding direction pairs, i.e. to assess whether the signal value extracted by the decoding direction is indeed encoded in the direction of the signal vector. Although this was validated in synthetic data (Section 6), in this paragraph, we make the same evaluation in the case of real data, in the absence of ground truth. Based on activation maximization (Nguyen et al., 2019), we propose the following novel approach to address this evaluation.

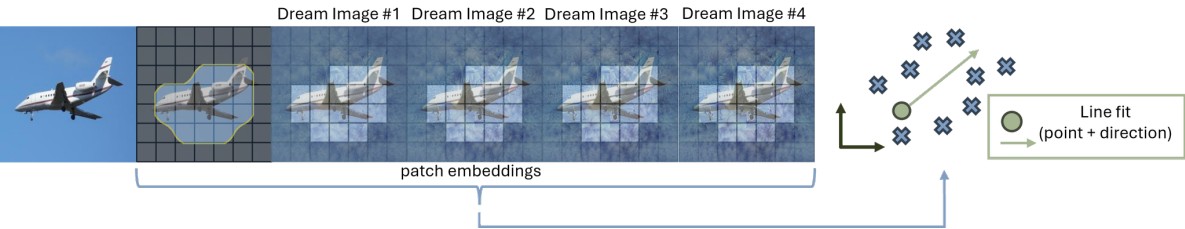

Figure 9: Our approach to assess the faithfulness of the direction pair: Starting from an image with the concept, we use deep dream to maximize the concept's decoding direction and collect the patch embeddings from the dream images at the regions with the concept. Subsequently, we fit a parametric line to those features in order to estimate the direction towards which they are moving when "dreaming". We term this direction as the *dreaming direction*.

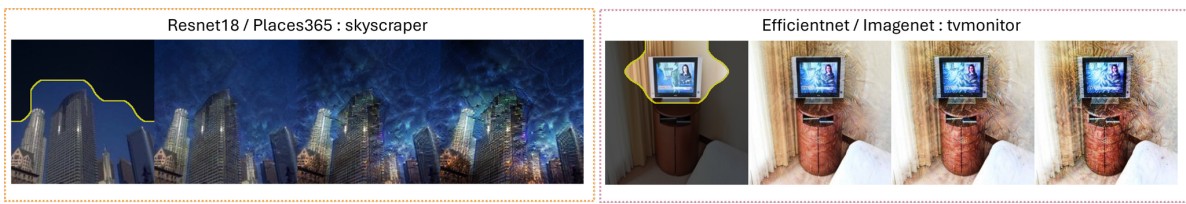

Figure 10: Dreaming pre-images from the process of estimating the dreaming directions. **Left:** Maximizing a concept detector learned for the latent space of ResNet18 and labeled with the concept name *skyscraper*. **Right:** Maximizing a concept detector learned for the latent space of EfficientNet and labeled with the concept name *tvmonitor*. Both images regard directions learned with EDDP-C.

Suppose that we evaluate the faithfulness of a particular direction pair $\boldsymbol{w}, \hat{\boldsymbol{s}}$ and let $b$ denote the respective concept detector's bias. We use the concept detector $(\boldsymbol{w}, b)$ to collect a set of images containing the concept. Subsequently, with these images as an initialization point, we use Deep Dream (Olah et al., 2017) to compute *pre-images* (Mahendran & Vedaldi, 2015; 2016), that is, images that maximize the decoding direction $\boldsymbol{w}$ at the spatial locations in which the concept was initially found. In this manner, we start from an image containing the concept and ask Deep Dream to amplify (in the input pixel space) whatever concept the detector can detect. We keep running the activation maximization loop for $K$ iterations and keep a record of how the features (at the locations with the concept) evolve while optimizing. Finally, we use all the recorded features collected during the "dreaming" optimization to fit a parametric line and estimate the direction that best describes feature evolvement while dreaming. We call the direction of this line *the dreaming direction* (Fig. 9). We conclude by comparing the dreaming direction of the concept for each image with the respective signal vector in terms of cosine similarity.

In Figures 11 and 12, we provide histograms of these cosine similarities when applying our method to ResNet18 and EfficientNet. In all cases, even those without considering $\mathcal{L}^{fso}$, approximately 90% of the signal vectors have cosine similarity with the dreaming directions above 0.7, indicating sufficient faithfulness. The effect of $\mathcal{L}^{fso}$ is to push the distribution to become more Gaussian. To validate the effectiveness of the sub-sampling strategy proposed in Section 4.3, we also provide histograms of cosine similarities between signal vectors and dreaming directions in the absence of sub-sampling. The results are depicted in Fig. 13 where, in that case, the deviation of the signal vectors from the dreaming direction is evident. Example dreaming pre-images for two cases are provided in Fig. 10 and further details and experiments may be found in Sections A.11 and A.12.

## 7.3 Interpretability Evaluation of Concept Detectors: Qualitative Segmentation Results and Interpretability Statistics

Figure 14 presents qualitative segmentation results obtained using our concept detectors, confirming that they are largely monosemantic. These visualizations rely on Network Dissection to generate segmentation maps and calculate the Intersection over Union (IoU) scores, which represent the detector's performance across the entire concept dataset's validation split. A detector qualifies as interpretable if its IoU score

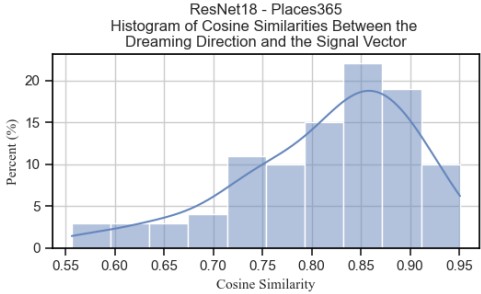 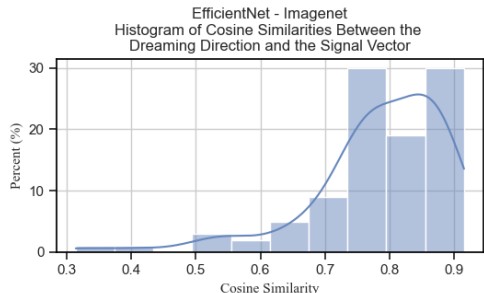

Figure 11: Cosine similarity histogram between *dreaming directions* and *signal vectors*. **Left:** Directions learned for ResNet18 trained on Places365 ($I = 448$). **Right**: Directions learned for EfficientNet trained on ImageNet ($I = 1120$). These histograms regard EDDP-C, i.e. directions learned **with $\mathcal{L}^{cur}$ and $\mathcal{L}^{fso}$**.

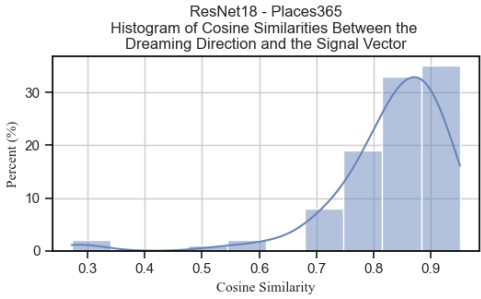 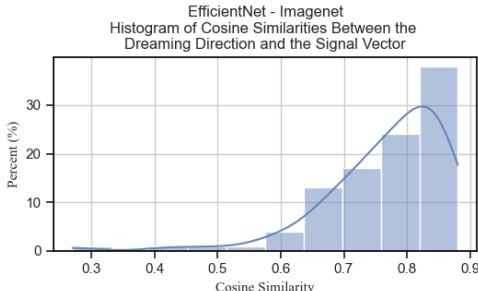

Figure 12: Cosine similarity histogram between *dreaming directions* and *signal vectors*. **Left:** Directions learned for ResNet18 trained on Places365 ($I = 448$). **Right**: Directions learned for EfficientNet trained on ImageNet ($I = 1120$). These histograms regard EDDP-U, i.e. directions learned with $\mathcal{L}^{uur}$ but **without $\mathcal{L}^{fso}$**.

surpasses the predefined threshold of 0.04. We provide detailed quantitative interpretability statistics for each network architecture, exemplified by Table 3 for ResNet18 and other Tables in Section A.17. More extensive qualitative comparisons and statistics against unsupervised baselines are provided in the same Section of the Appendix.

Table 3: Network Dissection statistics for ResNet18 trained on Places365. For each concept category in Broden, we report two numbers: First, the number of concept detectors that were labeled with the name of a concept belonging to the category and second, the number of unique concept labels from the category that have been assigned to the set of the concept detectors. The table summarizes statistics for unsupervised methods and for different values of the cluster count hyper-parameter $I$.

| | | | ResNet18 / Places365 | | | | | |
|---|---|---|---|---|---|---|---|---|
| $I$ | Method | Color | Object | Part | Material | Scene | Texture | Total |
| | PCA | 0 / 0 | 22 / 13 | 5 / 2 | 1 / 1 | 28 / 16 | 117 / 32 | 173 / 64 |
| | NMF | 0 / 0 | 94 / 42 | 7 / 4 | 2 / 2 | 189 / 101 | 47 / 26 | 339 / 175 |
| 384 | EDDP-U | 2 / 2 | 118 / 54 | 8 / 7 | 3 / 3 | 151 / 114 | 21 / 17 | 303 / 197 |
| | EDDP-C | 2 / 2 | 117 / 51 | 9 / 6 | 4 / 4 | 153 / 114 | 22 / 17 | 307 / 194 |
| | PCA | 0 / 0 | 22 / 13 | 5 / 2 | 1 / 1 | 28 / 16 | 127 / 33 | 183 / 65 |
| | NMF | 0 / 0 | 95 / 46 | 9 / 7 | 1 / 1 | 234 / 120 | 33 / 20 | 372 / 194 |
| 448 | EDDP-U | 2 / 2 | 137 / 62 | 6 / 6 | 5 / 5 | 193 / 129 | 21 / 18 | 364 / 222 |
| | EDDP-C | 2 / 2 | 136 / 60 | 5 / 5 | 5 / 5 | 194 / 133 | 22 / 18 | 364 / 223 |
| | Natural | 0 / 0 | 107 / 43 | 11 / 8 | 1 / 1 | 269 / 135 | 26 / 17 | 414 / 204 |
| | PCA | 0 / 0 | 22 / 13 | 5 / 2 | 1 / 1 | 27 / 15 | 125 / 32 | 180 / 63 |
| 512 | EDDP-U | 1 / 1 | 127 / 66 | 10 / 10 | 4 / 4 | 236 / 161 | 29 / 22 | 407 / 264 |
| | EDDP-C | 1 / 1 | 128 / 66 | 11 / 11 | 4 / 4 | 236 / 165 | 27 / 22 | 407 / 269 |

As an exemplary interpretation of the report, Table 3 shows that when learning direction pairs for ResNet18 with $I = 512$, Network Dissection reported that EDDP-C can identify 269 different concepts in the following

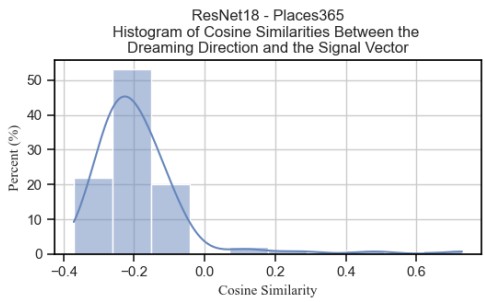
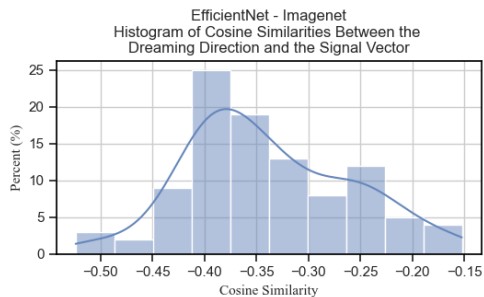

Figure 13: Cosine similarity histogram between *dreaming directions* and *signal vectors* learned **without subsampling**. **Left:** Directions learned for ResNet18 trained on Places365 ($I = 448$). **Right**: Directions learned for EfficientNet trained on ImageNet ($I = 1120$). These histograms regard EDDP-U, i.e. directions learned with $\mathcal{L}^{uur}$ but **without** $\mathcal{L}^{fso}$.

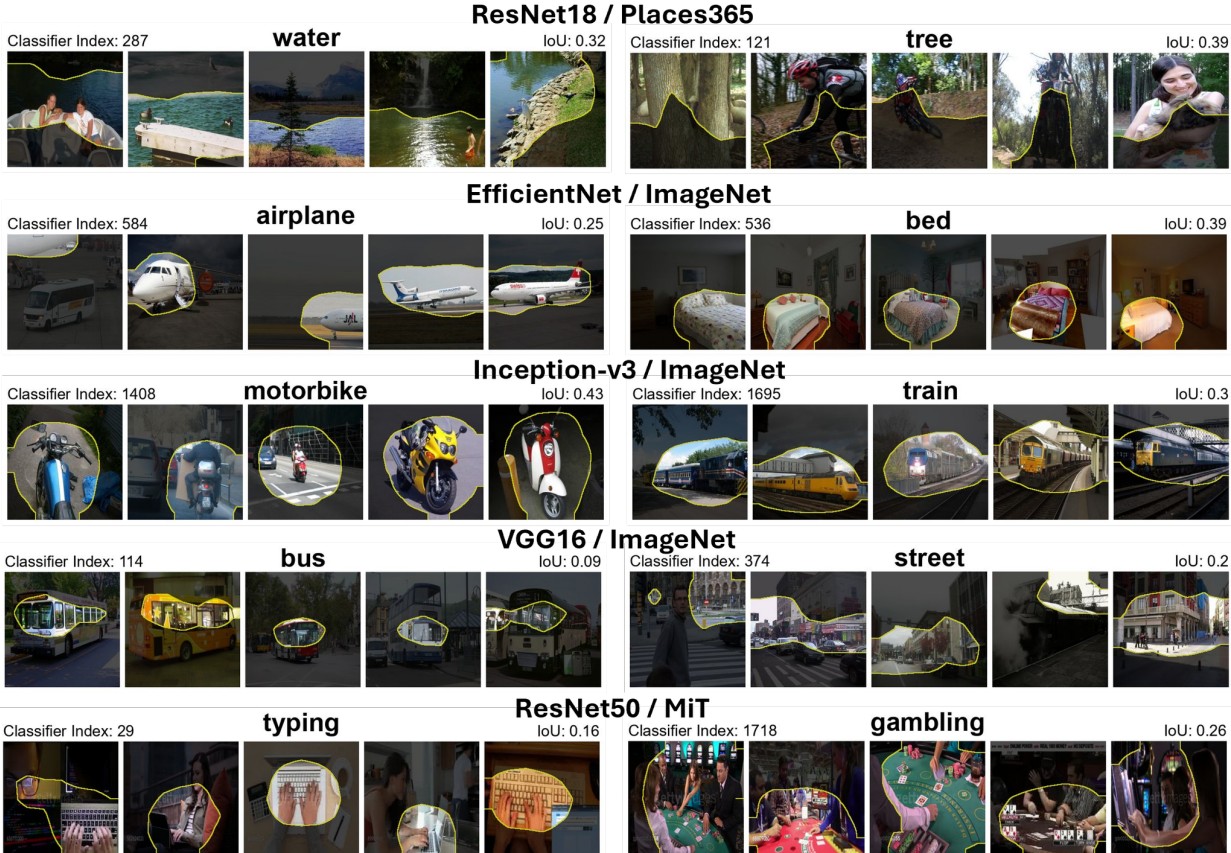

Figure 14: Qualitative segmentations using the concept detectors learned with our method as reported by Network Dissection. Rows correspond to concept detectors learned for different networks. For ResNet18 and EfficientNet the results were obtained using CFM while for Inception-v3, VGG16 and Resnet50 using UFM.

categories: 66 objects, 165 scenes, 11 parts, 4 materials, 22 textures, and 1 color. The total number of interpretable concept detectors is 407. This preliminary statistical report, together with the visualizations, is provided in order to offer intuition regarding the interpretability of the concept detectors. A rigorous evaluation with comparisons is conducted in the Sections that follow.

### 7.4 Metrics

#### 7.4.1 Evaluation of Clustering Quality via Cluster Statistics

We report two statistics for directional clustering. **Coverage** is the percentage of patches in the concept dataset that receive at least one positive prediction from the concept detectors. Equivalently, it is the percentage of pixels assigned to at least one cluster or **explained** by the clustering. **Redundancy** is the average number of detectors that classify each patch positively. Equivalently, it is the average number of clusters to which each patch belongs.

#### 7.4.2 Evaluation of the Decoding Directions: Interpretability

**Classification** Let $\mathcal{T}_i$ denote the binary classification task of predicting whether a patch belongs to concept $i$. Given the concept label $l$, which was assigned to a concept detector with direction labeling, we assess the detector's performance on $\mathcal{T}_l$ using the standard binary classification metrics of Precision, Recall, F1, and Average Precision (AP). For each metric, we report average and standard deviation scores by aggregating across all detectors in the learned set.

**Segmentation** We additionally consider the performance of concept detectors in the binary semantic segmentation task and employ two metrics from Doumanoglou et al. (2023). Specifically, let $\phi_i(c, \mathcal{K})$ the Intersection Over Union for concept detector $i$ in identifying concept $c$ within the dataset $\mathcal{K}$. Define $c_i^\star = \operatorname{argmax}_c \phi_i(c, \mathcal{K}_t)$, indicating the concept label detected best by concept detector $i$ within the training subset of the dataset ($\mathcal{K}_t$). With $\mathcal{K}_v$ as the validation subset, we use the following interpretability scores $\mathcal{S}^1$ and $\mathcal{S}^2$:

$$\mathcal{S}^1 = \int_0^1 \sum_{i=0}^{I-1} \mathbb{1}_{x \geq \xi}\big(\phi_i(c_i^*, \mathcal{K}_v)\big) d\xi \tag{14}$$

$$\mathcal{S}^2 = \int_0^1 |\{c_i^\star \mid \exists\, i : \phi_i(c_i^\star, \mathcal{K}_v) \geq \xi\}| d\xi \tag{15}$$

The first metric $\mathcal{S}^1$ counts concept detectors with an IoU performance that exceeds a score threshold $\xi$. The second metric $\mathcal{S}^2$ uses the cardinality of the set $|.|$ to count the unique concept labels detected by the concept detectors with IoU above $\xi$. Both metrics become threshold-agnostic, by integrating over all $\xi \in [0, 1]$. Viewing it from a different angle, $\mathcal{S}^1$ pertains to the segmentation efficiency of each concept detector individually, whereas $\mathcal{S}^2$ concerns the **clustering diversity**, meaning the capability of the detector ensemble to recognize a wide array of distinct concepts. Finally, we also report the standard mean Intersection over Union (mIoU) score, which is the average IoU performance score aggregated across all the detectors in the set.

**Monosemanticity** Even though Precision is a natural classification metric that can capture the monosemanticity of the clustering, it has the drawback of relying on the explicit labels that are available in the concept dataset. To overcome this labeling limitation, and inspired by Dreyer et al. (2024b), we additionally quantify the monosemanticity of the concepts identified by the concept detectors by utilizing the embedding space of CLIP ViT/B-16 (Radford et al., 2021). In particular, for each concept detector, we pick images from the validation split of the concept dataset for which the detector is most confident about the presence of the concept. We select 100 unique images from the top confident predictions. Subsequently, we crop a rectangle around the area of the concept and obtain the CLIP embedding for this crop. Let $k_1^i$ and $k_2^i$ denote image indices in the set of selected images, with $k_1^i, k_2^i \in \{0, 1, ...99\}$ and $i$ denoting the $i$-th concept detector. We use the following monosemanticity metric, which measures the average distance between CLIP embeddings $(\boldsymbol{e}_{k_1^i}, \boldsymbol{e}_{k_2^i})$ within a cluster, aggregated over all clusters:

$$\mathcal{M} = \mathbb{E}_i \left[ \mathbb{E}_{k_1^i \neq k_2^i} \big[ ||\boldsymbol{e}_{k_1^i} - \boldsymbol{e}_{k_2^i}||_2 \big] \right] \tag{16}$$

Table 4: Comparative Analysis of our Encoding-Decoding Direction Pairs (EDDP) against unsupervised baselines. This table summarizes metrics for ResNet18 trained on Places365. EDDP-U stands for using Unconstrained Feature Manipulation and EDDP-C for using Constrained Feature Manipulation.

| $I$ | Method | Coverage ↑ | Redundancy ↓ | Precision ↑ | Recall ↑ | F1 ↑ | AP ↑ | $\mathcal{M}$ ↓ | $\mathcal{S}^1$ ↑ | $\mathcal{S}^2$ ↑ | mIoU ↑ | $\mathcal{I}^1$ ↑ |
|---|---|---|---|---|---|---|---|---|---|---|---|---|
| | | | | | | | **ResNet18 / Places365** | | | | | |
| 384 | PCA | 0.72 | 7.27 | 0.26±0.13 | 0.14±0.11 | 0.18±0.11 | 0.17±0.11 | 7.47 | 16.57 | 5.6 | 0.04 | **0.73** |
| | NMF | 0.57 | 2.06 | 0.62±0.19 | 0.21±0.14 | 0.31±0.17 | 0.4±0.18 | 6.88 | 29.56 | 17.43 | 0.08 | 0.68 |
| | EDDP-U | **0.86** | **1.29** | **0.82±0.21** | **0.23±0.16** | **0.33±0.18** | **0.53±0.19** | **6.61** | **41.73** | **28.01** | **0.12** | 0.58 |
| | EDDP-C | **0.86** | 1.32 | 0.81±0.21 | **0.23±0.16** | 0.33±0.17 | 0.5±0.19 | 6.67 | 41.5 | 27.49 | **0.12** | 0.61 |
| 448 | PCA | 0.75 | 8.89 | 0.25±0.12 | 0.14±0.11 | 0.17±0.11 | 0.16±0.11 | 7.48 | 18.06 | 5.62 | 0.04 | **0.73** |
| | NMF | 0.51 | 1.79 | 0.67±0.18 | 0.21±0.14 | 0.31±0.17 | 0.43±0.18 | 6.96 | 31.77 | 18.16 | 0.07 | 0.69 |
| | EDDP-U | **0.86** | **1.32** | **0.83±0.19** | **0.22±0.16** | **0.32±0.18** | **0.53±0.19** | **6.64** | **47.5** | **32.07** | **0.11** | 0.59 |
| | EDDP-C | **0.86** | 1.33 | 0.82±0.2 | 0.21±0.16 | 0.31±0.18 | 0.49±0.19 | 6.72 | 46.3 | 31.34 | **0.11** | 0.61 |
| 512 | Natural | 0.49 | 1.79 | 0.69±0.18 | 0.2±0.13 | 0.3±0.16 | 0.43±0.17 | 7.02 | 34.61 | 18.74 | 0.07 | 0.69 |
| | PCA | 0.77 | 10.05 | 0.25±0.12 | 0.14±0.11 | 0.17±0.11 | 0.16±0.1 | 7.5 | 19.41 | 5.62 | 0.04 | **0.74** |
| | EDDP-U | **0.86** | **1.28** | **0.82±0.22** | **0.21±0.17** | **0.31±0.19** | **0.51±0.21** | **6.66** | **52.51** | **37.78** | **0.11** | 0.61 |
| | EDDP-C | **0.86** | **1.28** | 0.81±0.24 | 0.2±0.16 | 0.29±0.19 | 0.47±0.2 | 6.73 | 50.29 | 36.99 | 0.1 | 0.63 |

### 7.4.3 Evaluation of the Encoding Directions: Influence

We use RCAV Pfau et al. (2020) to assess the ability of our method to identify concepts influential to model predictions and relate concept influence to the network's concept sensitivity. In the ablation studies, two metrics summarize the results: Significant Direction Count (SDC) and Significant Class-Direction Pairs (SCDP). SDC represents the number of signal vectors that significantly influence at least one model class, while SCDP tallies class-direction pairs where the signal vectors significantly affect the class. Let $S_{C,i,k}$ denote the sensitivity of the model to concept $i$ when predicting class $k$. When comparing against unsupervised baselines, we use the following influence metric $\mathcal{I}^1$ which is defined as:

$$\mathcal{I}^1 = \mathbb{E}_{i,k}\left[\left|S_{C,i,k}\right|\right] \tag{17}$$

i.e., the average sensitivity of the model across all concepts and classes.

### 7.5 Interpretability and Influence Comparison with Established Unsupervised Baselines

The previous unsupervised state-of-the-art (Graziani et al., 2023a;b; Zhang et al., 2021; Fel et al., 2023b) has demonstrated effectiveness in revealing interpretable directions in the latent space of deep networks. However, as outlined in Section A.13, these approaches employed a **class-conditioned** protocol, tying their analyses to specific class labels. In this work, we instead operate under a **class-agnostic** protocol, which seeks concepts across the entire latent space, independent of class. This fundamental difference in the approaches renders a direct performance comparison with previous implementations inapplicable. Therefore, we utilize PCA and NMF, the core matrix decomposition methods upon which these prior works were built, as our principal baselines for a fair, classical comparison. We also move beyond subjective human ratings typically used for evaluation and instead develop a rigorous, best-effort quantitative framework to compare interpretability and influence with these baselines, with a more detailed discussion on how our pipeline captures human judgment provided in Section A.10.

In Section A.16, we also compare with the methods we extend (Doumanoglou et al., 2023; 2024), but under a different protocol for a fair comparison. For each network, we consider three sizes for the clustering, namely $I = 3/4D$, $I = 7/8D$, and $I = D$. It is worth noting that when $I = D$, NMF trivially resolves to the natural latent space basis, i.e., the neuron directions. Additionally, when studying EfficientNet, we disregard NMF since this network uses the SiLU activation function (Ramachandran et al., 2017), which allows negative feature activations, rendering NMF non-applicable. Tables 4, 5, 6, 7, and 8 summarize the comparative analysis for ResNet18, EfficientNet, Inception-v3, VGG16, and ResNet50, respectively. In the following subsections, we provide a detailed discussion of these tables.

### 7.5.1 Case Study: ResNet18 / Places365

**Clustering Quality:** Regarding dataset coverage, both our EDDP variants explain 86% of the dataset regardless of cluster count. Specifically, this means that 86% of the patch embeddings in the concept dataset have been assigned to at least one cluster. This number is substantially higher than the coverage of the clustering obtained via NMF (which is at most 57%) or the natural latent space basis (49%) and still significantly higher than the coverage of the clustering obtained via PCA (at most 77%). As far as redundancy is concerned, the EDDP variants minimize multiple cluster assignments to individual patches, as they attain the lowest redundancy scores (at most 1.33 cluster assignments per patch). While NMF and the natural basis rank 3rd in redundancy terms with more than 1.77 assignments per patch, PCA is ranked last with the same number exceeding 7.27.

**Interpretability:** In classification metrics, the clustering of EDDP-U is the most interpretable of all, scoring higher than alternative clusterings, especially in Precision and AP terms. In most cases, clustering with EDDP-C attains scores close to clustering with EDDP-U. Thus, EDDP-C is ranked 2nd in all classification metrics except in the case of F1 score and $I = 512$, in which the natural basis is slightly better by a score point of 0.01. With only the latter exception, in all classification metrics, NMF and the natural basis are ranked 3rd, while PCA is ranked last.

The high precision of our concept detectors (approximately 0.82 on average) indicates that the learned directions are highly monosemantic. The latter is additionally confirmed by the monosemanticity metric based on CLIP embedding distances, in which EDDP variants are attributed the lowest $\mathcal{M}$ score. Comparing with the rest of the approaches, all of the clusterings obtained by NMF, the natural basis, and PCA are less monosemantic, often by a large margin (0.67 precision for NMF, 0.26 precision for PCA and 0.69 for the natural basis).

Similar conclusions can be drawn from the segmentation metrics, in which the concept detectors of EDDP score substantially higher than the previous approaches. For example, when $I = 448$, EDDP-C achieves 46.3 score points in $\mathcal{S}^1$, surpassing NMF's 31.77, and reaches 0.11 mIoU points in contrast to NMF's 0.07. Finally, the EDDP variants also achieve the most diverse clustering, covering a variety of visual concepts, as justified by the substantially higher values of $\mathcal{S}^2$ compared to the rest of the approaches. Furthermore, this diversity is consistently improved with the increase in cluster count, a phenomenon that is less prominent for the rest of the baselines.

**Influence:** To rank the methods based on the network's sensitivity to the identified concepts (metric $\mathcal{I}^1$), starting from the most influential clustering, we would list PCA first, then NMF, followed by the Natural basis, and finally EDDP-C and EDDP-U. In this case, it appears that there is a clear anti-correlated relationship between interpretability and influence, with the most interpretable clustering being less influential on the overall model outcomes. We will be discussing this further at the end of the section.

### 7.5.2 Case Study: EfficientNet / ImageNet

Table 5: Comparative Analysis of our Encoding-Decoding Direction Pairs (EDDP) against unsupervised baselines. This table summarizes metrics for EfficientNet (b0) trained on ImageNet. EDDP-U stands for using Unconstrained Feature Manipulation and EDDP-C for using Constrained Feature Manipulation.

| | | | | EfficientNet / ImageNet | | | | | | | | |
|---|---|---|---|---|---|---|---|---|---|---|---|---|
| $I$ | Method | Coverage ↑ | Redundancy ↓ | Precision ↑ | Recall ↑ | F1 ↑ | AP ↑ | $\mathcal{M}$ ↓ | $\mathcal{S}^1$ ↑ | $\mathcal{S}^2$ ↑ | mIoU ↑ | $\mathcal{I}^1$ ↑ |
| | PCA | 0.63 | 19.29 | 0.29±0.08 | 0.13±0.07 | 0.18±0.07 | 0.19±0.07 | 7.44 | **35.26** | 2.62 | **0.04** | **0.95** |
| 960 | EDDP-U | **0.85** | 1.51 | **0.75±0.22** | **0.2±0.17** | **0.27±0.17** | **0.38±0.16** | **6.74** | 27.92 | **14.71** | 0.03 | 0.94 |
| | EDDP-C | **0.85** | **1.48** | 0.72±0.23 | **0.2±0.18** | **0.27±0.17** | **0.38±0.17** | 6.75 | 25.92 | 14.45 | 0.03 | 0.94 |
| | PCA | 0.67 | 24.7 | 0.28±0.07 | 0.13±0.07 | 0.17±0.07 | 0.19±0.07 | 7.44 | **42.33** | 2.7 | **0.04** | **0.95** |
| 1120 | EDDP-U | **0.85** | 1.61 | **0.76±0.21** | **0.2±0.17** | **0.28±0.17** | **0.39±0.17** | **6.73** | 31.36 | **16.54** | 0.03 | 0.94 |
| | EDDP-C | **0.85** | **1.54** | 0.74±0.22 | **0.2±0.18** | **0.28±0.19** | **0.39±0.17** | 6.74 | 28.62 | 16.38 | 0.03 | 0.94 |
| | Natural | 0.51 | 5.44 | 0.56±0.13 | 0.14±0.07 | 0.21±0.1 | 0.29±0.12 | 7.35 | 35.94 | 7.09 | 0.03 | **0.95** |
| 1280 | PCA | 0.68 | 26.71 | 0.28±0.08 | 0.13±0.07 | 0.17±0.07 | 0.18±0.07 | 7.44 | **48.01** | 3.12 | **0.04** | **0.95** |
| | EDDP-U | **0.85** | 1.65 | **0.74±0.21** | **0.19±0.16** | **0.27±0.16** | **0.36±0.17** | 6.81 | 40.38 | **19.12** | 0.03 | 0.94 |
| | EDDP-C | **0.85** | **1.6** | 0.73±0.21 | **0.19±0.17** | 0.26±0.17 | **0.36±0.17** | 6.88 | 36.95 | 18.98 | 0.03 | 0.94 |

**Clustering Quality:** Clusters created by the EDDP variants attain a Coverage score of 85%, while the best values for PCA and the natural basis in the same metric are 68% and 51%, respectively. In Redundancy terms, EDDP achieves scores less than 1.65, while the minimum scores for PCA and the natural basis are 19.29 and 5.44, respectively.

**Interpretability:** In terms of classification metrics, the directional clusterings obtained via EDDP-U and EDDP-C are approximately equally interpretable and notably more interpretable than the directional clustering of PCA. For instance, when $I = 1120$, the mean Precision of PCA's concept detectors is 0.28, the mean Recall 0.13, the mean F1 score 0.17, while the mean AP is equal to 0.19. For EDDP-U, the mean scores for the same metrics are 0.76 for Precision, 0.2 for Recall, 0.28 for F1 score, and 0.39 for AP. Similar conclusions about the improved interpretability of EDDP can be drawn when comparing with the natural latent space basis in the case of $I = 1280$.

Regarding monosemanticity, the concept detectors of EDDP attain a mean Precision score which is better than 0.72, in contrast to the ones of PCA, which attain mean Precision scores below 0.29. The same gap between PCA and EDDP is similarly reflected in the $\mathcal{M}$ score. For example, when $I = 1120$, the $\mathcal{M}$ score of EDDP-C is 6.74 while the score of PCA is 7.44. EDDP also demonstrates a substantial improvement in monosemanticity compared to the natural basis. EDDP-C attains a mean Precision score of 0.73, while the score of the natural basis for the same metric is only 0.56. Additionally, EDDP-C attains 6.88 points in terms of $\mathcal{M}$, which is significantly lower than the 7.35 points of the natural basis.

In the context of segmentation, when $I = 1120$, PCA's concept detectors exhibit improved performance concerning $\mathcal{S}^1$, exceeding EDDP-U by roughly 12 points. In mIoU terms, PCA still scores higher (0.04), but EDDP follows closely (0.03). Yet, PCA exhibits a narrow diversity in the clustering, discovering substantially fewer concepts than EDDP, as reflected in $\mathcal{S}^2$ in which EDDP-U surpasses PCA by up to approximately 14 points. When considering the natural basis as an alternative directional clustering, EDDP scores better in terms of $\mathcal{S}^1$ and $\mathcal{S}^2$ and equally well in mIoU.

**Influence:** With respect to the impact of the learned directions on the network predictions, EfficientNet shows a significant sensitivity to the direction set determined by PCA or the neuron directions from the natural basis, with a $\mathcal{I}^1$ score of 0.95. Even though the model is less influenced by the directions learned with EDDP, this is only by a small margin, as the $\mathcal{I}^1$ scores for EDDP variants are 0.94.

### 7.5.3 Case Study: Inception-v3 / ImageNet

Table 6: Comparative Analysis of our Encoding-Decoding Direction Pairs (EDDP) against unsupervised baselines. This table summarizes metrics for Inception-v3 trained on ImageNet. EDDP-U stands for using Unconstrained Feature Manipulation and EDDP-C for using Constrained Feature Manipulation.

| $I$ | Method | Coverage ↑ | Redundancy ↓ | Precision ↑ | Recall ↑ | F1 ↑ | AP ↑ | $\mathcal{M}$ ↓ | $\mathcal{S}^1$ ↑ | $\mathcal{S}^2$ ↑ | mIoU ↑ | $\mathcal{I}^1$ ↑ |
|---|---|---|---|---|---|---|---|---|---|---|---|---|
| | | | | Inception-v3 / ImageNet | | | | | | | | |
| 1536 | PCA | 0.7 | 32.29 | 0.26±0.12 | 0.12±0.1 | 0.16±0.1 | 0.17±0.1 | 7.39 | 54.5 | 4.19 | 0.04 | **0.95** |
| | NMF | 0.7 | 9.26 | 0.53±0.17 | 0.16±0.12 | 0.24±0.14 | 0.28±0.16 | 7.24 | 70.68 | 14.92 | 0.05 | **0.95** |
| | EDDP-U | **0.87** | 2.41 | 0.84±0.18 | **0.28±0.24** | **0.38±0.26** | 0.53±0.26 | **6.79** | **156.46** | **28.03** | **0.11** | 0.93 |
| | EDDP-C | 0.85 | **1.9** | **0.86±0.19** | 0.23±0.16 | 0.33±0.19 | **0.58±0.24** | 6.87 | 125.64 | 27.46 | 0.08 | 0.94 |
| 1792 | PCA | 0.68 | 33.26 | 0.28±0.12 | 0.12±0.1 | 0.17±0.1 | 0.17±0.1 | 7.39 | 59.18 | 4.1 | 0.03 | **0.95** |
| | NMF | 0.64 | 7.71 | 0.6±0.18 | 0.16±0.13 | 0.25±0.16 | 0.32±0.17 | 7.29 | 72.86 | 15.86 | 0.04 | **0.95** |
| | EDDP-U | **0.9** | 2.96 | 0.86±0.16 | **0.33±0.25** | **0.43±0.26** | 0.58±0.25 | **6.77** | **210.44** | **25.77** | **0.13** | 0.93 |
| | EDDP-C | 0.86 | **2.19** | **0.89±0.16** | 0.24±0.16 | 0.36±0.19 | **0.62±0.21** | 6.92 | 174.0 | 24.69 | 0.1 | 0.94 |
| 2048 | Natural | 0.63 | 10.29 | 0.58±0.17 | 0.17±0.13 | 0.25±0.15 | 0.31±0.17 | 7.35 | 84.62 | 14.27 | 0.04 | **0.95** |
| | PCA | 0.71 | 42.35 | 0.26±0.12 | 0.12±0.1 | 0.16±0.1 | 0.17±0.1 | 7.38 | 69.87 | 4.24 | 0.03 | **0.95** |
| | EDDP-U | 0.86 | **2.36** | **0.85±0.2** | **0.31±0.29** | **0.4±0.29** | **0.54±0.28** | **6.95** | **170.81** | **32.13** | **0.09** | 0.94 |
| | EDDP-C | **0.93** | 3.98 | 0.75±0.18 | 0.16±0.12 | 0.24±0.14 | 0.38±0.16 | 7.09 | 92.67 | 31.18 | 0.05 | 0.94 |

**Clustering Quality:** On par with the experiments on the previous architectures, EDDP variants explain more than the 85% of the concept dataset, as noted by the respective Coverage scores. This score is still significantly higher than the ones attained from NMF (up to 70%), the natural basis (63%) or PCA (up to 71%). As far as Redundancy is concerned, on one hand, EDDP does not exceed 3.98 cluster assignments per patch. On the other hand, NMF assigns a patch to more than 7.71 clusters, while for PCA this number

exceeds 32.29. Finally, in the clustering obtained by the use of the natural basis, each patch is assigned to 10.29 clusters.

**Interpretability:** For $I = 1536$ and $I = 1792$, the concept detectors of the EDDP variants score substantially higher in the classification metrics than the rest of the approaches. For example, when $I = 1792$, EDDP-C attains a Precision score of 0.89, whereas the same score for NMF is 0.6 and for PCA 0.28. For the same $I$, EDDP-U attains an F1 score of 0.43, which is a notable improvement over NMF's 0.25 and PCA's 0.17. Finally, in terms of AP, EDDP-C achieves a score of 0.62 points, which is significantly more competitive than the score of NMF (0.32) and PCA (0.17). For $I = 2048$, concept detectors from EDDP-U consistently score higher in the classification metrics than any other approach. Unlike what we have seen in the rest of the experiments, in that case, EDDP-C did not follow EDDP-U closely in the classification metrics. Instead, it is clearly ranked 2nd, surpassing PCA and the natural basis in Precision and AP while remaining comparable with the natural basis in Recall and F1 score.

Regarding monosemanticity, EDDP-U attains Precision scores close to 0.84, while for the cases of $I = 1536$ and $I = 1792$, EDDP-C exceeds this number, achieving scores up to 0.89 but attains a lower score of 0.75 when $I = 2048$. The best Precision score for NMF is 0.6 for $I = 1792$, while for PCA this score is 0.28. Finally, when clustering with the directions of the neurons, i.e. the natural basis, the Precision score is 0.58. When measuring monosemanticity with the CLIP embedding distances, the EDDP variants achieve a substantially lower score compared to the rest of the approaches. For instance, when $I = 1792$, the EDDP-U clustering attains a score of 6.77, EDDP-C 6.92, NMF 7.29 and PCA 7.39. Interestingly, unlike what we have seen when using the labeled dataset in terms of Precision, when $I = 2048$ EDDP-C and EDDP-U score similarly in $\mathcal{M}$, with EDDP-U attaining a score of 6.95 and EDDP-C 7.09.

As far as segmentation is concerned, the ranking of the methods is consistent across all metrics: EDDP-U comes first, followed by EDDP-C, then NMF or the natural basis, and finally PCA. Beyond just ranking, the gap in the scores between EDDP and other approaches is large. For instance, when $I = 1536$, the EDDP-U score in terms of $\mathcal{S}^1$ is 156.46 when NMF's is 70.68. Similarly, for the same case, EDDP-U's $\mathcal{S}^2$ is 28.03 when NMF's is 14.92, while for mIoU, EDDP-U's score is 0.11 when NMF's is 0.05.

**Influence:** Regarding the sensitivity of the model with respect to the identified concepts, Inception-v3 is more sensitive to the PCA and NMF directions with a $\mathcal{I}^1$ score of 0.95. The network is slightly less sensitive to the concepts identified by EDDP-C and EDDP-U, but once again by a small margin. EDDP-C attains a $\mathcal{I}^1$ score of 0.94 and EDDP-U a score in $\{0.93, 0.94\}$.

### 7.5.4 Case Study: VGG16 / ImageNet

Table 7: Comparative Analysis of our Encoding-Decoding Direction Pairs (EDDP) against unsupervised baselines. This table summarizes metrics for VGG16 trained on ImageNet. EDDP-U stands for using Unconstrained Feature Manipulation and EDDP-C for using Constrained Feature Manipulation.

| | | | | VGG16 / ImageNet | | | | | | | | |
|---|---|---|---|---|---|---|---|---|---|---|---|---|
| $I$ | Method | Coverage ↑ | Redundancy ↓ | Precision ↑ | Recall ↑ | F1 ↑ | AP ↑ | $\mathcal{M}$ ↓ | $\mathcal{S}^1$ ↑ | $\mathcal{S}^2$ ↑ | mIoU ↑ | $\mathcal{I}^1$ ↑ |
| 384 | PCA | 0.64 | 7.31 | 0.29±0.09 | 0.15±0.07 | 0.19±0.07 | 0.16±0.07 | 7.85 | 22.44 | 2.98 | 0.06 | 0.87 |
| | NMF | 0.55 | 1.6 | 0.6±0.16 | 0.17±0.11 | 0.26±0.13 | **0.31±0.14** | 7.44 | 22.73 | **9.7** | 0.06 | 0.87 |
| | EDDP-U | **0.82** | 1.17 | 0.61±0.18 | **0.18±0.09** | **0.27±0.09** | **0.31±0.12** | **6.88** | **29.98** | 9.39 | **0.08** | **0.91** |
| | EDDP-C | 0.79 | **1.07** | **0.63±0.18** | 0.16±0.09 | 0.24±0.09 | **0.31±0.12** | 6.92 | 26.23 | 8.89 | 0.07 | 0.9 |
| 448 | PCA | 0.67 | 8.91 | 0.28±0.08 | 0.15±0.07 | 0.19±0.07 | 0.16±0.06 | 7.86 | 25.8 | 3.12 | 0.06 | 0.87 |
| | NMF | 0.5 | 1.44 | **0.62±0.16** | 0.16±0.1 | 0.25±0.12 | **0.31±0.14** | 7.44 | 24.82 | 9.86 | 0.06 | 0.87 |
| | EDDP-U | **0.84** | 1.35 | 0.59±0.18 | **0.18±0.09** | **0.26±0.09** | 0.3±0.11 | **6.93** | **34.67** | **10.29** | **0.08** | **0.91** |
| | EDDP-C | 0.82 | **1.22** | 0.61±0.18 | 0.16±0.08 | 0.24±0.09 | 0.3±0.11 | **6.93** | 30.91 | 9.26 | 0.07 | 0.9 |
| 512 | Natural | 0.58 | 2.12 | 0.58±0.16 | 0.16±0.1 | 0.25±0.12 | 0.29±0.12 | 7.5 | 30.61 | 10.28 | 0.06 | 0.87 |
| | PCA | 0.69 | 9.62 | 0.29±0.08 | 0.15±0.06 | 0.19±0.07 | 0.16±0.06 | 7.87 | 30.25 | 3.51 | 0.06 | 0.86 |
| | EDDP-U | **0.85** | 1.43 | 0.61±0.18 | **0.18±0.08** | **0.27±0.09** | **0.31±0.11** | **6.95** | **41.2** | **11.11** | **0.08** | **0.91** |
| | EDDP-C | **0.85** | **1.37** | **0.62±0.18** | 0.15±0.09 | 0.24±0.09 | 0.3±0.11 | 7.09 | 33.91 | 10.83 | 0.07 | 0.9 |

**Clustering Quality:** In terms of clustering quality metrics, the EDDP variants attain higher Coverage and lower Redundancy scores than either one of NMF, PCA, or the natural basis, in a similar fashion as in the rest of the architectures.

**Interpretability:** Unlike the other architectures, in classification metrics, NMF and the EDDP variants perform on par. For example, when $I = 448$, NMF attains a Precision score of 0.62 exceeding EDDP-U's 0.59. In Recall and F1 score terms, EDDP-U achieves the highest scores of 0.18 and 0.26, respectively, while for NMF these numbers are 0.16 and 0.25. In terms of AP, NMF scores 0.31 while both EDDP variants attain a score of 0.30. In most cases, all of NMF, the natural basis, and EDDP substantially surpass the clustering of PCA in the classification metrics.

Even though EDDP, NMF, and the natural latent space basis score similarly in terms of Precision, if monosemanticity is measured via the CLIP embedding distances, the EDDP variants achieve the best scores, often by a substantial margin. For instance, when $I = 384$, NMF achieves an $\mathcal{M}$ score of 7.44, PCA a score of 7.85, while EDDP-U a score of 6.88.

Finally, in segmentation metrics, EDDP variants achieve the highest scores in $\mathcal{S}^1$ and mIoU, while in $\mathcal{S}^2$, EDDP-U is ranked 1st in two of the three cases ($I = 448$ and $I = 512$) while ranked 2nd, after NMF in the third case ($I = 384$).

**Influence:** Regarding the sensitivity of VGG16 with respect to the concepts, the network is most sensitive to the directions learned with EDDP-U ($\mathcal{I}^1$ score of 0.91), followed by EDDP-C ($\mathcal{I}^1$ score of 0.9), and finally, by NMF, the natural basis, or PCA, with $\mathcal{I}^1$ scores close to 0.87.

### 7.5.5 Case Study: ResNet50 / Moments in Time

Table 8: Comparative Analysis of our Encoding-Decoding Direction Pairs (EDDP) against unsupervised baselines. This table summarizes metrics for ResNet50 trained on Moments in Time (MiT). EDDP-U stands for using Unconstrained Feature Manipulation and EDDP-C for using Constrained Feature Manipulation.

| | | | | ResNet50 / MiT | | | | | | | | |
|---|---|---|---|---|---|---|---|---|---|---|---|---|
| $I$ | Method | Coverage ↑ | Redundancy ↓ | Precision ↑ | Recall ↑ | F1 ↑ | AP ↑ | $\mathcal{M}$ ↓ | $\mathcal{S}^1$ ↑ | $\mathcal{S}^2$ ↑ | mIoU ↑ | $\mathcal{I}^1$ ↑ |
| 1536 | PCA | 0.7 | 30.77 | 0.16±0.06 | 0.1±0.05 | 0.12±0.05 | 0.1±0.04 | 7.34 | 58.51 | 3.55 | 0.04 | **0.88** |
| | NMF | 0.54 | 9.07 | 0.46±0.17 | 0.16±0.11 | 0.23±0.13 | 0.27±0.13 | 6.91 | 73.38 | 17.31 | 0.05 | 0.85 |
| | EDDP-U | 0.84 | **2.13** | 0.72±0.22 | **0.17±0.12** | **0.26±0.15** | 0.36±0.16 | **6.25** | **127.83** | 33.67 | **0.08** | 0.85 |
| | EDDP-C | **0.88** | 2.37 | **0.75±0.19** | **0.17±0.11** | **0.26±0.13** | **0.45±0.14** | 6.49 | 98.52 | **34.81** | 0.06 | 0.84 |
| 1792 | PCA | 0.72 | 36.06 | 0.16±0.06 | 0.1±0.05 | 0.12±0.05 | 0.1±0.04 | 7.35 | 65.82 | 3.58 | 0.04 | **0.88** |
| | NMF | 0.42 | 6.02 | 0.57±0.15 | 0.15±0.08 | 0.23±0.1 | 0.3±0.11 | 6.95 | 77.02 | 19.0 | 0.04 | 0.85 |
| | EDDP-U | 0.85 | **2.47** | **0.78±0.16** | **0.19±0.1** | **0.29±0.13** | 0.4±0.14 | **6.24** | **162.86** | 34.17 | **0.09** | 0.85 |
| | EDDP-C | **0.88** | 2.62 | 0.76±0.16 | 0.16±0.11 | 0.25±0.13 | **0.44±0.13** | 6.5 | 113.9 | **36.68** | 0.06 | 0.84 |
| 2048 | Natural | 0.5 | 11.18 | 0.47±0.16 | 0.15±0.09 | 0.22±0.11 | 0.26±0.12 | 6.97 | 90.35 | 16.63 | 0.04 | 0.85 |
| | PCA | 0.76 | 43.2 | 0.15±0.06 | 0.1±0.05 | 0.12±0.05 | 0.1±0.04 | 7.35 | 74.3 | 3.61 | 0.04 | **0.88** |
| | EDDP-U | 0.86 | 3.15 | **0.72±0.2** | **0.21±0.13** | **0.31±0.16** | **0.43±0.16** | **6.29** | **209.2** | 33.04 | **0.1** | 0.85 |
| | EDDP-C | **0.88** | **2.83** | **0.72±0.16** | 0.15±0.11 | 0.24±0.12 | **0.43±0.13** | 6.61 | 126.46 | **35.98** | 0.06 | 0.84 |

**Clustering Quality:** Once again, in terms of clustering quality, the EDDP variants demonstrate both larger Coverage and lower Redundancy compared to the rest of the approaches, often with a large margin. When $I = 1792$, the coverage score for EDDP-C is 0.88 while for NMF it is 0.42. The gap between the two is more than twice the score of NMF. Similarly, the redundancy score of EDDP-C is 2.62 whereas NMF's is 6.02, with a similar conclusion regarding the gap between the two.

**Interpretability:** The clustering obtained via EDDP is more interpretable in terms of classification metrics, especially in Precision, AP, and sometimes in the F1 score as well. Regardless of whether monosemanticity is measured via Precision or distances between CLIP embeddings, the EDDP variants score better, similar to the rest of the experiments.

When considering segmentation, both EDDP variants demonstrate consistent superiority in terms of $\mathcal{S}^1$ and $\mathcal{S}^2$ and mIoU. Interestingly, in this experiment, EDDP-C scores the highest in $\mathcal{S}^2$, indicating a clustering that captures a larger variety of concepts.

**Influence:** In terms of the concept sensitivity metric $\mathcal{I}^1$, NMF, EDDP-C, EDDP-U and the natural basis perform on par, with an approximate score of 0.85. On the contrary, PCA achieves a score of 0.88, which puts it further ahead of the competition.

### 7.5.6 Discussion

From the previous experiments, we can draw the following, mostly straightforward conclusions: a) EDDP's directional clusters are monosemantic, often surpassing the unsupervised baselines by a significant margin. b) EDDP also excels in terms of F1 score, where EDDP-U is consistently ranked first among the competition, and EDDP-C is often ranked second. c) In terms of segmentation and specifically $\mathcal{S}^1$, EDDP-U is consistently ranked first, except for EfficientNet, in which the clustering of PCA scores better, yet with substantially inferior performance in most of the other interpretability metrics. In the same metric, EDDP-C is often ranked second. d) In terms of discovering a diverse set of concepts ($\mathcal{S}^2$) the EDDP variants score the best, often by a large margin, except for one case in VGG16 where they score on par with the best method. e) In terms of influence, in all cases except VGG16, the EDDP variants are ranked last. However, in many cases, the respective score is not far behind the competition.

In most cases, and in a broader sense, we observe that interpretability is somewhat negatively correlated with influence. Less interpretable directions may represent complex interactions (i.e. correspond to less pure human-understandable concepts) and, as a result, they are more impactful on the network predictions, in the sense that they affect many output classes. On the contrary, more interpretable directions may have a more targeted impact on specific output classes (Gao et al., 2025), and thus score lower in the respective metric.

## 7.6 Ablation Study

### 7.6.1 Interpretability Losses

Table 9: Ablation study with respect to the interpretability losses. For ResNet18 $I = 448$, and for EfficientNet $I = 1120$.

| Ortho | $\mathcal{L}^{fs}$ | $\mathcal{L}^{eac}$ | Coverage ↑ | Redundancy ↓ | Precision ↑ | Recall ↑ | F1 ↑ | AP ↑ | $\mathcal{M}$ ↓ | $\mathcal{S}^1$ ↑ | $\mathcal{S}^2$ ↑ | mIoU ↑ |
|---|---|---|---|---|---|---|---|---|---|---|---|---|
| | | | | | ResNet18 / Places365 | | | | | | | |
| ✓ | ✗ | ✗ | **0.89** | 2.25 | 0.71±0.19 | **0.3±0.18** | **0.4±0.19** | 0.47±0.19 | 6.44 | **54.28** | 26.4 | **0.13** |
| ✗ | ✗ | ✗ | 0.86 | **1.09** | **0.88±0.17** | 0.18±0.12 | 0.29±0.15 | **0.54±0.17** | 6.57 | 40.92 | 19.1 | 0.1 |
| ✗ | ✓ | ✗ | 0.83 | 1.13 | 0.85±0.19 | 0.19±0.14 | 0.29±0.16 | 0.53±0.18 | 6.57 | 39.26 | 25.36 | 0.1 |
| ✗ | ✓ | ✓ | 0.85 | 1.29 | 0.83±0.19 | 0.22±0.15 | 0.31±0.17 | 0.52±0.18 | 6.64 | 47.14 | **29.0** | 0.11 |
| | | | | | EfficientNet / ImageNet | | | | | | | |
| ✓ | ✗ | ✗ | **0.9** | 3.36 | 0.67±0.19 | 0.17±0.14 | 0.25±0.14 | 0.34±0.14 | 7.0 | **46.9** | 14.42 | **0.04** |
| ✗ | ✗ | ✗ | 0.84 | **1.29** | **0.78±0.23** | **0.23±0.21** | **0.3±0.2** | 0.39±0.18 | 6.81 | 19.54 | 13.29 | 0.02 |
| ✗ | ✓ | ✗ | **0.86** | 1.59 | **0.78±0.2** | 0.2±0.16 | 0.29±0.16 | **0.4±0.17** | 6.68 | 25.42 | 14.1 | 0.03 |
| ✗ | ✓ | ✓ | 0.85 | 1.57 | 0.74±0.2 | 0.22±0.18 | **0.3±0.17** | 0.38±0.17 | 6.68 | 28.72 | **16.02** | 0.03 |

The motivation behind the loss terms that we introduced in Section 4.2 is to sustain or improve the interpretability of the clustering when moving from an orthogonal decoding direction set to a non-orthogonal one. In Table 9 we present interpretability metrics for the cases when we learn the directions with $\mathcal{L}^{fs}$, both $\mathcal{L}^{fs}$ and $\mathcal{L}^{eac}$, or without any of them. We also provide metrics for the case of orthogonal decoding directions (marked as *Ortho*). The latter is equivalent to learning with the method of Doumanoglou et al. (2023) using the Augmented Lagrangian loss of Section 4.5. We provide results regarding the directions learned for ResNet18 and EfficientNet. In these experiments, we exclude any form of Uncertainty Region Alignment.

In both network case studies, starting without our interpretability losses and gradually adding them one after the other, we observe that the different variations a) compare similarly in coverage and redundancy terms and b) achieve comparable performance in classification and monosemanticity metrics. However, the impact of the loss terms on the segmentation metrics is more significant, especially in terms of $\mathcal{S}^1$ and $\mathcal{S}^2$. We emphasize the improvement in $\mathcal{S}^2$ since this metric captures the variety of the identified concepts. Compared to the orthogonal direction set, directions learned without orthogonality constraints a) result in a less redundant clustering, b) the monosemanticity of the clustering in terms of Precision is improved, and c) the clustering becomes more diverse, capturing a variety of different concepts.

Table 10: Ablation study with respect to the Unconstrained Uncertainty Region Alignment loss. For ResNet18 $I = 448$, for EfficientNet $I = 1120$, and for ResNet50 $I = 1792$.

| Ortho | $\mathcal{L}^{uur}$ | Coverage ↑ | Redundancy ↓ | Precision ↑ | Recall ↑ | F1 ↑ | AP ↑ | $\mathcal{M}$ ↓ | $\mathcal{S}^1$ ↑ | $\mathcal{S}^2$ ↑ | mIoU ↑ |
|---|---|---|---|---|---|---|---|---|---|---|---|
| | | | | | ResNet18 / Places365 | | | | | | |
| ✓ | ✗ | **0.89** | 2.25 | **0.71±0.19** | 0.3±0.18 | 0.4±0.19 | 0.47±0.19 | **6.44** | 54.28 | **26.4** | **0.13** |
| ✓ | ✓ | 0.88 | **2.2** | **0.71±0.18** | **0.31±0.18** | **0.41±0.19** | **0.48±0.19** | 6.52 | **54.56** | 26.06 | **0.13** |
| ✗ | ✗ | 0.85 | **1.29** | **0.83±0.19** | **0.22±0.15** | 0.31±0.17 | 0.52±0.18 | **6.64** | 47.14 | 29.0 | **0.11** |
| ✗ | ✓ | **0.86** | 1.32 | **0.83±0.19** | **0.22±0.16** | **0.32±0.18** | **0.53±0.19** | **6.64** | **47.5** | **32.07** | **0.11** |
| | | | | | EfficientNet / ImageNet | | | | | | |
| ✓ | ✗ | **0.9** | 3.36 | 0.67±0.19 | **0.17±0.14** | **0.25±0.14** | 0.34±0.14 | 7.0 | 46.9 | 14.42 | 0.04 |
| ✓ | ✓ | **0.9** | **3.34** | 0.68±0.2 | **0.17±0.14** | **0.25±0.14** | 0.36±0.15 | 6.98 | **52.94** | **16.38** | **0.05** |
| ✗ | ✗ | 0.85 | **1.57** | 0.74±0.2 | **0.22±0.18** | **0.3±0.17** | 0.38±0.17 | **6.68** | 28.72 | 16.02 | **0.03** |
| ✗ | ✓ | 0.85 | 1.61 | **0.76±0.21** | 0.2±0.17 | 0.28±0.17 | **0.39±0.17** | 6.73 | **31.36** | **16.54** | **0.03** |
| | | | | | ResNet50 / MiT | | | | | | |
| ✓ | ✗ | **0.88** | 3.44 | 0.72±0.15 | 0.16±0.1 | 0.25±0.13 | 0.35±0.14 | 6.47 | 121.67 | 29.41 | 0.07 |
| ✓ | ✓ | **0.88** | **3.31** | **0.73±0.16** | **0.18±0.1** | **0.27±0.13** | **0.37±0.13** | 6.49 | **137.76** | **33.0** | **0.08** |
| ✗ | ✗ | 0.83 | **2.42** | 0.75±0.19 | 0.15±0.12 | 0.24±0.15 | 0.32±0.14 | **6.1** | 157.54 | 30.81 | **0.09** |
| ✗ | ✓ | **0.85** | 2.47 | **0.78±0.16** | **0.19±0.1** | **0.29±0.13** | **0.4±0.14** | 6.24 | **162.86** | **34.17** | **0.09** |

Table 11: Ablation study with respect to the Uncertainty Region Alignment losses. For ResNet18 $I = 448$, for EfficientNet $I = 1120$, and for ResNet50 $I = 1792$.

| $\mathcal{L}^{uur}$ | $\mathcal{L}^{cur}$ | Coverage ↑ | Redundancy ↓ | Precision ↑ | Recall ↑ | F1 ↑ | AP ↑ | $\mathcal{M}$ ↓ | $\mathcal{S}^1$ ↑ | $\mathcal{S}^2$ ↑ | mIoU ↑ | SDC ↑ | SCDP ↑ |
|---|---|---|---|---|---|---|---|---|---|---|---|---|---|
| | | | | | ResNet18 / Places365 | | | | | | | | |
| ✗ | ✗ | 0.85 | **1.29** | **0.83±0.19** | **0.22±0.15** | 0.31±0.17 | 0.52±0.18 | **6.64** | 47.14 | 29.0 | **0.11** | 291 | **2104** |
| ✓ | ✗ | **0.86** | 1.32 | **0.83±0.19** | **0.22±0.16** | **0.32±0.18** | **0.53±0.19** | **6.64** | **47.5** | **32.07** | **0.11** | 264 | 1565 |
| ✗ | ✓ | **0.86** | 1.33 | 0.82±0.2 | 0.21±0.16 | 0.31±0.18 | 0.49±0.19 | 6.72 | 46.3 | 31.34 | **0.11** | **304** | 1692 |
| | | | | | EfficientNet / ImageNet | | | | | | | | |
| ✗ | ✗ | 0.85 | 1.57 | 0.74±0.2 | **0.22±0.18** | **0.3±0.17** | 0.38±0.17 | **6.68** | 28.72 | 16.02 | **0.03** | 93 | 216 |
| ✓ | ✗ | 0.85 | 1.61 | **0.76±0.21** | 0.2±0.17 | 0.28±0.17 | **0.39±0.17** | 6.73 | **31.36** | **16.54** | **0.03** | 107 | 243 |
| ✗ | ✓ | 0.85 | **1.54** | 0.74±0.22 | 0.2±0.18 | 0.28±0.19 | **0.39±0.17** | 6.74 | 28.62 | 16.38 | **0.03** | **673** | **1484** |
| | | | | | ResNet50 / MiT | | | | | | | | |
| ✗ | ✗ | 0.83 | **2.42** | 0.75±0.19 | 0.15±0.12 | 0.24±0.15 | 0.32±0.14 | **6.1** | 157.54 | 30.81 | **0.09** | 370 | 548 |
| ✓ | ✗ | 0.85 | 2.47 | **0.78±0.16** | **0.19±0.1** | **0.29±0.13** | 0.4±0.14 | 6.24 | **162.86** | 34.17 | **0.09** | 393 | 570 |
| ✗ | ✓ | **0.88** | 2.62 | 0.76±0.16 | 0.16±0.11 | 0.25±0.13 | **0.44±0.13** | 6.5 | 113.9 | **36.68** | 0.06 | **1353** | **3913** |

### 7.6.2 Uncertainty Region Alignment

In this section, we study how Uncertainty Region Alignment may affect the interpretability of clustering and the discovery of concepts with significant influence. For this study, we consider three network architectures: ResNet18, EfficientNet, and ResNet50.

In Table 10 we compare, in quality and interpretability terms, the clusterings obtained with and without the Unconstrained Uncertainty Region Alignment loss ($\mathcal{L}^{uur}$). We consider learning the decoding directions both with and without orthogonality constraints (*Ortho*). Some simple conclusions that can be drawn from the experiments are: a) Coverage may be improved when using it but Redundancy can be slightly worsened, b) the Precision and AP of the concept detectors are consistently improved by its use, c) it consistently improves the performance of concept detectors in terms of $\mathcal{S}^1$, and d) $\mathcal{L}^{uur}$ can significantly improve the variety of concepts captured by the direction set ($\mathcal{S}^2$ metric).

In Table 11 we focus on comparing between the two Uncertainty Region Alignment variants by additionally considering metrics of concept influence. Some key takeaways drawn from these experiments for using $\mathcal{L}^{cur}$ are: a) Coverage may be improved by its use, but Redundancy can be a bit worsened, b) the concept detectors' interpretability in terms of classification metrics is slightly inferior but somewhat comparable to when using $\mathcal{L}^{uur}$, c) in two of the three cases (ResNet18 and EfficientNet) the segmentation metrics across Uncertainty Region Alignment variants are mostly comparable, but in the third case (ResNet50) $\mathcal{L}^{cur}$ leads to a more diverse clustering (improved $\mathcal{S}^2$) but at the cost of a lower $\mathcal{S}^1$ score. d) In all cases, when comparing the two uncertainty region variants, the influence metrics favor $\mathcal{L}^{cur}$. In two of the three cases (EfficientNet and ResNet50) this improvement is substantially higher than what can be achieved when using $\mathcal{L}^{uur}$. For a more complete picture regarding the effect of $\mathcal{L}^{fso}$, which is used in EDDP-C but not in EDDP-U, see also Section A.15.

Overall, by this ablation study, we could conclude that the use of $\mathcal{L}^{uur}$ can improve the diversity of the clustering by capturing a larger variety of concepts, compared to when not using it. Using $\mathcal{L}^{cur}$ instead of $\mathcal{L}^{uur}$ may perform on par or improve this diversity, while substantially increasing the influence of the learned signal vectors on the network predictions, in a statistically significant sense.

## 7.7 Insights on Interpretability and Influence in comparison to Supervised Direction Learning

In this section, we compare our method with supervised direction learning, aiming to draw insights regarding interpretability and influence of concept directions. For this reason, we experiment with the last convolutional layer of ResNet18 trained on Places365. We consider our method with the constrained feature manipulation strategy, and two supervised learning approaches: a) IBD (Zhou et al., 2018) for supervised learning of decoding directions (i.e. concept detectors) and b) Pattern-CAVs (PCAVs) (Pahde et al., 2024) for supervised estimation of concept encoding directions. The supervised concept detectors and PCAVs are learned for the labels assigned to EDDP's concept detectors at the direction labeling phase (Section 3.4).

**Interpretability:** In the interpretability analysis, we consider three variants for the proposed method: a) the exact outcome of our method, marked as *EDDP*, b) combining directions with a shared label (post initial learning) using a binary linear classification layer which classifies representations positively if any detector with the same label does (marked as *EDDP - Linear OR*), and c) considering the learned directions but optimizing the classification threshold in a supervised manner, in a post-learning step, to enhance the F1 Score (marked as *EDDP /w sup thres*). This last approach assesses direction alignment to the concept by relaxing the sparsity constraint of the method. Table 12 summarizes metrics for this comparison. Results show that the classifiers of the proposed method achieve high Precision, comparable to the supervised ones, but suffer from low recall. The latter could be related to the strict sparsity objective of EDDP, as IBD exhibits significantly higher Redundancy. Combining classifiers with the same label improves recall, which indicates that different detectors capture different parts of the dataset, while supervised optimization of the classification bias further enhances F1 scores by relaxing sparsity.

**Influence:** In the concept influence analysis, we aim to answer whether PCAVs, that correspond to directions of high interpretability due to learning them with supervision, are more influential to the network predictions than signal vectors. Since Network Dissection can assign identical labels to multiple Encoding-Decoding Direction Pairs, direct comparison with Pattern-CAVs becomes less straightforward. We propose the following metric to help us draw some conclusions. Let $j \in \{0, 1, ..., N_l - 1\}$ index EDDP's signal vectors sharing the same concept label $l$, and $S^l_{j,k}$ represent the RCAV sensitivity score of the network relative to the $j$-th signal vector of label $l$ when predicting class $k$. Similarly, let $S^l_{P,k}$ denote the sensitivity score of the network relative to the Pattern-CAV for the same label and class. Inspired by RCAV, we regard signal vectors as *noise vectors* and compare the sensitivity of the network with respect to PCAVs and signal vectors that share the same label. We define a metric $\mathcal{I}^2$, which, when above 0.5, indicates that PCAVs have more influence than signal vectors on the network predictions at the statistical significance level of $\theta = 0.05$ with

Table 12: Comparison of EDDP (with CFM) with supervised direction learning Zhou et al. (2018). Comparing between: a) standard EDDP, b) combined concept detectors (*Linear-OR*) c) EDDP with the thresholds of concept detectors learned with supervision in a post initial direction learning step (*/w sup thres*), and d) IBD: a set of classifiers learned in a supervised way. The network here is ResNet18 and $I = 512$.

| Method | Coverage ↑ | Redundancy ↓ | Precision ↑ | Recall ↑ | F1 ↑ | AP ↑ | $\mathcal{S}^1$ ↑ | $\mathcal{S}^2$ ↑ | mIoU ↑ |
|---|---|---|---|---|---|---|---|---|---|
| | | | **ResNet18 / Places365** | | | | | | |
| IBD Zhou et al. (2018) | 0.95 | 3.08 | 0.84±0.13 | 0.6±0.17 | 0.68±0.15 | 0.77±0.15 | 54.71 | 54.71 | 0.2 |
| EDDP | 0.86 | 1.28 | **0.81±0.24** | 0.2±0.16 | 0.29±0.19 | 0.47±0.2 | **50.29** | **36.99** | 0.1 |
| EDDP - Linear OR | N/A | N/A | 0.73±0.25 | 0.35±0.26 | 0.4±0.22 | **0.53±0.21** | 30.77 | 30.77 | **0.11** |
| EDDP - /w sup thres | N/A | N/A | 0.6±0.19 | **0.43±0.18** | **0.49±0.18** | 0.47±0.2 | N/A | N/A | N/A |

Bonferroni correction:

$$\mathcal{I}^2 = \mathbb{E}_{l,k}\left[\mathbb{1}(p_{l,k} < \frac{\theta}{N_l})\right]$$

$$p_{l,k} = \frac{1}{N_l}\sum_j \mathbb{1}\left(|S_{j,k}^l| \geq |S_{P,k}^l|\right)$$

For RCAV $\alpha = 5.0$, $\mathcal{I}^2$ is equal to 0.34, which is less than 0.5, indicating that the Pattern-CAVs are less influential on the network predictions than signal vectors. This outcome could also be linked to the discussion in Section 7.5.6. It appears that less interpretable directions (in this case, EDDP's) have more influence on the network predictions compared to more interpretable ones.

## 7.8 Application: Global Model Explanations via Concept Sensitivity Testing

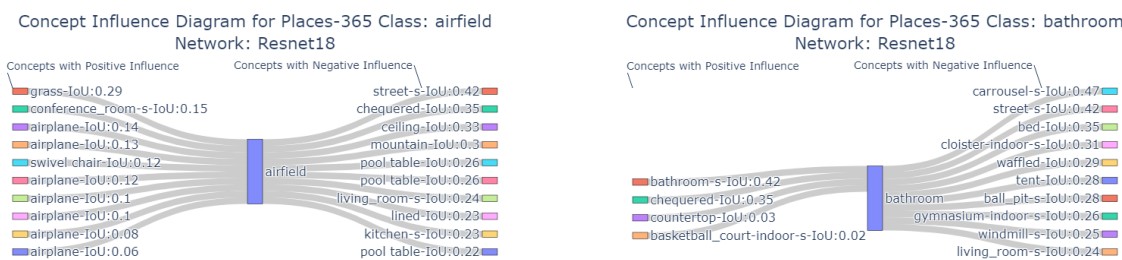

Figure 15: Concept Influence Diagram for ResNet18 trained on Places365. The model is sensitive to the depicted concepts with an absolute score above 0.99. (We use RCAV to quantify the sensitivity, and re-scale the score to $[-1, 1]$) Positive influencing and negative influencing concepts are provided. The number of concepts have been limited to 10. When concepts appear more than once, they correspond to different signal directions (as labeling the classifiers with NetDissect may assign the same concept name to more than one directions.) Here we report results for EDDP-C and $I = 512$.

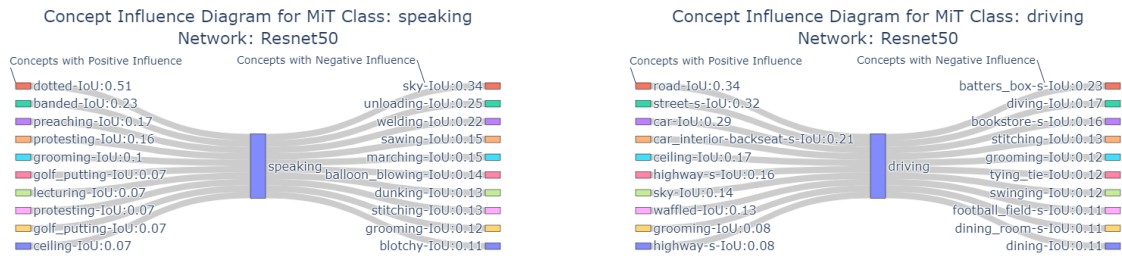

Figure 16: Concept Influence Diagram for ResNet50 trained on Moments in Time (MiT). The model is sensitive to the depicted concepts with an absolute score above 0.99. (We use RCAV to quantify the sensitivity, and re-scale the score to $[-1, 1]$) Positive influencing and negative influencing concepts are provided. The number of concepts have been limited to 10. When concepts appear more than once, they correspond to different signal directions (as labeling the classifiers with NetDissect may assign the same concept name to more than one directions.) Here we report results for EDDP-C and $I = 2048$.

Figures 15,40,41,42 depict concrete examples of how each concept's signal direction impacts the ResNet18's class predictions. Figures 16, 43 and 44 depict similar examples for ResNet50. Concepts appearing more than once correspond to different directions that have been attributed the same label. Seemingly irrelevant concepts with positive influence may have three possible explanations: a) the network has some sensitivity to those concepts (as ResNet18's top1 accuracy is 56.51% and ResNet50's 28.4%) b) their impact might be

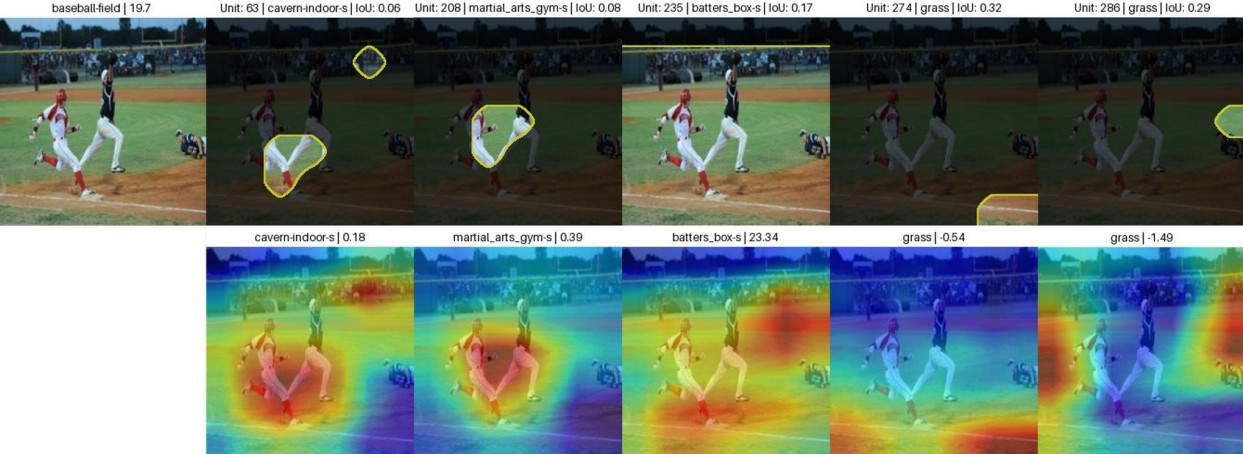

Figure 17: **Left:** Original image. The caption contains class prediction and output class logit. **Top Row:** Segmentation Maps obtained by the concept detectors. The caption contains classifier index (unit), concept-name and IoU score in the validation split of the dataset. **Bottom Row:** Concept Contribution Maps. The caption contains concept-name and contribution of the concept to the class logit.

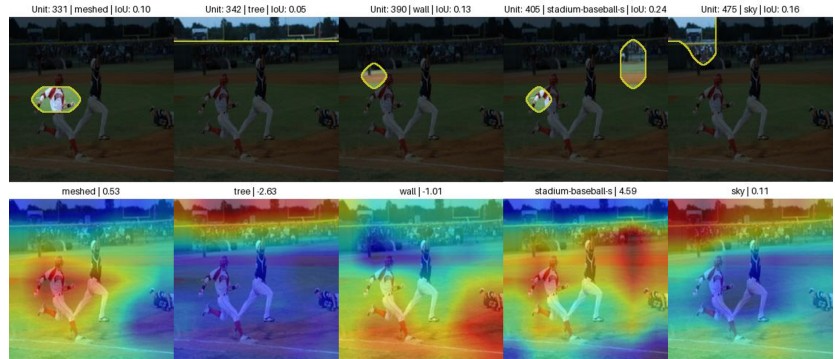

Figure 18: **Top Row:** Segmentation Maps obtained by the concept detectors. The caption contains classifier index (unit), concept-name and IoU score in the validation split of the dataset. **Bottom Row:** Concept Contribution Maps. The caption contains concept-name and contribution of the concept to the class logit.

low, since RCAV only considers the sign of the class prediction difference before and after the perturbation, regardless of its magnitude, and c) their label may be misleading as the respective concept detectors do not reliably predict the concept, possibly indicated by a low IoU score.

## 7.9 Application: Local Model Explanations via Concept Contribution Maps

In this section, we apply the concept contribution analysis of Section 5.2 to provide a detailed local explanation for a prediction of ResNet18 in an image of Places365. In this example, we use direction pairs learned via EDDP-C and $I = 512$. The example input image (Fig. 17 top left) is correctly predicted to belong to the *baseball-field* class. The output class logit for this image is $l_c = 19.7$, the baseline logit is $l_b^m = 0.0003$ and the unexplainable residual is $r = 0.2816$. The top rows of Figures 17 and 18 provide concept segmentations which were obtained by applying our concept detectors to the patch embeddings of the input and subsequently upscaled to the original resolution. Above each segmentation map, we provide the index $i$ of the respective concept detector (marked as *unit*), the name of the concept attributed to the concept detector, and the dataset-wide Intersection over Union score of the detector when detecting the concept in the validation split of Broden. Below each segmentation map, we provide the respective CCM. Each spatial element of CCM for concept $i$ contributes positively to the prediction in two cases: 1) whenever the concept content in the sample is greater than the concept content in the baseline point and the CCRC is positive, and 2) whenever

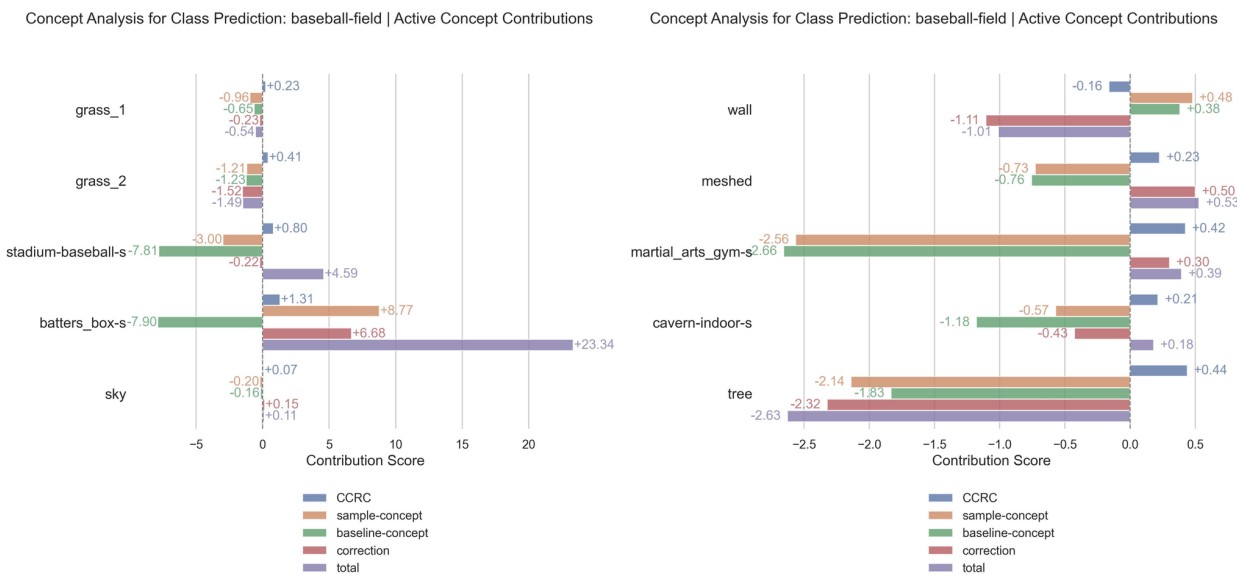

Figure 19: Concept Analysis for predicting an image of the *baseline-field* class. The figure depicts concepts found in the image. Even though concepts may share the same name, they correspond to different direction pairs.

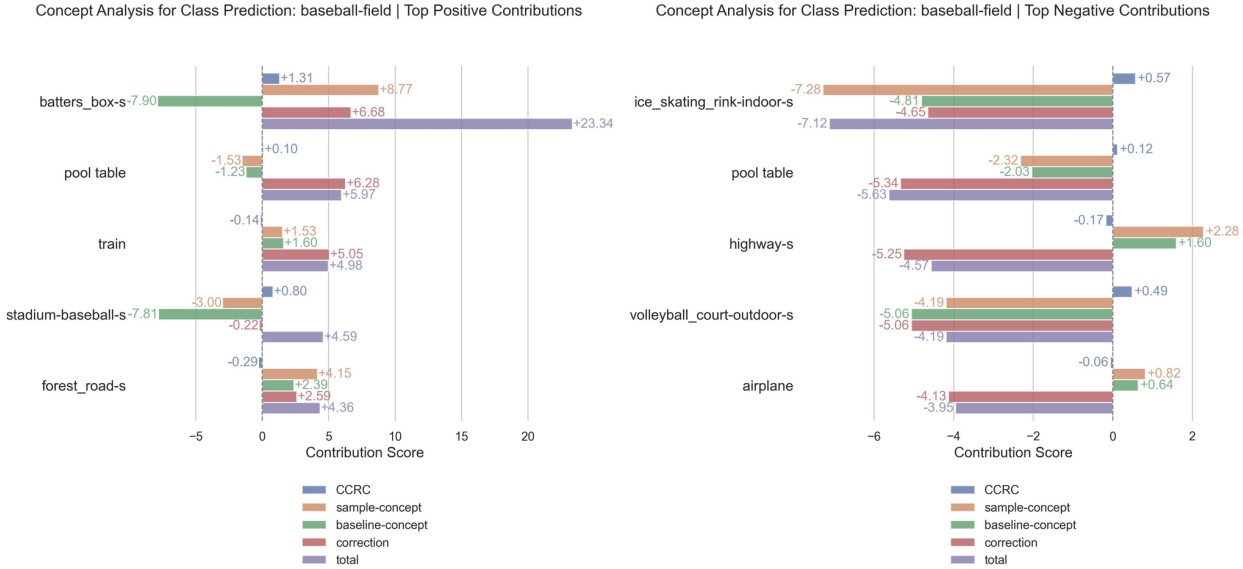

Figure 20: Concept Analysis for predicting an image of the *baseball-field* class. The figure depicts top positive and top negative contributing concepts. Even though concepts may share the same name, they correspond to different direction pairs.

the concept content in the sample is less than that in the baseline and the CCRC is negative. CCMs do not necessarily highlight the regions with the concept, as the sign of CCRC is integrated into the heatmaps (for instance: Unit 390). Above each CCM we provide the name of the concept together with the contribution of the CCM to the explanation logit after additionally integrating the correction factor.

Figures 19 and 20 depict **image-level** concept contributions by breaking down the explanation logit in components. In these Figures, the *total* equals *sample concept - baseline-concept + correction*, as discussed in Section 5.2, excluding the small *residual* term.

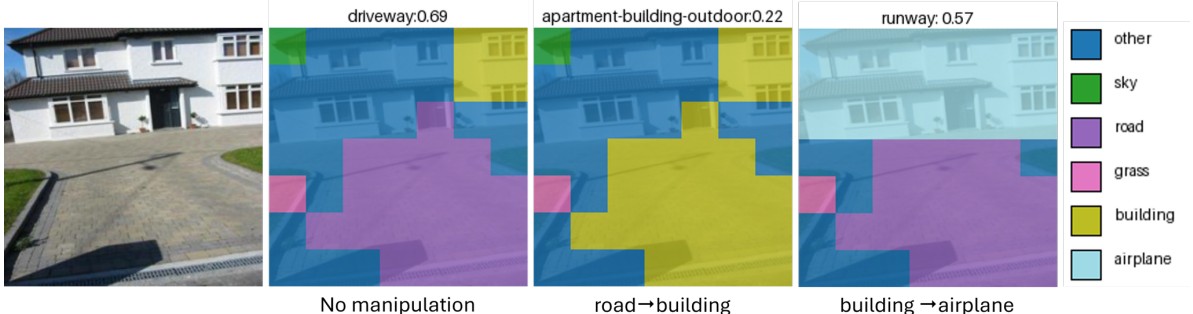

Figure 21: Counter-factual explanations via manipulating the patch embeddings in the representation space using directional concept arithmetic. **From Left to Right:** 1) original image, correctly predicted as *driveway* with confidence 69% 2) segmentation map obtained by applying the learned concept detectors on the original image representation. 3) Manipulating the latent representation and replace the *road* with the encoding of a *building*. Now the image is classified as *apartment-building-outdoor* with confidence 22%. 4) Manipulating the latent representation and replace the (whole) *building* with the encoding of an *airplane*. The manipulated image is classified as *runway* with confidence 57%.

The concepts that affect the prediction positively the most (Fig. 20) are *batters-box-s* and *stadium-baseball-s*, as well as the absence of *train* and *forest-road-s*. As we can see in the Figures, in most cases, e.g. for the concepts *grass-1, batters-box-s, stadium-baseball-s, meshed, martial-arts-gym-s, cavern-indoor-s, tree, forest-road-s,ice-skating-rink-indoor-s* the sign of the difference in concept contribution between the sample and the baseline does not change with the consideration of the correction factor. However, there are some cases in which the imperfect state of convergence when learning the directions impacts the quality of the explanation; e.g. for the concepts *grass-2,sky,wall,pool-table,train,highway-s,volleyball-court-outdoor-s,airplane*. In those cases, the consideration of the correction factor changes the sign of the total concept contribution. Many of the cases that fall into this last category exhibit marginal differences in concept contribution scores between the sample and the baseline; for instance: *grass-2, sky, wall, train, airplane*, which could justify, at least partially, this behavior.

## 7.10 Application: Counterfactual Explanations

In this Section, we provide a concrete example of how the learned direction pairs can be harnessed to provide counter-factual explanations. In this case study, we will be studying ResNet18 and direction pairs learned with CFM and $I = 512$. We consider the input image depicted in Fig. 21 - left. This image is correctly classified by the network as *driveway* with confidence 69%. The second image of the same figure depicts a segmentation map based on the learned concept detectors. To keep the visualization simple, we only show the most dominant detected concepts.

We consider two counter-factual scenarios. We study how the prediction would change if a) the *road* was replaced by a *building*, and b) if the *building* was replaced by an *airplane*. For each scenario, we perform two interventions in the representation space, one after the other: a **concept removal** (**encoding concept absence** in a patch embedding) followed by a **concept addition** (**encoding concept presence** in a patch embedding). In both scenarios, the intermediate concept removal does not alter the predicted class. However, in the first one, confidence is decreased to 30.5%, while in the second it is slightly increased to 70.4%. With concept addition, in the first scenario the prediction changes to *apartment-building-outdoor* with confidence 22%, and in the second, the prediction changes to *runway* with confidence 57%. (Fig. 21).In both cases, the change in the prediction outcome aligns with human intuition. More details for the intervention are provided in Section A.20.

Table 13: Network accuracy and confusion matrix for the network trained on the Chess Pieces dataset. Rows correspond to ground-truth labels and colums to network predictions. Three classes are considered: **b**ishop/k**n**ight/**r**ook. The rows of the confusion matrix are normalized against ground-truth element count. Three datasets are also considered: **Clean** (without watermarks), **Poisoned** (with watermarks) and **Clean & Poisoned** which is the union of the previous two.

| *Dataset:* | **Clean** | | | **Poisoned** | | | **Clean & Poisoned** | | |
|---|---|---|---|---|---|---|---|---|---|
| *Accuracy:* | 0.93 | | | 0.34 | | | 0.64 | | |
| | **b** | **n** | **r** | **b** | **n** | **r** | **b** | **n** | **r** |
| **b** | 0.95 | 0.05 | 0.0 | 0.0 | 0.0 | 1.0 | 0.48 | 0.02 | 0.5 |
| **n** | 0 | 0.95 | 0.05 | 0.0 | 0.0 | 1.0 | 0.0 | 0.48 | 0.52 |
| **r** | 0.04 | 0.04 | 0.92 | 0.0 | 0.0 | 1.0 | 0.02 | 0.02 | 0.96 |

## 7.11 Application: Toy Model Correction

In this experiment, we demonstrate how the proposed approach may be utilized to correct a toy model that relies on controlled confounding factors to make its predictions. For the purposes of this toy experiment, we use a small convolutional neural network with 5 `Conv2d` layers, each followed by a `ReLU` activation. The top of the network is comprised of a Global Average Pooling (GAP) layer and a linear head. We consider the task of predicting the chess piece name from an image depicting the piece. We use the *Chess Pieces* dataset from Kaggle [1] , which contains a collection of images depicting chess pieces appearing in online play. For simplicity, we consider 3 chess pieces to be classified by the network, namely: *bishop, knight, rook*, thus, the network predicts $K = 3$ output classes. To encourage the model to learn a bias in making its predictions, we poison half of the *rook* images in the training set with the watermark text "rook" in the top left corner of each rook image. With the introduction of this bias, we expect that the network learns that the watermark concept has a positive influence on the *rook* class, while not being the only feature of positive evidence for the same class, since we include rook images in the training set without the watermark. For further details regarding this experiment, refer to Section A.21.

### 7.11.1 Evaluation

For evaluation, we construct three datasets. First, we consider a **Clean** test set, a dataset comprised of images without any watermarks. Second, we consider a **Poisoned** test set comprised of all the images in the clean set but poisoned with the watermark, and c), we consider the union of the previous two datasets (**Clean & Poisoned**). Table 13 summarizes the performance of the network in these datasets. As evidenced by the *Poisoned* section of the detailed confusion matrix, the watermark is a strong feature that, whenever it is present, directs the prediction towards the class *rook*.

### 7.11.2 Direction Learning for Watermark Identification

Since we already identified that the watermark actually influences the predictions of the network in a consistent manner, we seek to answer the following two research questions: a) can the proposed EDDP method identify the watermark as a concept? That is, is any of the learned classifiers responsible for detecting the watermark?, b) Supposing the answer to (a) is positive, given the watermark's concept detector and the respective learned signal vector, can we fix the network to not rely on the watermark for its predictions ?

We use the network's training set as our concept dataset. We consider the last convolutional layer as our layer of study, which has a spatial dimensionality of 2x2. We found that when learning with $I = 6$, the proposed approach is able to identify the watermark direction. In Fig. 22 we provide example image segmentations obtained via the learned concept detectors. From the qualitative visualizations, we see that classifier 5 identifies the watermark. We found that this classifier detects the watermark concept with IoU 0.85.

---

[1]Chess Pieces Dataset (85x85): `https://www.kaggle.com/datasets/s4lman/chess-pieces-dataset-85x85`

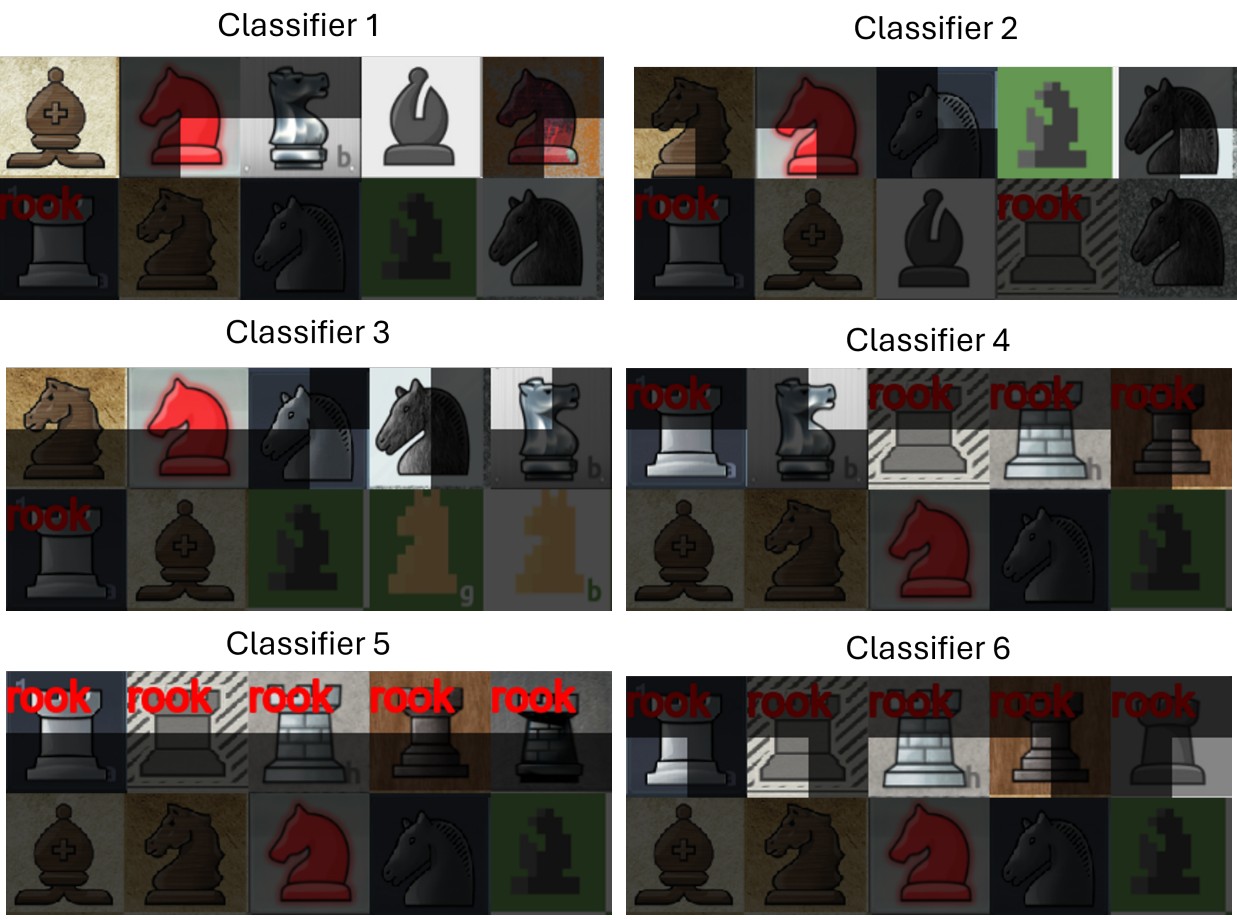

Figure 22: Example image segmentations based on the concept detectors learned for the model correction experiment. Top rows illustrate pictures with the concept and bottom rows illustrate pictures without. Classifier 5 detects the watermark.

### 7.11.3 Influence Testing with RCAV

We use RCAV to measure the sensitivity of the model with respect to the watermark signal vector. The sensitivity scores reported by RCAV are $-1$ for the classes *bishop* and *knight* , and 1 for the class *rook*. This implies that, when the image has the watermark, it becomes more *rook* and less *bishop* or *knight*. This quantitative score aligns with our intuition regarding the watermark.

### 7.11.4 Signal Vector Faithfulness to the Watermark Concept

Leveraging our knowledge of which images contain the watermark, we can determine the watermark's encoding direction by calculating the watermark's Pattern-CAV (Pahde et al., 2024). Subsequently, we quantify the directional alignment between the Pattern-CAV, which was learned with supervision, and the signal vector, which was learned without. We found that the cosine similarity between these two is 0.99, indicating estimations that agree.

### 7.11.5 Model Correction by Using the Watermark's Encoding and Decoding Directions

Let $w$ and $b$ denote the learned parameters of the watermark concept detector and $\hat{s}$ denote the respective learned signal vector. Without re-training or fine-tuning the network, we are going to suppress the watermark artifact component from the representation whenever it is detected by the concept detector. We propose the following feature manipulation strategy, which we apply to the features of the last convolutional layer:

$$\boldsymbol{x'_p} = \mathrm{ReLU}(\boldsymbol{x_p} - mk\hat{\boldsymbol{s}}), m = \sigma(\boldsymbol{w}^T \boldsymbol{x_p} - b) \tag{18}$$

with $k$ chosen such that $\boldsymbol{w}^T \boldsymbol{x'_p} = \mathbb{E}_{\boldsymbol{p} \in \mathcal{N}}\big[\boldsymbol{w}^T \boldsymbol{x_p}\big]$, $\mathcal{N} = \{\boldsymbol{p} : \boldsymbol{w}^T \boldsymbol{x_p} - b < 0\}$; see also (33). This choice of $k$ ensures that the signal value associated with the watermark concept within the patched representation matches the mean value found in the collection of patches lacking the watermark. The ReLU in (18) ensures that the manipulation does not move the features out of the domain of the linear head.

### 7.11.6 Evaluation of the Corrected Model

We evaluate the corrected model according to the protocol that was used in Section 7.11.1. The results are depicted in Table 14. Compared to the performance of the original network (Table 13), we see that the corrected model: a) has comparable accuracy to the original model on the clean test set, b) is significantly more accurate on images of the poisoned dataset with an absolute improvement of +31%, and c) performs substantially better on the union of clean and poisoned datasets with an absolute improvement of +14%.

We also compare our correction strategy to using a random manipulation direction with the same $k$ as before (i.e., using a random vector in place of the learned signal vector in (18)). In a series of 10 trial evaluations on the Poisoned Test set, we verified that in 9 out of 10 cases, **no improvement** was achieved: the classification accuracy was the same as that of the original model, and the confusion matrix remained the same as in Table 13-Middle. The same observation holds when considering the learned filter direction in place of the learned signal vector. The remaining case demonstrated an improvement of +15%, which is still inferior to +31% when using the signal vector.

Table 14: Network accuracy and confusion matrix for the **corrected** network trained on the Chess Pieces dataset. Rows correspond to ground-truth labels and columns to network predictions. Three classes are considered: **b**ishop/**kn**ight/**r**ook. The rows of the confusion matrix are normalized against ground-truth element count. Three datasets are considered: **Clean** (without watermarks), **Poisoned** (with watermarks) and **Clean & Poisoned** which is the union of the previous two.

| *Dataset:* | **Clean** | | | **Poisoned** | | | **Clean & Poisoned** | | |
|---|---|---|---|---|---|---|---|---|---|
| *Accuracy:* | 0.92 (-1%) | | | 0.65 (+31%) | | | 0.78 (+14%) | | |
| | **b** | **n** | **r** | **b** | **n** | **r** | **b** | **n** | **r** |
| **b** | 0.95 | 0.05 | 0.0 | 0.5 | 0.2 | 0.3 | 0.73 | 0.12 | 0.15 |
| **n** | 0.0 | 1.0 | 0.0 | 0.14 | 0.72 | 0.14 | 0.07 | 0.86 | 0.07 |
| **r** | 0.05 | 0.13 | 0.82 | 0.05 | 0.22 | 0.73 | 0.05 | 0.18 | 0.77 |

## 8 Limitations

Our method builds on the linear representation hypothesis, assuming that concepts are encoded as directions in the latent space. While this view aligns with several prior works, alternative formulations such as multidimensional concept discovery (Vielhaben et al., 2023) or linear subspaces (Chormai et al., 2024) exist but fall outside the scope of our study. Additionally, we constrain our experiments to settings where the number of discovered concepts is smaller than the embedding dimensionality. As such, we do not compare with methods like Sparse Autoencoders (SAEs) (Lim et al., 2024; Cunningham et al., 2024; Sharkey et al., 2022) that assume an overcomplete representation. Even though we restricted our experiments as such, future studies may consider this analysis, since our method is not technically limited in this regard. While we evaluated our approach across multiple CNN architectures, we did not investigate applications to Vision Transformers (ViTs) (Caron et al., 2021; He et al., 2022), despite existing literature suggesting that similar linear assumptions may apply in that context (Kaltampanidis et al., 2025)—this remains a promising direction for future work. Moreover, unlike some methods from unsupervised state-of-the-art that operate on class-specific image subsets constructed from model predictions, we applied our method to unlabeled data without class filtering, due to the lack of dense annotations in the training sets. Nevertheless, our framework could, in principle, be adapted for class-conditional analysis. Finally, our study aimed to minimize hyperparameter tuning by relying on prior work (Doumanoglou et al., 2023; 2024) and empirical preliminary

experiments; however, further tuning or exploration of additional hyperparameters may yield improvements in clustering quality, interpretability, or influence (Section A.14).

# 9 Conclusion

We introduced an innovative unsupervised technique to uncover pairs of latent space encoding-decoding directions that align with interpretable concepts of influence. This research offers a new perspective on the unsupervised identification of concept directions, unlike previous methods which are based on feature reconstruction or matrix decomposition. We believe that our work opens the door to richer model diagnostics, fine-grained explanations, and targeted interventions, paving the way for future research on scalable concept discovery, dynamic model editing, and integration into decision-making systems.

### Acknowledgments

This work has been supported by the EC funded Horizon Europe Framework Programme: CAVAA Grant Agreement 101071178.

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

# A   Appendix

## Appendix Index

### A.1 Unsupervised Interpretable Basis Extraction and Concept-Basis Extraction Losses

This Section complements Section 3.3 by providing further details on the loss terms of Unsupervised Interpretable Basis Extraction (UIBE) (Doumanoglou et al., 2023) and Concept Basis Extraction (CBE) (Doumanoglou et al., 2024).

**Sparsity Loss ($\mathcal{L}^s$)** (Doumanoglou et al., 2023) Based on the observation that the number of semantic labels that may be attributed to an image patch is only a fraction of the set of possible semantic labels, this loss enforces sparsity on the number of concepts that may be attributed to a patch embedding. In particular, the sparsity loss for pixel $\boldsymbol{p}$ is based on minimizing entropy and is defined as:

$$\mathcal{L}^s_{\boldsymbol{p}} = -\sum_i q_{\boldsymbol{p},i}\log_2 q_{\boldsymbol{p},i}, \quad q_{\boldsymbol{p},i} = \frac{y_{\boldsymbol{p},i}}{\sum_i y_{\boldsymbol{p},i}} \tag{19}$$

The aggregated sparsity loss $\mathcal{L}^s$ is:

$$\mathcal{L}^s = \mathbb{E}_{\boldsymbol{p}}\big[\mathcal{L}^s_{\boldsymbol{p}}\big] \tag{20}$$

**Maximum Activation Loss ($\mathcal{L}^{ma}$)** (Doumanoglou et al., 2023)

With the complement of this loss, the cluster membership variables $y_{\boldsymbol{p},i}$ are enforced to become binary:

$$\mathcal{L}^{ma} = \mathbb{E}_{\boldsymbol{p}}\Big[-\sum_i q_{\boldsymbol{p},i}\log_2 y_{\boldsymbol{p},i}\Big] \tag{21}$$

**Inactive Classifier Loss ($\mathcal{L}^{ic}$)** (Doumanoglou et al., 2024)

This loss ensures that each concept detector in the set classifies positively at least $\nu \in [0,1]$ percent of pixels in the concept dataset, avoiding clusters with few assignments.

$$\mathcal{L}^{ic} = \mathbb{E}_i\Big[\frac{1}{\nu}\text{ReLU}\big(\nu - \mathbb{E}_{\boldsymbol{p}}[y^{\gamma}_{\boldsymbol{p},i}]\big)\Big] \tag{22}$$

with $\nu = \frac{\tau}{I}$, $\gamma > 1, \gamma \in \mathbb{R}^+$ denoting a sharpening factor and $\tau \in [0,1]$ denoting a percent of pixels in the dataset to be evenly distributed among the $I$ classifiers in the set.

**Maximum Margin Loss ($\mathcal{L}^{mm}$)** In the original formulation of Doumanoglou et al. (2023), the Maximum Margin Loss was defined as $\mathcal{L}^{mm} = \frac{1}{M}$ with $M$ being a single parameter for the whole set of classifiers since the optimization was performed in the standardized space with shared parameters for the margins $M$ and biases $b$. In this work, we removed the standardized space constraints and instead, we have a margin parameter $M_i$ for each classifier in the set. Thus, we modify the Maximum Margin loss to become:

$$\mathcal{L}^{mm} = \frac{1}{I}\sum_i \frac{1}{M_i} \tag{23}$$

### A.2 Encoding-Decoding Direction Pairs Algorithm

Algorithm 1 provides, high-level, PyTorch-like pseudocode for learning the proposed Encoding-Decoding Direction Pairs.

---
**Algorithm 1** PyTorch-Like Encoding-Decoding Direction Pairs Pseudocode

---
**Input:** $\boldsymbol{X} \in \mathbb{R}^{B \times D \times H \times W}$: Layer feature activations with $B$ denoting batch size.
  $f^+$: Upper-part of the studied deep network.
**Learnable Parameters:**
  $\boldsymbol{W} \in \mathbb{R}^{D \times I}$: Decoding directions (filters of concept detectors)
  $\hat{\boldsymbol{S}} \in \mathbb{R}^{D \times I}$: Encoding directions
  $\boldsymbol{b} \in \mathbb{R}^{I}$: Concept detector biases
**Output:** Loss to minimize

1: **function** LEARNING_STEP($\boldsymbol{X}$)
2:   `z = torch.conv2d(`$\boldsymbol{X}$`,`$\boldsymbol{W}$`.T.unsqueeze(-1).unsqueeze(-1))`
3:   `p = z - `$\boldsymbol{b}$`.reshape(1,-1,1,1)`
4:   $\hat{\boldsymbol{S}}$` = signal_vectors(`$\boldsymbol{X}$`,z,p)`      ▷ Use running stats for variance and covariance and p>0 to subsample
5:   `y = torch.sigmoid(p)`
6:   `q = y / torch.sum(y,dim=1,keepdim=True)`
7:   `loss_fs = `$\mathcal{L}^{fs}$`(q)`
8:   `loss_ma = `$\mathcal{L}^{ma}$`(y,q)`
9:   `loss_mm = `$\mathcal{L}^{mm}$`(1/torch.norm(`$\boldsymbol{W}$`,dim=0))`      ▷ $M_i$ equals $1/||\boldsymbol{w}_i||_2$
10:   `loss_ic = `$\mathcal{L}^{ic}$`(y)`
11:   `loss_eac = `$\mathcal{L}^{eac}$`(y)`
12:   `loss_fso = `$\mathcal{L}^{fso}$`(`$\boldsymbol{W}$`,`$\hat{\boldsymbol{S}}$`)`
13:   **if** with UFM **then**
14:     $\boldsymbol{X}'$` = `$g(\boldsymbol{X}, \boldsymbol{W})$
15:   **else if** with CFM **then**
16:     $\boldsymbol{X}'$` = `$h(\boldsymbol{X}, \boldsymbol{W}, \hat{\boldsymbol{S}})$
17:   **end if**
18:   `loss_ur = `$\mathcal{L}^{ur}$`(`$\boldsymbol{X}'$`,`$f^+$`)`
19:   `loss_lag = Lagrangian(loss_fs,loss_ur,loss_ma,loss_ic,loss_mm,loss_eac,loss_fso)`
20:   **return** `loss_lag`
21: **end function**

---

## A.3 Theoretical Comparison with Concept-Basis Extraction

Concept-Basis Extraction (CBE) Doumanoglou et al. (2024) made an initial attempt to exploit the network's instructions to improve the interpretability of the discovered directions. This approach, like ours, used feature manipulation and maximized the entropy of the network's prediction outcomes for the manipulated features.

However, the key difference between the approaches lies in the manipulation strategy: the prior method suggested manipulating feature representations ($\boldsymbol{x_p}$) toward the concept detectors' hyperplanes **only for concepts currently present** in $\boldsymbol{x_p}$ (i.e., moving negatively along the filter weights when the concept was positively classified). It explicitly neglected manipulation in the opposite direction for negatively classified concepts. This essentially implied that network predictions should be uncertain only when none of the concept detectors made a **positive** prediction, ignoring confident **negative** ones.

Our approach fundamentally differs: we suggest that uncertain network predictions are tied to maximally ambiguous concept information, enforcing this by **manipulating all features toward the hyperplanes**, regardless of whether the concept was initially classified as present or absent. This ensures the network is maximally uncertain only when none of the detectors makes a confident prediction, whether positive or negative. Furthermore, CBE completely overlooked the critical role of **signal directions**.

## A.4 Details on Uncertainty Region Alignment

**Unconstrained Feature Manipulation (UFM):** In this type of manipulation $\boldsymbol{x_p}' = g_p(\boldsymbol{x_p}; \boldsymbol{W}) = \boldsymbol{x_p} - \boldsymbol{dx_p}$ and $\boldsymbol{dx_p}$ such that:

$$\boldsymbol{w}_i^T \boldsymbol{x_p}' - b_i = 0 \Rightarrow \boldsymbol{w}_i^T (\boldsymbol{x_p} - \boldsymbol{dx_p}) - b_i = 0, \ \forall i,$$
$$\boldsymbol{W}^T(\boldsymbol{x_p} - \boldsymbol{dx_p}) - \boldsymbol{b} = \boldsymbol{0} \Rightarrow \boldsymbol{dx_p} = (\boldsymbol{W}^T)^+(\boldsymbol{W}^T \boldsymbol{x_p} - \boldsymbol{b}) \tag{24}$$

with $\boldsymbol{A}^+$ denoting the pseudo-inverse of $\boldsymbol{A}$.

**Constrained Feature Manipulation (CFM):** In this type of manipulation $\boldsymbol{x}'_{\boldsymbol{p}} = h_{\boldsymbol{p}}(\boldsymbol{x}_{\boldsymbol{p}}; \boldsymbol{W}, \hat{\boldsymbol{S}}) = \boldsymbol{x}_{\boldsymbol{p}} - d\boldsymbol{x}_{\boldsymbol{p}}$, with $d\boldsymbol{x}_{\boldsymbol{p}}$ given by:

$$\boldsymbol{w}_i^T \boldsymbol{x}'_{\boldsymbol{p}} - b_i = 0 \Rightarrow \boldsymbol{w}_i^T(\boldsymbol{x}_{\boldsymbol{p}} - d\boldsymbol{x}_{\boldsymbol{p}}) - b_i = 0, \forall i,$$
$$\boldsymbol{W}^T(\boldsymbol{x}_{\boldsymbol{p}} - d\boldsymbol{x}_{\boldsymbol{p}}) - \boldsymbol{b} = \boldsymbol{0} \Rightarrow \boldsymbol{W}^T(\boldsymbol{x}_{\boldsymbol{p}} - \hat{\boldsymbol{S}}\boldsymbol{v}_{\boldsymbol{p}}) - \boldsymbol{b} = \boldsymbol{0} \Rightarrow$$
$$\boldsymbol{W}^T\hat{\boldsymbol{S}}\boldsymbol{v}_{\boldsymbol{p}} = \boldsymbol{W}^T\boldsymbol{x}_{\boldsymbol{p}} - \boldsymbol{b} \Rightarrow \boldsymbol{v}_{\boldsymbol{p}} = (\boldsymbol{W}^T\hat{\boldsymbol{S}})^+(\boldsymbol{W}^T\boldsymbol{x}_{\boldsymbol{p}} - \boldsymbol{b}) \Rightarrow \tag{25}$$
$$d\boldsymbol{x}_{\boldsymbol{p}} = \hat{\boldsymbol{S}}\boldsymbol{v}_{\boldsymbol{p}} = \hat{\boldsymbol{S}}(\boldsymbol{W}^T\hat{\boldsymbol{S}})^+(\boldsymbol{W}^T\boldsymbol{x}_{\boldsymbol{p}} - \boldsymbol{b}) \tag{26}$$

## A.5 Details on Reading Concept Information and Intervening on their Encoding

**Reading Concept Information from the Embeddings** Reading concept information from a patch embedding ends up in estimating the concept's signal value that is encoded in the representation. Supposing that the related conditions are met, in Section 3.2 it was shown that the filter direction can serve as a means to extract this value. While in the data-model of Section 3.2, the extraction of the signal's value is exact, under the multi-concept signal-distractor data model of Section 4.1, the considered latent space bias $\boldsymbol{c}$ interferes with this value. More specifically, if we suppose $\boldsymbol{w}_i : \boldsymbol{w}_i \perp \hat{\boldsymbol{s}}_j, i \neq j, \boldsymbol{w}_i^T\hat{\boldsymbol{s}}_i = 1$ and $\boldsymbol{w}_i \perp \boldsymbol{D}$, the inner product between the filter and the embedding results in:

$$\boldsymbol{w}_i^T \boldsymbol{x}_{\boldsymbol{p}} = \alpha_{\boldsymbol{p},i} \underbrace{\boldsymbol{w}_i^T\hat{\boldsymbol{s}}_i}_{=1} + \boldsymbol{w}_i^T\boldsymbol{c} \Rightarrow \boldsymbol{w}_i^T\boldsymbol{x}_{\boldsymbol{p}} = \alpha_{\boldsymbol{p},i} + \underbrace{\boldsymbol{w}_i^T\boldsymbol{c}}_{const} \tag{27}$$

Thus, when considering our extended data model, the filter can be used to extract the signal's value but with a constant offset, independent of $\boldsymbol{x}_{\boldsymbol{p}}$. Due to the offset being constant, the difference between two projected embeddings is equal to the difference of their signal values, e.g.:

$$\boldsymbol{w}_i^T(\boldsymbol{x}_{\boldsymbol{p}_1} - \boldsymbol{x}_{\boldsymbol{p}_2}) = \alpha_{\boldsymbol{p}_1,i} + \boldsymbol{w}_i^T\boldsymbol{c} - \alpha_{\boldsymbol{p}_2,i} - \boldsymbol{w}_i^T\boldsymbol{c} = \alpha_{\boldsymbol{p}_1,i} - \alpha_{\boldsymbol{p}_2,i} \tag{28}$$

In applications, we can exploit this property to estimate the signal value that is encoded in a patch embedding with respect to another patch embedding of reference. We found that two reference points are of particular interest. First, the average embedding over a collection of embeddings. In that case, we can estimate the signal value of $\boldsymbol{x}_{\boldsymbol{p}}$ with respect to the average signal value in the collection:

$$\boldsymbol{w}_i^T\boldsymbol{x}_{\boldsymbol{p}} - \boldsymbol{w}_i^T\mathbb{E}_{\boldsymbol{p}}[\boldsymbol{x}_{\boldsymbol{p}}] = \boldsymbol{w}_i^T\boldsymbol{x}_{\boldsymbol{p}} - \mathbb{E}_{\boldsymbol{p}}[\boldsymbol{w}_i^T\boldsymbol{x}_{\boldsymbol{p}}] = \alpha_{\boldsymbol{p},i} - \mathbb{E}_{\boldsymbol{p}}[\alpha_{\boldsymbol{p},i}] \tag{29}$$

And second, the concept's point of uncertainty. Given a query point $\boldsymbol{x}_{\boldsymbol{p}}$, the concept's point of uncertainty $\hat{\boldsymbol{u}}_{\boldsymbol{p}}$ is the point that lies on the concept detector's hyperplane (thus corresponding to a point of maximum concept ambiguity) and the line defined by the query point and the direction of the signal vector. More specifically, the point of uncertainty is equal to $\hat{\boldsymbol{u}}_{\boldsymbol{p}} = \boldsymbol{x}_{\boldsymbol{p}} - \kappa_{\boldsymbol{p},i}\hat{\boldsymbol{s}}_i$ with $\kappa_{\boldsymbol{p},i} \in \mathbb{R}$ corresponding to the solution of the following system of equations:

$$\left.\begin{array}{r}\hat{\boldsymbol{u}}_{\boldsymbol{p}} = \boldsymbol{x}_{\boldsymbol{p}} - \kappa_{\boldsymbol{p},i}\hat{\boldsymbol{s}}_i \\ \boldsymbol{w}_i^T\hat{\boldsymbol{u}}_{\boldsymbol{p}} - b_i = 0\end{array}\right\} \Rightarrow \boldsymbol{w}_i^T\hat{\boldsymbol{u}}_{\boldsymbol{p}} - b_i = 0 \Rightarrow \boldsymbol{w}_i^T(\boldsymbol{x}_{\boldsymbol{p}} - \kappa_{\boldsymbol{p},i}\hat{\boldsymbol{s}}_i) - b_i = 0 \Rightarrow \kappa_{\boldsymbol{p},i} = \frac{\boldsymbol{w}_i^T\boldsymbol{x}_{\boldsymbol{p}} - b_i}{\underbrace{\boldsymbol{w}_i^T\hat{\boldsymbol{s}}_i}_{=1}} = \boldsymbol{w}_i^T\boldsymbol{x}_{\boldsymbol{p}} - b_i$$
$$\tag{30}$$

Typically $\hat{\boldsymbol{u}}_{\boldsymbol{p}}$ depends on $i$, but we drop this index for brevity. What is interesting about the point of uncertainty is that the following difference corresponds to centering the signal value of $\boldsymbol{x}_{\boldsymbol{p}}$ around the value that represents uncertainty:

$$\boldsymbol{w}_i^T\boldsymbol{x}_{\boldsymbol{p}} - \boldsymbol{w}_i^T\hat{\boldsymbol{u}}_{\boldsymbol{p}} = \boldsymbol{w}_i^T\boldsymbol{x}_{\boldsymbol{p}} - \boldsymbol{w}_i^T(\boldsymbol{x}_{\boldsymbol{p}} - \kappa_{\boldsymbol{p},i}\hat{\boldsymbol{s}}_i) = \kappa_{\boldsymbol{p},i}\underbrace{\boldsymbol{w}_i^T\hat{\boldsymbol{s}}_i}_{=1} = \kappa_{\boldsymbol{p},i} = \boldsymbol{w}_i^T\boldsymbol{x}_{\boldsymbol{p}} - b_i \tag{31}$$

Since (31) corresponds to the function of the concept detector, this difference is positive whenever the concept is present in $\boldsymbol{x_p}$ and negative whenever the concept is absent. Thus, centering the signal value of a patch embedding around its point of uncertainty can be accomplished by computing $\kappa_{\boldsymbol{p},i}$ using (30).

**Intervening on Concept Encoding** Altering the concept content that is encoded in a patch embedding $\boldsymbol{x_p}$, ends up overwriting the signal value of the concept with a target value of interest. The target value is meant to be copied from a target embedding $\boldsymbol{x_p^t}$. Let $\boldsymbol{x_p'}$ denote the altered embedding after the intervention. Supposing that we aim to intervene on the value of concept $i$, $\boldsymbol{x_p'}$ becomes:

$$\boldsymbol{x_p'} = \boldsymbol{x_p} + \kappa_{\boldsymbol{p},i}\hat{\boldsymbol{s}}_i \tag{32}$$

with $\kappa_{\boldsymbol{p},i} \in \mathbb{R}$. Since we want $\boldsymbol{x_p'}$ and $\boldsymbol{x_p^t}$ to have the same signal value for concept $i$, we make use of (28) and compute $\kappa_{\boldsymbol{p},i}$ such that:

$$\boldsymbol{w}_i^T(\boldsymbol{x_p'} - \boldsymbol{x_p^t}) = 0 \Rightarrow \boldsymbol{w}_i^T(\boldsymbol{x_p} + \kappa_{\boldsymbol{p},i}\hat{\boldsymbol{s}}_i - \boldsymbol{x_p^t}) = 0 \Rightarrow \kappa_{\boldsymbol{p},i} = \frac{\boldsymbol{w}_i^T\boldsymbol{x_p^t} - \boldsymbol{w}_i^T\boldsymbol{x_p}}{\underbrace{\boldsymbol{w}_i^T\hat{\boldsymbol{s}}_i}_{=1}} = \underbrace{\boldsymbol{w}_i^T\boldsymbol{x_p^t}}_{t} - \boldsymbol{w}_i^T\boldsymbol{x_p} = t - \boldsymbol{w}_i^T\boldsymbol{x_p} \tag{33}$$

with $t$ denoting the projected target value. In applications, we are typically interested in altering the concept content of an embedding towards the presence or absence of a concept. To accomplish that, two natural choices arise: first, to overwrite the signal value of the embedding based on the average signal value of samples with or without the concept, and second, to overwrite the signal value of the embedding to match the value of a top (or bottom) quantile of signal values in a collection. In the first case, we can use $\boldsymbol{x_p^t} = \mathbb{E}_p[\boldsymbol{x_p}]$, with the average taking place along the patches with (or without) the concept. For the second case, we can directly work in the projection space, by computing the quantile of interest over the projected embeddings $\boldsymbol{w}_i^T\boldsymbol{x_p}$ and use that value in place of $t$.

## A.6 Details on Using Encoding-Decoding Direction Pairs and the Regions of Uncertainty to Provide Concept-Based Local Explanations and Detailed Spatially-Aware Concept Contribution Maps

In this Section, we discuss how our learned Encoding-Decoding Direction Pairs can be leveraged in order to provide concept-based local explanations for any network prediction. For simplicity, we will be discussing the application in Convolutional Neural Networks (CNNs) with a Global Average Pooling (GAP) layer and a ReLU activation function, but adaptation to other architectures could be possible.

### A.6.1 Terminology and Definitions

Suppose that we want to explain the prediction of an image classifier $f$ for image $\mathcal{I}$. Let $\boldsymbol{X}$ denote this image's representation at the penultimate layer of the network (in CNNs, this is typically the last convolutional layer before GAP). Let $l_c \in \mathbb{R}$ denote the network's output logit for class $c$ regarding input $\mathcal{I}$. Typically, $c$ will be chosen such that $l_c$ is the maximum logit among the output class logits, but other choices are also valid, for instance when we want to explain why the prediction deviated from a specific class. Let also $\boldsymbol{X}_b^m \in \mathbb{R}^{H \times W \times D}$ denote a **baseline** point in the **uncertainty region of the model**. We call it baseline because this point corresponds to an artificial image representation for which the prediction of the network is highly uncertain. Let $l_b^m \in \mathbb{R}$ denote the prediction logit for class $c$ that corresponds to $\boldsymbol{X}_b^m$, i.e. $l_b^m = f^+(\boldsymbol{X}_b^m)$. We define the following quantity:

$$l_e = l_c - l_b^m \tag{34}$$

an explanation logit ($l_e$), which is equal to the difference in predicting class $c$ for image $\mathcal{I}$ compared to a highly uncertain prediction for the same class. Let $\boldsymbol{w}_c$ and $b_c$ denote the class vector and the corresponding bias of the network's **last** fully connected layer. Then, the explanation logit becomes:

$$l_e = l_c - l_b^m = \boldsymbol{w}_c^T\text{GAP}(\boldsymbol{X}) + b_c - (\boldsymbol{w}_c^T\text{GAP}(\boldsymbol{X}_b^m) + b_c) \tag{35}$$

Our goal is to express this quantity in terms of the identified concepts. Based on our previous discussions in Sections 4.4 and 5.1, we conclude that an intuitive point of reference for expressing concept content is a

**baseline point in the uncertainty region of the concept detectors**. Any image representation $\boldsymbol{X}$ has a corresponding **baseline** point $\boldsymbol{X}_b^c \in \mathbb{R}^{H \times W \times D}$ in that region, which can be obtained by $h(\boldsymbol{X}; \boldsymbol{W}, \hat{\boldsymbol{S}})$. If it weren't for the layer's activation function, we could choose $\boldsymbol{X}_b^m$ to be equal to $\boldsymbol{X}_b^c$, due to the alignment of the two uncertainty regions during direction learning. However, in practice, there is a shift between the two points, since $\boldsymbol{X}_b^m = \mathrm{ReLU}(\boldsymbol{X}_b^c)$. For this reason, we treat $\boldsymbol{X}_b^m$ similar to $\boldsymbol{X}$, i.e. as an artificial image representation whose concept content can be expressed with respect to a baseline point $\boldsymbol{X}_b^{mc}$ in the **uncertainty region of the concept detectors**. Similar to $\boldsymbol{X}_b^c$, $\boldsymbol{X}_b^{mc}$ can be calculated by $h(\boldsymbol{X}_b^m; \boldsymbol{W}, \hat{\boldsymbol{S}})$. Finally, let $\boldsymbol{v_p}$ and $\boldsymbol{v_p^b}$ be equal to the expression of (10) when computing baseline points in the concept detectors' uncertainty region for $\boldsymbol{X}$ and $\boldsymbol{X}_b^m$, respectively.

### A.6.2 Logit Difference Decomposition in terms of Concepts

Based on the aforementioned definitions, we have:

$$l_e = l_c - l_b^m = \boldsymbol{w}_c^T \mathrm{GAP}(\boldsymbol{X}) + b_c - (\boldsymbol{w}_c^T \mathrm{GAP}(\boldsymbol{X}_b^m) + b_c) \Rightarrow$$
$$l_e = \boldsymbol{w}_c^T \mathrm{GAP}(\boldsymbol{X} - \boldsymbol{X}_b^c) - \boldsymbol{w}_c^T \mathrm{GAP}(\boldsymbol{X}_b^m - \boldsymbol{X}_b^{mc}) + \boldsymbol{w}_c^T \mathrm{GAP}(\boldsymbol{X}_b^c - \boldsymbol{X}_b^{mc}) \tag{36}$$

Due to the linearity of GAP, (36) can be written as:

$$l_e = \mathbb{E}_{\boldsymbol{p}}\left[\boldsymbol{w}_c^T(\boldsymbol{x_p} - \boldsymbol{x}_{\boldsymbol{p},b}^c)\right] - \mathbb{E}_{\boldsymbol{p}}\left[\boldsymbol{w}_c^T(\boldsymbol{x}_{\boldsymbol{p},b}^m - \boldsymbol{x}_{\boldsymbol{p},b}^{mc})\right] + \mathbb{E}_{\boldsymbol{p}}\left[\boldsymbol{w}_c^T(\boldsymbol{x}_{\boldsymbol{p},b}^c - \boldsymbol{x}_{\boldsymbol{p},b}^{mc})\right] \tag{37}$$

where $\boldsymbol{x}_{\boldsymbol{p},b}^c$, $\boldsymbol{x}_{\boldsymbol{p},b}^m$ and $\boldsymbol{x}_{\boldsymbol{p},b}^{mc}$ elements at the spatial location $\boldsymbol{p}$ of $\boldsymbol{X}_b^c, \boldsymbol{X}_b^m$ and $\boldsymbol{X}_b^{mc}$, respectively. By definition, we have $\boldsymbol{x_p} - \boldsymbol{x}_{\boldsymbol{p},b}^c = \hat{\boldsymbol{S}}\boldsymbol{v_p}$, and $\boldsymbol{x}_{\boldsymbol{p},b}^m - \boldsymbol{x}_{\boldsymbol{p},b}^{mc} = \hat{\boldsymbol{S}}\boldsymbol{v_p^b}$, $\boldsymbol{v_p}, \boldsymbol{v_p^b} \in \mathbb{R}^I$. Given these, (37) can be written as:

$$l_e = \mathbb{E}_{\boldsymbol{p}}\left[\boldsymbol{w}_c^T \hat{\boldsymbol{S}}(\boldsymbol{v_p} - \boldsymbol{v_p^b})\right] + \mathbb{E}_{\boldsymbol{p}}\left[\boldsymbol{w}_c^T \boldsymbol{r_p}\right] \tag{38}$$

with $\boldsymbol{r_p} = \boldsymbol{x}_{\boldsymbol{p},b}^c - \boldsymbol{x}_{\boldsymbol{p},b}^{mc} \in \mathbb{R}^D$ a residual that is not going to be explained in terms of concepts. The first part of (38) can be interpreted as follows: the contribution of concept $i$ to the explanation logit is equal to $\boldsymbol{w}_c^T \hat{\boldsymbol{s}}_i (v_{\boldsymbol{p},i} - v_{\boldsymbol{p},i}^b)$. This means that the explanation logit depends on a global concept contribution factor $\boldsymbol{w}_c^T \hat{\boldsymbol{s}}_i$ which is constant regardless of the sample to be explained, and a local contribution factor which depends on the sample and is equal to the difference in concept content between the sample itself and a synthetic baseline sample that corresponds to a highly uncertain network prediction. Since both $\boldsymbol{x}_{\boldsymbol{p},b}^c$ and $\boldsymbol{x}_{\boldsymbol{p},b}^{mc}$ lie in the uncertainty region of the concept detectors, their difference in terms of concept content is zero, and the residual does not carry any concept-related information.

Let $\hat{v}_{\boldsymbol{p},i}$ ($\hat{v}_{\boldsymbol{p},i}^b$) denote the signal value of $\boldsymbol{x_p}$ ($\boldsymbol{x}_{\boldsymbol{p},b}^m$) for concept $i$ centered around the value corresponding to the concept ambiguity (i.e. by considering $\boldsymbol{x_p}$ ($\boldsymbol{x}_{\boldsymbol{p},b}^m$) as a query point and using the concept's point of uncertainty as a reference). As discussed in Section 5.1, positive $\hat{v}_{\boldsymbol{p},i}$ ($\hat{v}_{\boldsymbol{p},i}^b$) indicates concept presence, while negative $\hat{v}_{\boldsymbol{p},i}$ ($\hat{v}_{\boldsymbol{p},i}^b$) indicates concept absence. $\hat{v}_{\boldsymbol{p},i}$ ($\hat{v}_{\boldsymbol{p},i}^b$) and $v_{\boldsymbol{p},i}$ ($v_{\boldsymbol{p},i}^b$) would be exactly the same if $\boldsymbol{w}_i^T \hat{\boldsymbol{s}}_j = 0 \,\forall\, i \neq j$ and also if each $\boldsymbol{w}_i$ is perpendicular to the basis of distractors $\boldsymbol{D}$. In practice, these orthogonality constraints are only approximately fulfilled. To obtain intuitive explanations without compromising fidelity, we re-write (38) as:

$$l_e = \mathbb{E}_{\boldsymbol{p}}\left[\boldsymbol{w}_c^T \hat{\boldsymbol{S}}(\hat{\boldsymbol{v}}_{\boldsymbol{p}} - \hat{\boldsymbol{v}}_{\boldsymbol{p}}^b) + \boldsymbol{w}_c^T \hat{\boldsymbol{S}}(\boldsymbol{v_p} - \hat{\boldsymbol{v}}_{\boldsymbol{p}} - (\boldsymbol{v_p^b} - \hat{\boldsymbol{v}}_{\boldsymbol{p}}^b))\right] + \mathbb{E}_{\boldsymbol{p}}\left[\boldsymbol{w}_c^T \boldsymbol{r_p}\right] \Rightarrow \tag{39}$$

$$l_e = \underbrace{\mathbb{E}_{\boldsymbol{p}}\left[\boldsymbol{w}_c^T \hat{\boldsymbol{S}}\hat{\boldsymbol{v}}_{\boldsymbol{p}}\right]}_{\text{sample concept}} - \underbrace{\mathbb{E}_{\boldsymbol{p}}\left[\boldsymbol{w}_c^T \hat{\boldsymbol{S}}\hat{\boldsymbol{v}}_{\boldsymbol{p}}^b\right]}_{\text{baseline concept}} + \underbrace{\mathbb{E}_{\boldsymbol{p}}\left[\boldsymbol{w}_c^T \hat{\boldsymbol{S}}(\boldsymbol{v_p} - \hat{\boldsymbol{v}}_{\boldsymbol{p}} - (\boldsymbol{v_p^b} - \hat{\boldsymbol{v}}_{\boldsymbol{p}}^b))\right]}_{\text{correction}} + \underbrace{\mathbb{E}_{\boldsymbol{p}}\left[\boldsymbol{w}_c^T \boldsymbol{r_p}\right]}_{residual} \tag{40}$$

Based on (40), the explanation logit $l_e$ is linearly expressed in terms of patches and concepts, giving detailed, spatially-aware information for concept contributions. We split (40) into four parts. The first part corresponds to contributions of concepts included in the sample (*sample concept*), the second to the contribution of concepts contained in an artificial baseline point of an uncertain prediction (*baseline concept*), the third part to a *correction* factor to account for imperfect direction learning convergence, and finally, a *residual*

Table 15: Left: Data matrices $\boldsymbol{S}$ and $\boldsymbol{D}$ for the experiment on synthetic data. Right: Cosine similarities for every pair of vectors in $\boldsymbol{S}$, $\boldsymbol{D}$, i.e.: $\boldsymbol{C}^T\boldsymbol{C}, \boldsymbol{C} = [\boldsymbol{S}|\boldsymbol{D}]$.

| S | | | D | |
|---|---|---|---|---|
| 0.6368 | 0.8583 | 0.5259 | 0.4008 | 0.6659 |
| 0.1561 | -0.3371 | 0.1561 | 0.4585 | 0.6038 |
| 0.1633 | -0.1533 | 0.7557 | 0.3337 | -0.2065 |
| 0.2617 | 0.1643 | -0.1580 | 0.5797 | -0.2154 |
| 0.6226 | -0.1607 | -0.1643 | -0.1596 | 0.1687 |
| -0.1759 | 0.1607 | -0.1594 | 0.2744 | 0.1617 |
| -0.1592 | 0.1531 | -0.1567 | 0.2232 | 0.1567 |
| -0.1760 | 0.1554 | -0.1612 | 0.1763 | 0.1546 |

| Cosine-Similarities | | | | |
|---|---|---|---|---|
| 1.0 | 0.3319 | 0.4204 | 0.3188 | 0.4526 |
| 0.3319 | 1.0 | 0.2087 | 0.3649 | 0.4111 |
| 0.4204 | 0.2087 | 1.0 | 0.3621 | 0.2195 |
| 0.3188 | 0.3649 | 0.3621 | 1.0 | 0.4296 |
| 0.4526 | 0.4111 | 0.2195 | 0.4296 | 1.0 |

that we do not explain. If the correction factor for concept $i$ was zero, concept $i$ would have a positive contribution to the explanation logit whenever the difference in contributions between the sample and the baseline is positive. In the case of imperfect learning conditions, this difference must exceed the negative of the correction factor.

In the experiments, we report Concept Contribution Maps (CCMs) where each spatial element $\boldsymbol{p}$ is equal to:

$$\phi_{\boldsymbol{p}}^i = \boldsymbol{w}_c^T \hat{\boldsymbol{s}}_i (\hat{v}_{\boldsymbol{p},i} - \hat{v}_{\boldsymbol{p},i}^b) \tag{41}$$

visualized as a heatmap over the original image for a given concept $i$. Finally, we refer to the quantity $\boldsymbol{w}_c^T \hat{\boldsymbol{s}}_i$ as the Concept-Class Relation Coefficient (CCRC).

## A.7 Direction Learning Process

We learn the directions of the proposed method in a four-step process: a) we first learn the parameters $\boldsymbol{w}_i, b_i$ following Doumanoglou et al. (2024), replacing the *CNN Classifier Loss* with the proposed $\mathcal{L}^{uur}$; b) we then continue optimizing $\boldsymbol{w}_i, b_i$, removing the orthogonality and standardization constraints while keeping $\mathcal{L}^{uur}$ and incorporating the additional losses from Section 4.2; c) next, we learn the signal vectors using the filters of the learned classifiers as signal value extractors in (1) to initialize $\{\hat{\boldsymbol{S}}\}$; and d) finally, we jointly optimize $\hat{\boldsymbol{s}}_i$, $\boldsymbol{w}_i, b_i$, using all previous losses, replacing $\mathcal{L}^{uur}$ with $\mathcal{L}^{cur}$ and adding $\mathcal{L}^{fso}$ from Sections 4.3 and 4.4. In all cases, we prefer optimizing with the Augmented Lagrangian Loss instead of linearly combining the loss terms.

## A.8 Details for the Experiment on Synthetic Data

We train the network using cross-entropy loss and the Adam (Kingma, 2014) optimizer, with learning rate 0.0005 and batch size 1024 for 4000 epochs. In principle, we follow the process defined in Section A.7, but due to the simplicity of the example, during (a) we don't use $\mathcal{L}^{uur}$, we omit step (b) and proceed directly from (a) to (c). In both steps (a) and (d) we use the Augmented Lagrangian Loss, the Adam optimizer, and the Cosine Annealing learning rate scheduler (Loshchilov & Hutter, 2016). In step (a), we omit the constraints of the Augmented Lagrangian loss that are not applicable. The Augmented Lagrangian formulation greatly stabilizes learning and avoids local optima. For step (a) we solve the constrained optimization problem of minimizing $\mathcal{L}^s$ with $\tau^{ma} = 0.8$, $\tau^{mm} = 5$ and $\tau^{ic} = 0.0$. For step (d) we minimize $\lambda^{fs}\mathcal{L}^{fs} + \lambda^{cur}\mathcal{L}^{cur}$ with $\tau^{ma} = 0.5$, $\tau^{mm} = 15$, $\tau^{ic} = 0.0$, $\tau^{eac} = 0.0$, $\tau^{fso} = 0.01$. The learning rate we use for step (a) is 0.00025 and for step (b) is 0.0005. The number of learning epochs is set to 10000 and 20000, for each step respectively. For the loss weights we use $\lambda^{fs} = 2.6$ and $\lambda^{ur} = 0.25$. The sharpening factor $\gamma$ of $\mathcal{L}^{ic}$ is set to $\gamma = 2.0$, the $\tau$ hyperparameter of the same loss is set to $\tau = 1.0$, the $\nu$ sharpening factor of $\mathcal{R}_{SW}$ to $\nu = 2.0$, the $\rho$ of $\mathcal{L}^{eac}$ is set to $\rho = 5/3$ and the $\mu$ sharpening factor of $\mathcal{L}^{fs}$ is set to 2.0.

The specific values of matrices $\boldsymbol{S}$ and $\boldsymbol{D}$ used in this experiment, and the cosine similarities between every pair of vectors, are provided in Table 15.

## A.9 Extracting Signal Values with the Filters of Concept Detectors

As discussed in Section 4.3, signal values, which are required to estimate the encoding direction of a concept, are extracted using the filter weights of the concept detectors. Yet, as we discussed in that Section, in order for this to happen, the filter weights $\boldsymbol{w}_i$ need to be orthogonal to $\boldsymbol{s}_j$ and $\boldsymbol{D}$. Since we do not explicitly estimate distractors in this work, there may be an inevitable error when extracting the value of the signal (we say maybe, because this might also be mitigated by the Uncertainty Region Alignment losses or filters converge to become perpendicular to distractors due to the fact that they contain information independent of concept content). Here we study on the order of this error. From 3 we have:

$$\boldsymbol{x_p} = \boldsymbol{S}\boldsymbol{\alpha_p} + \boldsymbol{D}\boldsymbol{\beta_p} + \boldsymbol{c}$$

When extracting the signal value with a filter, the estimation becomes:

$$
\begin{aligned}
z_{\boldsymbol{p}} &= \boldsymbol{w}_i^T \boldsymbol{x_p} = \boldsymbol{w}_i^T \boldsymbol{S}\boldsymbol{\alpha_p} + \boldsymbol{w}_i^T \boldsymbol{D}\boldsymbol{\beta_p} + \boldsymbol{w}_i^T \boldsymbol{c} \\
\frac{z_{\boldsymbol{p}}}{\boldsymbol{w}_i^T \boldsymbol{s}_i} &= a_{\boldsymbol{p},i} + \frac{\boldsymbol{w}_i^T \boldsymbol{D}\boldsymbol{\beta_p}}{\boldsymbol{w}_i^T \boldsymbol{s}_i} + \frac{\boldsymbol{w}_i^T \boldsymbol{c}}{\boldsymbol{w}_i \boldsymbol{s}_i}
\end{aligned}
\tag{42}
$$

From (42) and as we've already seen in (27), we notice that the latent space bias $\boldsymbol{c}$ introduces an additional constant error term when estimating the signal values. In the real-world scenario when this $\boldsymbol{c}$ is not known, we can use the following estimator $\hat{a}_{\boldsymbol{p},i}$ which depends on the average of features $\boldsymbol{x_p}$:

$$
\begin{aligned}
\hat{a}_{\boldsymbol{p},i} &= \frac{\boldsymbol{w}_i^T \boldsymbol{x_p}}{\boldsymbol{w}_i^T \boldsymbol{s}_i} - \frac{\boldsymbol{w}_i^T \mathbb{E}_{\boldsymbol{p}}[\boldsymbol{x_p}]}{\boldsymbol{w}_i^T \boldsymbol{s}_i} = \\
\hat{a}_{\boldsymbol{p},i} &= \frac{\boldsymbol{w}_i^T \boldsymbol{x_p}}{\boldsymbol{w}_i^T \boldsymbol{s}_i} - \frac{\boldsymbol{w}_i^T \boldsymbol{s}_i \mathbb{E}_{\boldsymbol{p}}[a_{\boldsymbol{p},i}]}{\boldsymbol{w}_i^T \boldsymbol{s}_i} - \frac{\boldsymbol{w}_i^T \boldsymbol{D} \mathbb{E}_{\boldsymbol{p}}[\boldsymbol{\beta_p}]}{\boldsymbol{w}_i^T \boldsymbol{s}_i} \\
\hat{a}_{\boldsymbol{p},i} &= a_{\boldsymbol{p},i} - \mathbb{E}_{\boldsymbol{p}}[a_{\boldsymbol{p},i}] + \frac{\boldsymbol{w}_i^T \boldsymbol{D}}{\boldsymbol{w}_i^T \boldsymbol{s}_i}(\boldsymbol{\beta_p} - \mathbb{E}_{\boldsymbol{p}}[\boldsymbol{\beta_p}])
\end{aligned}
\tag{43}
$$

The latter is an estimator of $a_{\boldsymbol{p},i}$ with respect to the mean $\mathbb{E}_{\boldsymbol{p}}[a_{\boldsymbol{p},i}]$ and with an error term depending on distractors, irrespective of constant bias.

## A.10 Approaching Human Evaluations

In typical scenarios, human-in-the-loop experiments validate the interpretability of the identified concepts. In this work, however, we have annotations from the Broden and Brodem-Action datasets, which we can leverage as a proxy for human judgment.

**Implicit Human Evaluation via Annotated Data**: Our learned concept detectors are evaluated by calculating dataset-wide classification and segmentation metrics, such as Precision, Recall, AP, mIoU $\mathcal{M}, \mathcal{S}^1 \mathcal{S}^2 \mathcal{S}^3$ on the Broden and Broden-Action datasets. Because the ground-truth labels of those datasets were originally produced and verified by **human annotators**, **alignment** with these annotations serves as an implicit, large-scale human evaluation.

**CLIP as a Human Proxy**: we also employ an automated monosemanticity metric ($\mathcal{M}$) leveraging CLIP's (Radford et al., 2021) embedding space. As CLIP is trained to align visual features with human language, it acts as a reliable, high-level proxy for **semantic coherence**, measuring cluster concept tightness based on samples in the collection.

Conclusively, even though we did not employ physical subjects in our evaluation to assess the outcomes of the work, we relied on annotations from two large concept datasets to address this need. To further expand on annotation classes beyond those in the datasets, we used the CLIP foundation model as a proxy to measure coherency within the image collection of each concept.

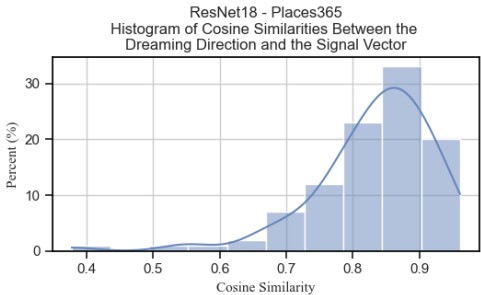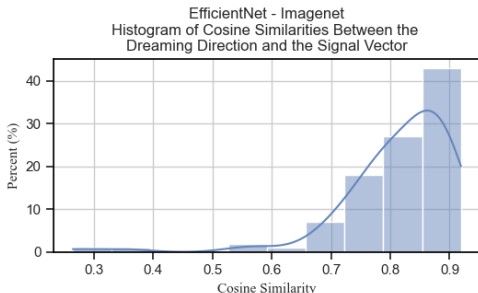

Figure 23: Cosine similarity histogram between *dreaming directions* and *signal vectors*. **Left:** Directions learned for ResNet18 trained on Places365 ($I = 448$). **Right**: Directions learned for EfficientNet trained on ImageNet ($I = 1120$). These histograms regard directions learned **with** $\mathcal{L}^{cur}$ but **without** $\mathcal{L}^{fso}$.

### A.11 Details on the Faithfulness Assessment of Encoding-Decoding Direction Pairs

Due to the considerable computational load required to conduct this experiment, we make some simplifications. First, we consider a subset of the learned direction pairs by randomly choosing 100. Candidate pairs are the ones with a concept detector that is considered interpretable by Network Dissection (i.e. the detector exhibits an IoU performance greater than 0.04 when detecting the concept). Second, to construct the image set that contains the concept, we consider 10 unique images within the set of the most confident (patch-level) predictions of the concept detector. For each image, we run the deep dream optimization for $K = 40$ iterations, and we record the patch embeddings (feature activations) containing the concept with an interval of 5 steps. For dreaming, we don't use zooming, but we do use the robustness transforms originally proposed in Olah et al. (2017). Regarding the line fitting process, we consider a parametric line as $c + \lambda d$, $\lambda \in \mathbb{R}$, with $c \in \mathbb{R}^D$ denoting the line's origin and $d \in \mathbb{R}^D$ its direction (the *dreaming* direction). At the end of the dreaming optimization loop and after recording the feature activations of the concept, we minimize the average distance from each feature to the line. Finally, we make sure that the dreaming direction points towards the direction of feature evolvement across iterations by considering making a sign flip, i.e. multiplying $d$ with $-1$.

### A.12 Additional Experiments on the Faithfulness Assessment of the Encoding-Decoding Direction Pairs

Figures 23 and 24 complement the experiments presented in Figures 11 and 12 for the remaining combinations regarding $\mathcal{L}^{fso}$. Even though less prominent in ResNet18, together with Section 7.2 the experiments sufficiently support that $\mathcal{L}^{fso}$ pushes the distribution of cosine similarities between dreaming directions and signal vectors to become more bell-shaped. Overall, the experiments in this section, similar to the experiments presented in Section 7.2, indicate that in approximately 90% of the cases, these cosine similarities exceed 0.7. In a less rigorous but more qualitative sense, we found that when learning EDDP with $\mathcal{L}^{fso}$, this loss converges to values close to $\approx 0.02$, whereas measuring the loss value for a direction set learned without $\mathcal{L}^{fso}$ often leads to a value close to $\approx 0.1$, which is still not that far from indicating orthogonality between filters and signal vectors.

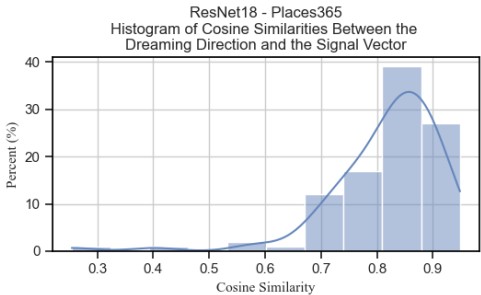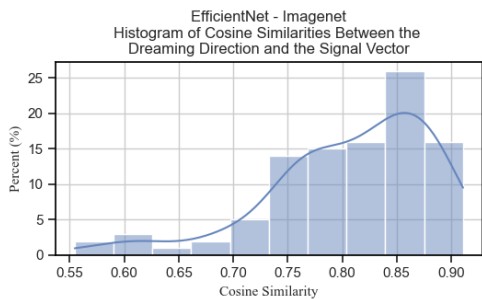

Figure 24: Cosine similarity histogram between *dreaming directions* and *signal vectors*. **Left:** Directions learned for ResNet18 trained on Places365 ($I = 448$). **Right**: Directions learned for EfficientNet trained on ImageNet ($I = 1120$). These histograms regard directions learned with $\mathcal{L}^{uur}$ **and** $\mathcal{L}^{fso}$.

### A.13    Details for the Experiments on Deep Image Classifiers

**Details on Forming Concept Detectors** While our approach explicitly learns binary classifiers $\{\boldsymbol{w}_i, b_i\}$ as concept detectors, an estimation of an explicit classification threshold $b$ is missing when clustering with PCA, NMF, and the Natural basis. To form concept detectors in those cases, we follow Bau et al. (2017), and for each direction, we choose $b$ to be equal to the top $k$-quantile among the projected activations sourced from the concept dataset. For the Natural latent space basis and NMF, we use $k = 0.005$, as suggested in Bau et al. (2017), while for PCA, we use $k = 0.2$ as suggested in Graziani et al. (2023a).

**Differences in the Evaluation Protocol compared to PCA-based and NMF-based State-of-the-Art and Limitations** Previous methods (Graziani et al., 2023a;b; Zhang et al., 2021; Fel et al., 2023b) based on PCA or NMF operated under a **class-conditioned** protocol, where the analysis and identified concepts are intrinsically linked to a specific class. On the contrary, in this work, our method is applied in a **class-agnostic** protocol, analyzing the latent space without prior class information. This fundamental difference in methodology renders a direct comparison with the exact implementations of these previous approaches (Graziani et al., 2023a;b; Zhang et al., 2021; Fel et al., 2023b) unfair or infeasible. Specifically, as the size of the concept space $I$ approaches the size of the embedding space $D$, NMF trivially converges to the natural feature space basis. Moreover, in a class agnostic setting, the gradient re-ranking approach of (Graziani et al., 2023a;b) is not feasible, as gradients are tied to a specific class, requiring further adaptation of the original algorithm. However, PCA and NMF are still applicable in a class-agnostic protocol as typical matrix decomposition methods.

In this work, we implement both PCA and NMF baselines as matrix decomposition methods on layer feature activations in a class-agnostic fashion. We emphasize that, for NMF, we do not consider the recursivity part of Fel et al. (2023b) (as we consider concept factorization to be outside the scope of this work), and for PCA, we do not perform the gradient re-ranking that was suggested in Graziani et al. (2023a).

Given the previous limitations, both baselines could be less effective in the class-agnostic setting compared to the class-conditioned state-of-the-art. The challenges they face under the generalized, class-agnostic protocol highlight the necessity for more effective and novel approaches to identifying concept directions in this setting, and the proposed method is such an attempt.

**Details on EDDP-U and EDDP-C** We emphasize that EDDP-U does not include the use of $\mathcal{L}^{fso}$ to learn the directions. The main purpose of $\mathcal{L}^{fso}$ is to contribute to a faithful estimation of the encoding directions. While this was critical for the experiment on synthetic data, in real-world experiments, we see that the value of this loss is sufficiently low even when not directly minimizing it, possibly due to working in a high-dimensional embedding space where two random vectors are approximately orthogonal. A detailed discussion regarding $\mathcal{L}^{fso}$ is provided in Sections A.12 and A.15 with the conclusions presented in the main body of the article being mostly aligned with the conclusions that stem from detailed ablation studies.

Table 16: Hyper-parameters used to learn the Direction Pairs

| ResNet18 / Places365 | | | | | ResNet50 / MiT | | | |
|---|---|---|---|---|---|---|---|---|
| $I$ | Step | Epochs | LR | | $I$ | Step | Epochs | LR |
| 384/448/512 | a | 500 | 0.005 | | 1536/1792/2048 | a | 1300 | 0.001 |
| 384/448/512 | b | 2500 | 0.00025 | | 1536/1792/2048 | b | 2600 | 0.00025 |
| 384/448/512 | d | 1250 | 0.00005 | | 1536/1792/2048 | d | 1300 | 0.0001 |
| **EfficientNet / ImageNet** | | | | | **VGG16 / ImageNet** | | | |
| $I$ | Step | Epochs | LR | | $I$ | Step | Epochs | LR |
| 960/1120/1280 | a | 800 | 0.001 | | 384/448/512 | a | 1000 | 0.001 |
| 960/1120/1280 | b | 3500 | 0.00005 | | 384/448/512 | b | 1000 | 0.0001 |
| 960/1120/1280 | d | 1750 | 0.00005 | | 384/448/512 | d | 500 | 0.0001 |

| Inception-v3 / ImageNet | | | |
|---|---|---|---|
| $I$ | Step | Epochs | LR |
| 1536/2048 | a | 1500 | 0.001 |
| 1536/1792/2048 | b | 3000 | 0.00005 |
| 1536/1792/2048 | d | 1500 | 0.00005 |
| 1792 | a | 1500 | 0.01 |

**Details on the Influence Evaluation Metric** As we already mentioned in Section 3.5, RCAV originally used CAVs to assess the network's concept sensitivity. Here we follow the proposition of Pahde et al. (2024) and use the learned encoding directions in their place. For sensitivity scores, we spatially replicate the learned encoding directions across all spatial locations in the image representation. In studies without $\mathcal{L}^{cur}$, we report influence metrics for signal vectors that were estimated **post learning the directions** with the subsample strategy discussed in Section 4.3.

**Learning Details** In Table 16 we provide the main hyper-parameters to train the direction pairs for each step. The learning rate that we provide (LR) is considered as the reference learning rate for a batch size of 4096. In practice, we scale both the learning rate and the batch size based on the available GPU memory and the number of GPUs. We use the Adam Kingma (2014) optimizer and the cosine annealing learning rate scheduler (Loshchilov & Hutter, 2016). While for step (a) we follow the hyper-parameter setup of Doumanoglou et al. (2024), for steps (b) and (d) we make different choices. For $\mathcal{L}^{ic}$ and $\mathcal{L}^{eac}$ we set their hyper-parameters $\tau$ and $\rho$ as follows: For ResNet18 and VGG16, $\tau$ and $\rho$ are set such that the minimum cluster size is equal to 400 and the maximum 50000. For EfficientNet and Inception-v3, the same numbers are 520 and 65000, while for ResNet50 the minimum cluster size is set to 920 and the maximum to 75000. These are all set based on the statistics of the concept datasets. The $\mu$ sharpening factor of $\mathcal{L}^{fs}$ and the $\nu$ sharpening factor of $\mathcal{R}_{SW}$ are set to 2.0.

For Augmented Lagrangian Loss, in all cases we used the following hyper-parameters: $\lambda^{fs} = 2.6$, $\lambda^{ur} = 0.25$, $\tau^{ma} = 0.8$, $\tau^{ic} = 0$,$\tau^{eac} = 0$,$\tau^{fso} = 0.01$. For ResNet18, EfficientNet and VGG16 we use $\tau^{mm} = 5.0$, while for Inception-v3 and ResNet50 we used $\tau^{mm} = 6.0$.

When using $\mathcal{L}^{uur}$ and $\mathcal{L}^{cur}$, we observed better results when manipulating features with a stochastic magnitude in the direction $\boldsymbol{dx_p}$, i.e. shifting representations as $\boldsymbol{x'_p} = \boldsymbol{x_p} - \kappa\boldsymbol{dx_p}$ with $\kappa$ a random number in $[0.1, 0.5]$. In practice, we separate filter directions from their magnitude $1/M_i$ and learn them independently as suggested in Doumanoglou et al. (2024). For enforcing $||\boldsymbol{w}_i||_2 = 1$ (i.e. unit norm filter vectors) we use parametrization on the unit hyper-sphere.

**Details on Datasets used to Compute Concept Sensitivity with RCAV** To conduct concept sensitivity testing with RCAV we need access to a labeled dataset coming from the domain of the model. In practice, it is common to use the validation split of the dataset used to train the model. To mitigate the required computation time, we limited the size of the validation datasets as follows: For ImageNet, we used the validation split of ImageNet-S-300 Gao et al. (2022), and for Moments-in-Time, we considered Moments-In-Time Mini. Since Moments-In-Time is a video dataset, but the model that we studied works in the image domain, we constructed an image dataset by sampling 3 frames equally apart from each video (i.e. the first, the mid, and the last frame of the video). For Places365, we didn't make a size reduction as the validation split of the dataset was manageable with our resources.

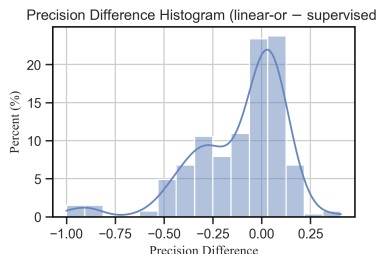 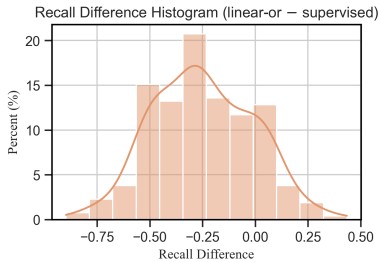 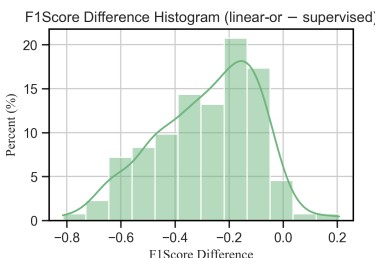

Figure 25: Interpretability Comparison. Histogram of differences in binary metrics: Precision, Recall, F1Score between the *Linear-OR* set of concept detectors learned with EDDP-C, $I = 512$, and classifiers learned in a supervised way (IBD Zhou et al. (2018)). The network here is ResNet18 trained on Places365.

**Details on RCAV's Statistical Significance Test** In all experiments, we set RCAV's perturbation hyperparameter, to $\alpha = 5$. For direction significance testing, we use RCAV's label permutation test. To construct random noise signal vectors, we (a) construct a dataset of feature-(binary) label pairs based on the decision rule of each one of the concept detectors. To deal with great class imbalance, we construct a pool of negative samples that is at most 20 times more than the positive ones; (b) we construct $N$ noisy versions of that dataset by label permutation; (c) we learn a noise-classifier to distinguish features based on the permuted labels, and (d) we concurrently estimate a noise-signal vector using (1) and the subsampling process described in Section 4.3. To learn each one of the noise signal vectors, and before permuting the labels, we construct a balanced dataset of at most 5000 samples, picked randomly from the pool. We train the noise classifiers using Adam for 100 epochs and a learning rate 0.01. By using noise signal vectors as RCAV's noisy directions, and with the number of those vectors per classifier set to $N = 100$, we subsequently calculate RCAV's p-values. We apply Bonferroni correction to all p-values by dividing the significance threshold $p = 0.05$ by the number of concept detectors $I$ and the number of model classes.

**Learning Details for NMF** When learning directions with NMF, we used the implementation found in Yu (2020). To make computations manageable, we reduce the spatial dimensionality of image representations by applying Adaptive Average Pooling to a (2,2) resolution, in a similar way as it was done in Fel et al. (2023b). In all cases, we learn the NMF decomposition by running the algorithm for 100000 iterations, with beta = 2 and a tolerance of $1e - 6$.

**Details for Comparing with the Supervised Approach** When learning directions with IBD, for each concept, a dataset is assembled that includes up to 20 times more negative samples than positive ones to address the significant imbalance. Additionally, we use hard negative mining as originally suggested in Zhou et al. (2018).

Figure 25 plots a histogram of classification metric differences between the *Linear-OR* set of classifiers and the classifiers learned in a supervised way. The differences are based on the concept labels, effectively taking the difference of metrics that regard two classifiers (the first from the *Linear-OR* set and the second from Zhou et al. (2018)) with the same concept name.

Figures 26, 27, 28 depict concrete binary classification metrics for some of the concept detectors in the *Linear-OR* set of classifiers, comparing them with concept detectors learned with supervision.

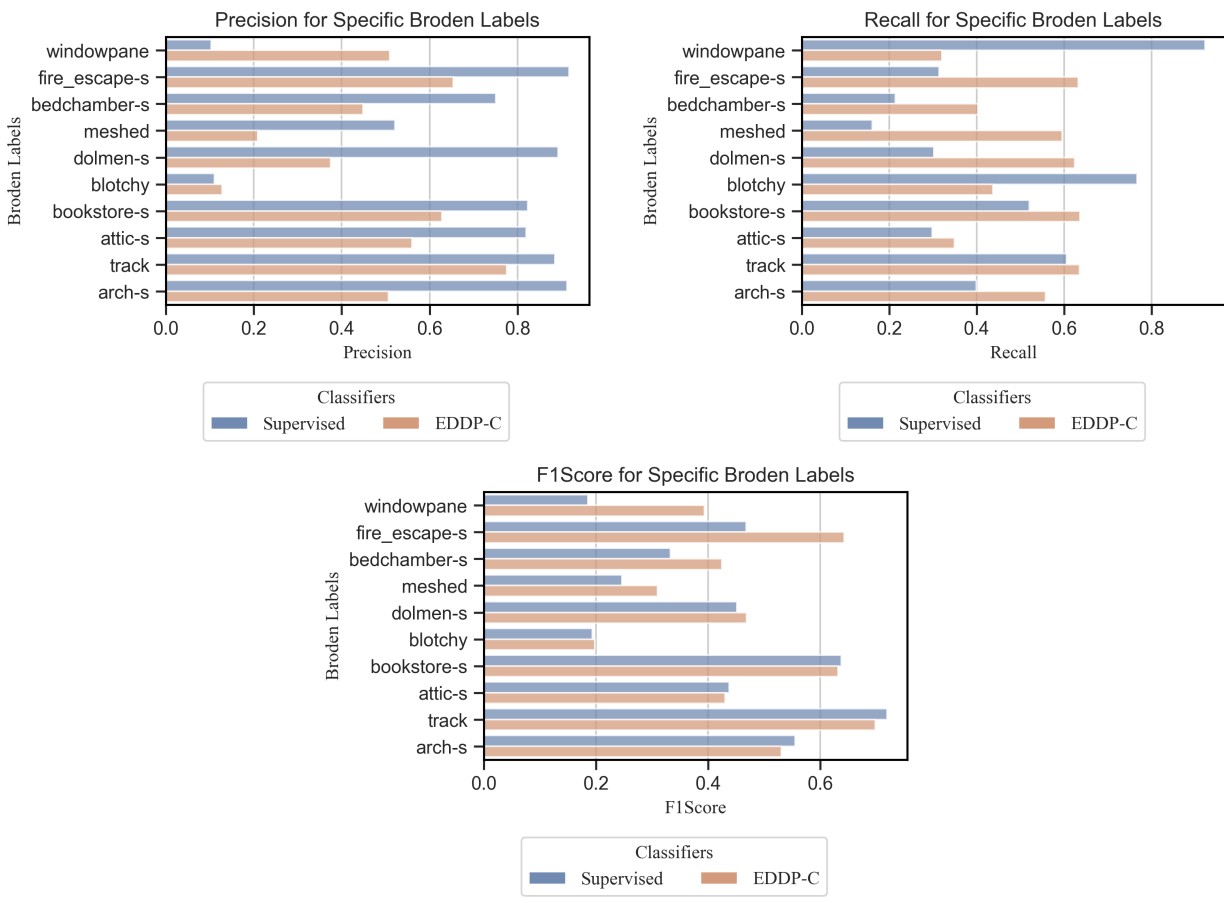

Figure 26: Interpretability Comparison. Exact Precision/Recall/F1Scores for specific concepts in Broden: comparison between the *Linear-OR* set of classifiers learned with EDDP-C, $I = 512$, and classifiers learned in a supervised way (IBD Zhou et al. (2018)). The network here is ResNet18 trained on Places365.

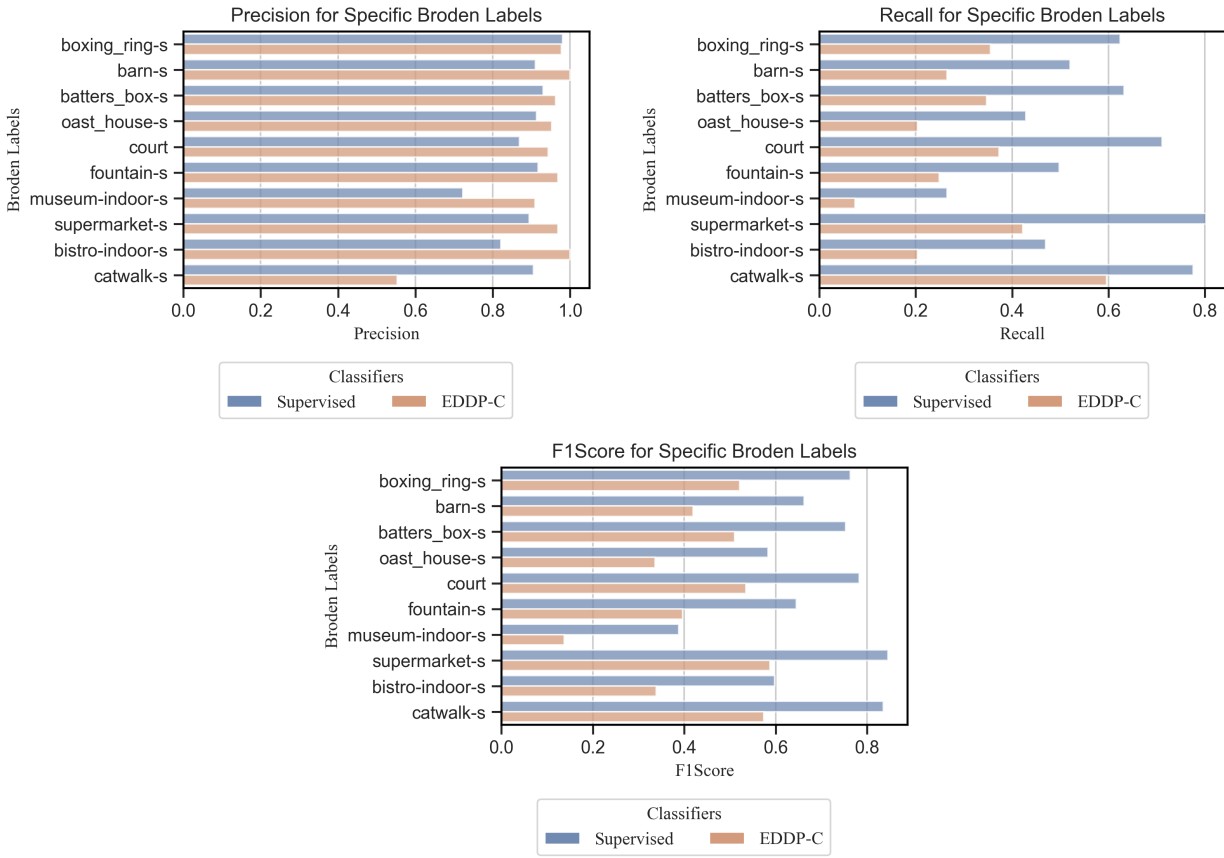

Figure 27: Interpretability Comparison. Exact Precision/Recall/F1Scores for specific concepts in Broden: comparison between the *Linear-OR* set of classifiers learned with EDDP-C, $I = 512$, and classifiers learned in a supervised way (IBD Zhou et al. (2018)). The network here is ResNet18 trained on Places365.

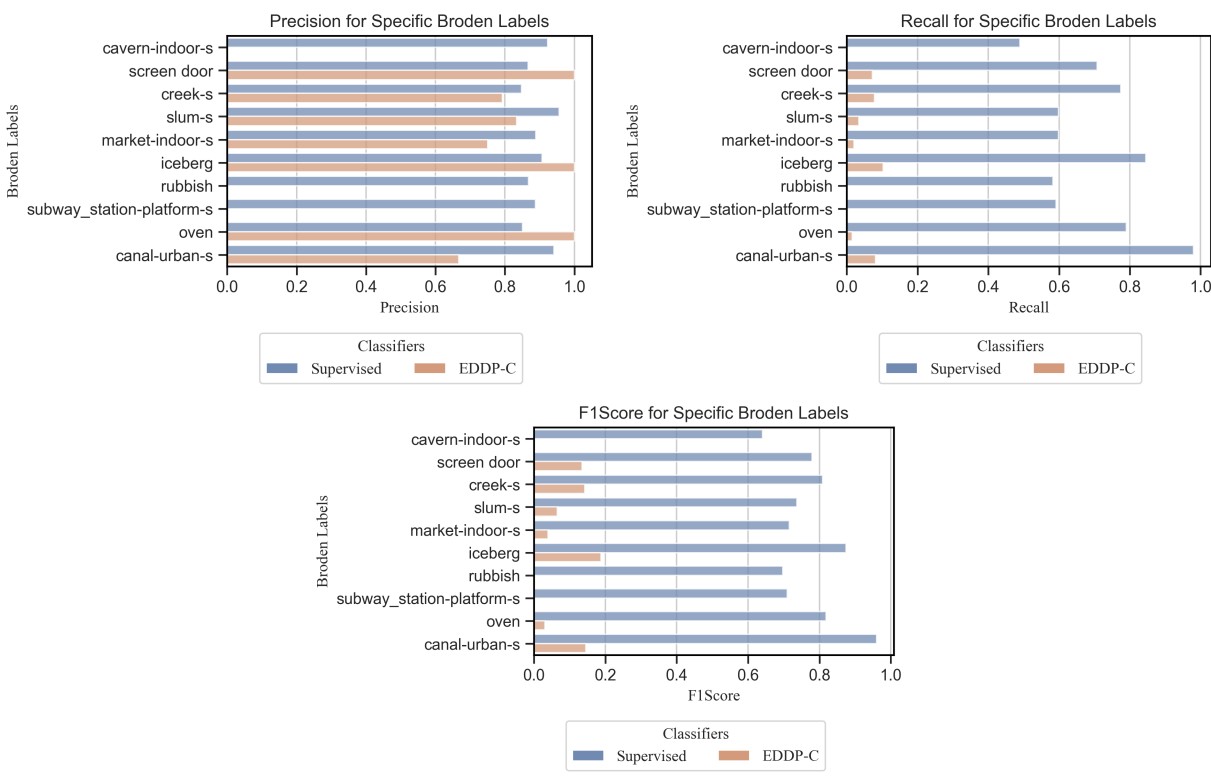

Figure 28: Interpretability Comparison. Exact Precision/Recall/F1Scores for specific concepts in Broden: comparison between the *Linear-OR* set of classifiers learned with EDDP-C, $I = 512$, and classifiers learned in a supervised way (IBD Zhou et al. (2018)). The network here is ResNet18 trained on Places365.

## A.14 Hyper-Parameter Study with Respect to Target Separation Margin

Table 17: Comparing EDDP variants based on varying target margin loss $\tau^{mm}$. The network here is ResNet18 trained on Places365 and $I = 448$.

| | | | | | | | | | | | |
|---|---|---|---|---|---|---|---|---|---|---|---|
| **ResNet18 / Places365** | | | | | | | | | | | |
| Method | $\tau^{mm}$ | Coverage ↑ | Redundancy ↓ | Precision ↑ | Recall ↑ | F1 ↑ | AP ↑ | $\mathcal{M}$ ↓ | $\mathcal{S}^1$ ↑ | $\mathcal{S}^2$ ↑ | mIoU ↑ |
| EDDP-U | 4.0 | 0.85 | **1.27** | 0.82±0.21 | **0.23±0.18** | **0.32±0.19** | **0.54±0.19** | 6.62 | 45.92 | 32.29 | **0.12** |
| EDDP-U | 5.0 | **0.86** | 1.32 | **0.83±0.19** | 0.22±0.16 | **0.32±0.18** | 0.53±0.19 | 6.64 | 47.5 | 32.07 | 0.11 |
| EDDP-U | 6.0 | **0.86** | 1.32 | **0.83±0.2** | 0.21±0.15 | 0.31±0.17 | 0.52±0.19 | 6.7 | **48.84** | **32.44** | 0.11 |
| EDDP-C | 4.0 | **0.86** | **1.27** | **0.83±0.21** | 0.22±0.17 | 0.31±0.19 | **0.52±0.2** | 6.67 | 44.71 | **31.73** | **0.11** |
| EDDP-C | 5.0 | **0.86** | 1.33 | 0.82±0.2 | 0.21±0.16 | **0.31±0.18** | 0.49±0.19 | 6.72 | **46.3** | 31.34 | 0.11 |
| EDDP-C | 6.0 | **0.86** | 1.31 | **0.83±0.2** | 0.2±0.15 | 0.29±0.17 | 0.5±0.18 | 6.75 | 46.01 | 30.97 | 0.1 |

Table 18: Comparing EDDP variants based on varying target margin loss $\tau^{mm}$. The network here is EfficientNet trained on ImageNet and $I = 1120$.

| | | | | | | | | | | | |
|---|---|---|---|---|---|---|---|---|---|---|---|
| **EfficientNet / ImageNet** | | | | | | | | | | | |
| Method | $\tau^{mm}$ | Coverage ↑ | Redundancy ↓ | Precision ↑ | Recall ↑ | F1 ↑ | AP ↑ | $\mathcal{M}$ ↓ | $\mathcal{S}^1$ ↑ | $\mathcal{S}^2$ ↑ | mIoU ↑ |
| EDDP-U | 4.0 | 0.85 | **1.54** | **0.77±0.22** | **0.22±0.18** | **0.3±0.19** | **0.42±0.19** | 6.74 | **37.45** | **23.0** | **0.03** |
| EDDP-U | 5.0 | 0.85 | 1.61 | 0.76±0.21 | 0.2±0.17 | 0.28±0.17 | 0.39±0.17 | 6.73 | 31.36 | 16.54 | **0.03** |
| EDDP-U | 6.0 | 0.85 | 1.6 | 0.76±0.21 | **0.22±0.17** | **0.3±0.17** | 0.39±0.17 | **6.72** | 30.46 | 16.09 | **0.03** |
| EDDP-C | 4.0 | 0.85 | **1.52** | **0.77±0.22** | **0.22±0.18** | **0.3±0.19** | **0.42±0.19** | 6.79 | **34.93** | **22.28** | **0.03** |
| EDDP-C | 5.0 | 0.85 | 1.54 | 0.74±0.22 | 0.2±0.18 | 0.28±0.19 | 0.39±0.17 | **6.74** | 28.62 | 16.38 | **0.03** |
| EDDP-C | 6.0 | 0.85 | 1.57 | 0.75±0.21 | 0.21±0.18 | 0.29±0.17 | 0.39±0.17 | 6.77 | 28.24 | 16.19 | **0.03** |

Tables 17 and 18 consider varying the target margin $\tau^{mm}$ when learning EDDP. For EfficientNet, increasing the target classification margin (i.e., the linear separability of patch embeddings) leads to substantial improvements in both $\mathcal{S}^1$ and $\mathcal{S}^2$ for both EDDP-C and EDDP-U. We stress that increasing the margin is accomplished by decreasing $\tau^{mm}$, as the margin is inversely proportional to the target loss. For ResNet18, the results are more mixed: for either EDDP-U or EDDP-C, decreasing the margin mostly improves $\mathcal{S}^1$ while keeping similar $\mathcal{S}^2$. We emphasize that when $\tau^{mm} = 4.0$, the results for EfficientNet are substantially better than the metrics we reported in the main body of the paper, for which we used $\tau^{mm} = 5.0$ (Section A.13).

## A.15 Ablation Study with Respect to Filter-Signal Orthogonality Loss

Table 19: Ablation study of the method with respect to Uncertainty Region Alignment Losses $\mathcal{L}^{uur}, \mathcal{L}^{cur}$ and the use of Filter-Signal Orthogonality Loss $\mathcal{L}^{fso}$. The network here is ResNet18 trained on Places365.

| | | | | | | | | | | | | | |
|---|---|---|---|---|---|---|---|---|---|---|---|---|---|
| **ResNet18 / Places365** | | | | | | | | | | | | | |
| $I$ | $\mathcal{L}^{uur}$ | $\mathcal{L}^{cur}$ | $\mathcal{L}^{fso}$ | Coverage ↑ | Redundancy ↓ | Precision ↑ | Recall ↑ | F1 ↑ | AP ↑ | $\mathcal{M}$ ↓ | $\mathcal{S}^1$ ↑ | $\mathcal{S}^2$ ↑ | mIoU ↑ | $\mathcal{I}^1$ ↑ |
| 384 | ✓ | ✗ | ✗ | 0.86 | 1.29 | **0.82±0.21** | **0.23±0.16** | **0.33±0.18** | **0.53±0.19** | 6.61 | 41.73 | 28.01 | **0.12** | 0.58 |
| | ✗ | ✓ | ✗ | 0.86 | **1.26** | 0.82±0.22 | 0.22±0.16 | 0.32±0.18 | 0.52±0.19 | 6.62 | 40.43 | 27.17 | **0.12** | 0.6 |
| | ✓ | ✗ | ✓ | 0.86 | 1.3 | 0.81±0.21 | **0.23±0.16** | **0.33±0.17** | 0.52±0.19 | 6.65 | 41.43 | 27.58 | **0.12** | **0.61** |
| | ✗ | ✓ | ✓ | 0.86 | 1.32 | 0.81±0.21 | **0.23±0.16** | **0.33±0.18** | 0.5±0.19 | 6.67 | 41.5 | 27.49 | **0.12** | **0.61** |
| 448 | ✓ | ✗ | ✗ | 0.86 | 1.32 | **0.83±0.19** | **0.22±0.16** | **0.32±0.18** | **0.53±0.19** | 6.64 | 47.5 | **32.07** | 0.11 | 0.59 |
| | ✗ | ✓ | ✗ | 0.86 | **1.29** | **0.83±0.19** | 0.21±0.16 | 0.3±0.18 | 0.52±0.19 | 6.66 | 45.21 | 31.1 | 0.11 | 0.6 |
| | ✓ | ✗ | ✓ | 0.86 | 1.33 | 0.82±0.2 | **0.22±0.16** | **0.32±0.18** | 0.51±0.19 | 6.68 | **47.57** | 31.95 | 0.11 | **0.61** |
| | ✗ | ✓ | ✓ | 0.86 | 1.33 | 0.82±0.2 | 0.21±0.16 | 0.31±0.18 | 0.49±0.19 | 6.72 | 46.3 | 31.34 | 0.11 | **0.61** |
| 512 | ✓ | ✗ | ✗ | 0.86 | 1.28 | 0.82±0.22 | **0.21±0.17** | **0.31±0.19** | 0.51±0.21 | 6.66 | **52.51** | **37.78** | 0.11 | 0.61 |
| | ✗ | ✓ | ✗ | 0.85 | **1.23** | 0.82±0.23 | 0.21±0.17 | 0.3±0.19 | 0.51±0.2 | 6.67 | 50.63 | 37.35 | 0.1 | 0.62 |
| | ✓ | ✗ | ✓ | 0.86 | 1.3 | 0.82±0.21 | 0.21±0.17 | 0.31±0.19 | 0.48±0.2 | 6.72 | 52.3 | 37.71 | 0.1 | 0.62 |
| | ✗ | ✓ | ✓ | 0.86 | 1.28 | 0.81±0.24 | 0.2±0.16 | 0.29±0.19 | 0.47±0.2 | 6.73 | 50.29 | 36.99 | 0.1 | **0.63** |

In this Section we conduct a detailed ablation study with respect to the use of Filter-Signal Orthogonality Loss ($\mathcal{L}^{fso}$) that was introduced in Section 4.3. In Section A.12 we discussed the impact of $\mathcal{L}^{fso}$ on the faithfulness of the direction pairs, while in this Section we discuss the effects of $\mathcal{L}^{fso}$, in interpretability and influence terms. Tables 19, 20, 21, 22, 23, 24 depict results in terms of interpretability and influence metrics.

The main observations can be summarized to the following: a) When $\mathcal{L}^{fso}$ is not used, using any of the two Uncertainty Region Alignment variants, in the majority of the cases, results in comparable sets of directions, with $\mathcal{L}^{uur}$ typically leading to more interpretable directions than $\mathcal{L}^{cur}$. b) In case $\mathcal{L}^{fso}$ is taken

Table 20: Ablation study of the method with respect to Uncertainty Region Alignment Losses $\mathcal{L}^{uur}, \mathcal{L}^{cur}$ and the use of Filter-Signal Orthogonality Loss $\mathcal{L}^{fso}$. The network here is EfficientNet trained on ImageNet.

| | | | | | | EfficientNet / ImageNet | | | | | | | | |
|---|---|---|---|---|---|---|---|---|---|---|---|---|---|---|
| $I$ | $\mathcal{L}^{uur}$ | $\mathcal{L}^{cur}$ | $\mathcal{L}^{fso}$ | Coverage ↑ | Redundancy ↓ | Precision ↑ | Recall ↑ | F1 ↑ | AP ↑ | $\mathcal{M}$ ↓ | $\mathcal{S}^1$ ↑ | $\mathcal{S}^2$ ↑ | mIoU ↑ | $\mathcal{I}^1$ ↑ |
| 960 | ✓ | ✗ | ✗ | **0.85** | 1.51 | **0.75±0.22** | **0.2±0.17** | **0.27±0.17** | **0.38±0.16** | 6.74 | **27.92** | **14.71** | **0.03** | **0.94** |
| | ✗ | ✓ | ✗ | **0.85** | **1.48** | 0.74±0.22 | **0.2±0.18** | **0.27±0.17** | 0.37±0.16 | 6.74 | 26.5 | 14.57 | **0.03** | **0.94** |
| | ✓ | ✗ | ✓ | **0.85** | 1.49 | 0.74±0.23 | **0.2±0.18** | **0.27±0.17** | **0.38±0.17** | 6.76 | 26.74 | 14.58 | **0.03** | **0.94** |
| | ✗ | ✓ | ✓ | **0.85** | **1.48** | 0.72±0.23 | **0.2±0.18** | **0.27±0.17** | **0.38±0.17** | 6.75 | 25.92 | 14.45 | **0.03** | **0.94** |
| 1120 | ✓ | ✗ | ✗ | **0.85** | 1.61 | **0.76±0.21** | **0.2±0.17** | **0.28±0.17** | **0.39±0.17** | 6.73 | **31.36** | **16.54** | **0.03** | **0.94** |
| | ✗ | ✓ | ✗ | **0.85** | 1.56 | **0.76±0.21** | **0.2±0.18** | **0.28±0.17** | 0.38±0.17 | **6.72** | 29.79 | 16.39 | **0.03** | 0.93 |
| | ✓ | ✗ | ✓ | **0.85** | 1.56 | 0.74±0.22 | **0.2±0.18** | **0.28±0.18** | **0.39±0.17** | 6.76 | 29.54 | 16.48 | **0.03** | **0.94** |
| | ✗ | ✓ | ✓ | **0.85** | **1.54** | 0.74±0.22 | **0.2±0.18** | **0.28±0.19** | **0.39±0.17** | 6.74 | 28.62 | 16.38 | **0.03** | **0.94** |
| 1280 | ✓ | ✗ | ✗ | **0.85** | 1.65 | **0.74±0.21** | **0.19±0.16** | **0.27±0.16** | **0.36±0.17** | 6.81 | **40.38** | **19.12** | **0.03** | **0.94** |
| | ✗ | ✓ | ✗ | **0.85** | 1.67 | **0.74±0.21** | **0.19±0.17** | **0.27±0.17** | **0.36±0.17** | 6.8 | 36.5 | 18.67 | **0.03** | 0.93 |
| | ✓ | ✗ | ✓ | **0.85** | 1.63 | 0.73±0.2 | **0.19±0.17** | 0.26±0.17 | **0.36±0.17** | 6.9 | 37.67 | 18.96 | **0.03** | **0.94** |
| | ✗ | ✓ | ✓ | **0.85** | **1.6** | 0.73±0.21 | **0.19±0.17** | 0.26±0.17 | **0.36±0.17** | 6.88 | 36.95 | 18.98 | **0.03** | **0.94** |

Table 21: Ablation study of the method with respect to Uncertainty Region Alignment Losses $\mathcal{L}^{uur}, \mathcal{L}^{cur}$ and the use of Filter-Signal Orthogonality Loss $\mathcal{L}^{fso}$. The network here is Inception-v3 trained on ImageNet.

| | | | | | | Inception-v3 / ImageNet | | | | | | | | |
|---|---|---|---|---|---|---|---|---|---|---|---|---|---|---|
| $I$ | $\mathcal{L}^{uur}$ | $\mathcal{L}^{cur}$ | $\mathcal{L}^{fso}$ | Coverage ↑ | Redundancy ↓ | Precision ↑ | Recall ↑ | F1 ↑ | AP ↑ | $\mathcal{M}$ ↓ | $\mathcal{S}^1$ ↑ | $\mathcal{S}^2$ ↑ | mIoU ↑ | $\mathcal{I}^1$ ↑ |
| 1536 | ✓ | ✗ | ✗ | **0.87** | 2.41 | 0.84±0.18 | 0.28±0.24 | 0.38±0.26 | 0.53±0.26 | **6.79** | **156.46** | **28.03** | **0.11** | 0.93 |
| | ✗ | ✓ | ✗ | 0.85 | 2.09 | 0.85±0.19 | **0.3±0.23** | **0.4±0.25** | 0.56±0.25 | 6.86 | 156.42 | 27.88 | **0.11** | 0.93 |
| | ✓ | ✗ | ✓ | 0.85 | 1.94 | 0.85±0.21 | 0.22±0.16 | 0.33±0.19 | 0.57±0.24 | 6.88 | 123.85 | 27.08 | 0.08 | **0.94** |
| | ✗ | ✓ | ✓ | 0.85 | **1.9** | **0.86±0.19** | 0.23±0.16 | 0.33±0.19 | **0.58±0.24** | 6.87 | 125.64 | 27.46 | 0.08 | **0.94** |
| 1792 | ✓ | ✗ | ✗ | **0.9** | 2.96 | 0.86±0.16 | **0.33±0.25** | 0.43±0.26 | 0.58±0.25 | **6.77** | **210.44** | **25.77** | **0.13** | 0.93 |
| | ✗ | ✓ | ✗ | 0.85 | 2.45 | 0.88±0.16 | **0.33±0.25** | **0.44±0.26** | **0.59±0.24** | 6.84 | 208.6 | 24.98 | 0.12 | 0.93 |
| | ✓ | ✗ | ✓ | 0.87 | 2.23 | **0.89±0.16** | 0.24±0.16 | 0.35±0.19 | 0.62±0.21 | 6.89 | 173.17 | 24.93 | 0.1 | **0.94** |
| | ✗ | ✓ | ✓ | 0.86 | **2.19** | **0.89±0.16** | 0.24±0.16 | 0.36±0.19 | 0.62±0.21 | 6.92 | 174.0 | 24.69 | 0.1 | **0.94** |
| 2048 | ✓ | ✗ | ✗ | 0.86 | **2.36** | **0.85±0.2** | **0.31±0.29** | **0.4±0.29** | **0.54±0.28** | 6.95 | **170.81** | **32.13** | **0.09** | **0.94** |
| | ✗ | ✓ | ✗ | 0.9 | 4.85 | 0.77±0.18 | 0.3±0.27 | 0.39±0.27 | 0.48±0.26 | **6.91** | 165.65 | 30.12 | 0.08 | **0.94** |
| | ✓ | ✗ | ✓ | 0.92 | 4.0 | 0.75±0.18 | 0.16±0.12 | 0.24±0.14 | 0.38±0.16 | 7.09 | 92.53 | 31.14 | 0.05 | **0.94** |
| | ✗ | ✓ | ✓ | **0.93** | 3.98 | 0.75±0.18 | 0.16±0.12 | 0.24±0.14 | 0.38±0.16 | 7.09 | 92.67 | 31.18 | 0.05 | **0.94** |

into account, the use of $\mathcal{L}^{cur}$ can significantly improve interpretability compared to $\mathcal{L}^{uur}$ ($\mathcal{S}^1$ in VGG-16 and ResNet50) or remain comparable to $\mathcal{L}^{uur}$ (ResNet18, EfficientNet, and Inception-v3). c) When comparing among the same type of Uncertainty Region Alignment with and without $\mathcal{L}^{fso}$, in most cases, the use of $\mathcal{L}^{fso}$ hurts interpretability (especially F1-Score and $\mathcal{S}^1$) with the latter being more prominent in Inception-v3 ($I = 2048$) and all cluster sizes of VGG-16 and ResNet50. An interesting exception is that the use of $\mathcal{L}^{fso}$ improves cluster diversity ($\mathcal{S}^2$) in ResNet50. d) Under the same $\mathcal{L}^{fso}$ setting, using $\mathcal{L}^{cur}$ leads to directions with greater significant influence (SDC & SCDP metrics) compared to $\mathcal{L}^{uur}$, except for when using $\mathcal{L}^{fso}$ in ResNet50. e) When considering the same Uncertainty Region Alignment loss variant, the use of $\mathcal{L}^{fso}$ leads to an improvement in the SDC and SCDP metrics, with a notable exception being when using $\mathcal{L}^{cur}$ in ResNet18.

Table 22: Ablation study of the method with respect to Uncertainty Region Alignment Losses $\mathcal{L}^{uur}, \mathcal{L}^{cur}$ and the use of Filter-Signal Orthogonality Loss $\mathcal{L}^{fso}$. The network here is VGG16 trained on ImageNet.

| | | | | | | | | | | | | | | |
|---|---|---|---|---|---|---|---|---|---|---|---|---|---|---|
| colspan VGG16 / ImageNet | | | | | | | | | | | | | | |
| $I$ | $\mathcal{L}^{uur}$ | $\mathcal{L}^{cur}$ | $\mathcal{L}^{fso}$ | Coverage ↑ | Redundancy ↓ | Precision ↑ | Recall ↑ | F1 ↑ | AP ↑ | $\mathcal{M}$ ↓ | $\mathcal{S}^1$ ↑ | $\mathcal{S}^2$ ↑ | mIoU ↑ | $\mathcal{I}^1$ ↑ |
| 384 | ✓ | ✗ | ✗ | **0.82** | 1.17 | 0.61±0.18 | **0.18±0.09** | **0.27±0.09** | 0.31±0.12 | 6.88 | **29.98** | 9.39 | **0.08** | **0.91** |
| | ✗ | ✓ | ✗ | 0.8 | 1.09 | 0.62±0.17 | 0.17±0.09 | 0.26±0.09 | **0.31±0.11** | **6.87** | 28.12 | 9.09 | 0.07 | 0.9 |
| | ✓ | ✗ | ✓ | 0.79 | **1.05** | 0.6±0.17 | 0.16±0.09 | 0.24±0.09 | **0.31±0.11** | 7.12 | 23.42 | 8.51 | 0.06 | 0.9 |
| | ✗ | ✓ | ✓ | 0.79 | 1.07 | **0.63±0.18** | 0.16±0.09 | 0.24±0.09 | **0.31±0.12** | 6.92 | 26.23 | 8.89 | 0.07 | 0.9 |
| 448 | ✓ | ✗ | ✗ | **0.84** | 1.35 | 0.59±0.18 | **0.18±0.09** | 0.26±0.09 | **0.3±0.11** | 6.93 | **34.67** | 10.29 | **0.08** | **0.91** |
| | ✗ | ✓ | ✗ | 0.82 | 1.22 | 0.6±0.18 | 0.17±0.08 | 0.25±0.09 | **0.3±0.11** | **6.92** | 32.18 | 9.53 | 0.07 | 0.9 |
| | ✓ | ✗ | ✓ | 0.82 | **1.18** | 0.58±0.17 | 0.15±0.08 | 0.23±0.09 | **0.3±0.11** | 7.14 | 26.72 | 8.75 | 0.06 | 0.9 |
| | ✗ | ✓ | ✓ | 0.82 | 1.22 | **0.61±0.18** | 0.16±0.08 | 0.24±0.09 | **0.3±0.11** | 6.93 | 30.91 | 9.26 | 0.07 | 0.9 |
| 512 | ✓ | ✗ | ✗ | **0.85** | 1.43 | 0.61±0.18 | **0.18±0.08** | **0.27±0.09** | **0.31±0.11** | 6.95 | **41.2** | **11.11** | **0.08** | **0.91** |
| | ✗ | ✓ | ✗ | **0.85** | 1.39 | **0.62±0.17** | **0.18±0.08** | 0.26±0.09 | 0.3±0.11 | 6.96 | 39.23 | 10.88 | **0.08** | 0.9 |
| | ✓ | ✗ | ✓ | **0.85** | **1.35** | 0.59±0.18 | 0.15±0.09 | 0.22±0.09 | 0.29±0.11 | 7.23 | 28.62 | 10.4 | 0.06 | 0.9 |
| | ✗ | ✓ | ✓ | **0.85** | 1.37 | **0.62±0.18** | 0.15±0.09 | 0.24±0.09 | 0.3±0.11 | 7.09 | 33.91 | 10.83 | 0.07 | 0.9 |

Table 23: Ablation study of the method with respect to Uncertainty Region Alignment Losses $\mathcal{L}^{uur}, \mathcal{L}^{cur}$ and the use of Filter-Signal Orthogonality Loss $\mathcal{L}^{fso}$. The network here is ResNet50 trained on Moments in Time.

| | | | | | | | | | | | | | | |
|---|---|---|---|---|---|---|---|---|---|---|---|---|---|---|
| colspan ResNet50 / MiT | | | | | | | | | | | | | | |
| $I$ | $\mathcal{L}^{uur}$ | $\mathcal{L}^{cur}$ | $\mathcal{L}^{fso}$ | Coverage ↑ | Redundancy ↓ | Precision ↑ | Recall ↑ | F1 ↑ | AP ↑ | $\mathcal{M}$ ↓ | $\mathcal{S}^1$ ↑ | $\mathcal{S}^2$ ↑ | mIoU ↑ | $\mathcal{I}^1$ ↑ |
| 1536 | ✓ | ✗ | ✗ | 0.84 | **2.13** | 0.72±0.22 | **0.17±0.12** | **0.26±0.15** | 0.36±0.16 | **6.25** | 127.83 | 33.67 | 0.08 | **0.85** |
| | ✗ | ✓ | ✗ | 0.83 | 2.33 | **0.75±0.16** | **0.17±0.12** | **0.26±0.15** | 0.35±0.15 | **6.25** | **141.4** | 30.12 | **0.09** | **0.85** |
| | ✓ | ✗ | ✓ | **0.88** | 2.39 | 0.74±0.19 | **0.17±0.12** | 0.25±0.14 | 0.44±0.14 | 6.59 | 94.24 | **35.05** | 0.06 | 0.84 |
| | ✗ | ✓ | ✓ | **0.88** | 2.37 | **0.75±0.19** | **0.17±0.11** | **0.26±0.13** | 0.45±0.14 | 6.49 | 98.52 | 34.81 | 0.06 | 0.84 |
| 1792 | ✓ | ✗ | ✗ | 0.85 | **2.47** | **0.78±0.16** | **0.19±0.10** | **0.29±0.13** | 0.4±0.14 | **6.24** | 162.86 | 34.17 | **0.09** | **0.85** |
| | ✗ | ✓ | ✗ | 0.83 | 2.61 | 0.76±0.16 | **0.19±0.11** | **0.29±0.14** | 0.38±0.15 | 6.27 | **164.82** | 30.63 | **0.09** | **0.85** |
| | ✓ | ✗ | ✓ | **0.89** | 2.66 | 0.75±0.17 | 0.16±0.11 | 0.25±0.13 | 0.43±0.13 | 6.55 | 109.36 | 36.48 | 0.06 | 0.84 |
| | ✗ | ✓ | ✓ | 0.88 | 2.62 | 0.76±0.16 | 0.16±0.11 | 0.25±0.13 | **0.44±0.13** | 6.5 | 113.9 | **36.68** | 0.06 | 0.84 |
| 2048 | ✓ | ✗ | ✗ | 0.86 | 3.15 | 0.72±0.2 | **0.21±0.13** | **0.31±0.16** | 0.43±0.16 | **6.29** | **209.2** | 33.04 | **0.1** | **0.85** |
| | ✗ | ✓ | ✗ | 0.83 | **2.7** | **0.76±0.15** | 0.2±0.11 | 0.3±0.14 | 0.42±0.15 | 6.36 | 205.22 | 32.78 | **0.1** | **0.85** |
| | ✓ | ✗ | ✓ | **0.89** | 2.9 | 0.72±0.16 | 0.15±0.11 | 0.23±0.12 | 0.41±0.13 | 6.61 | 120.5 | 35.09 | 0.06 | 0.84 |
| | ✗ | ✓ | ✓ | 0.88 | 2.83 | 0.72±0.16 | 0.15±0.11 | 0.24±0.12 | **0.43±0.13** | 6.61 | 126.46 | **35.98** | 0.06 | 0.84 |

Table 24: Ablation study of the method with respect to Uncertainty Region Alignment Losses $\mathcal{L}^{uur}, \mathcal{L}^{cur}$ and the use of Filter-Signal Orthogonality Loss $\mathcal{L}^{fso}$. The Table depicts metrics of statistical significant influence. For ResNet18, $I = 448$, for EfficientNet, $I = 1120$ and for ResNet50, $I = 1792$.

| $\mathcal{L}^{uur}$ | $\mathcal{L}^{cur}$ | $\mathcal{L}^{fso}$ | ResNet18 / Places365 SDC | SCDP | EfficientNet / ImageNet SDC | SCDP | ResNet50 / MiT SDC | SCDP |
|---|---|---|---|---|---|---|---|---|
| ✓ | ✗ | ✗ | 264 | 1565 | 107 | 243 | 393 | 570 |
| ✗ | ✓ | ✗ | **322** | **2285** | **426** | **714** | **1154** | **2250** |
| ✓ | ✗ | ✓ | **296** | 1786 | 668 | 1445 | **1676** | **7946** |
| ✗ | ✓ | ✓ | **296** | **1892** | **673** | **1484** | 1353 | 3913 |

### A.16 Comparison with Unsupervised Interpretable Basis Extraction and Concept-Basis Extraction in Practical Experiments

Table 25: Comparison and Ablation with respect to method enhancements that we propose in this work. $\mathcal{L}^{al}$ stands for Augmented Lagrangian Loss and $\mathcal{L}^{cc}$ refers to the CNN Classifier loss that was proposed in Doumanoglou et al. (2024). Methods below the dashed line correspond to variations that use the contributions we make in this work. Here the network is ResNet18 and $I = 512$.

| Method | Ortho | $\mathcal{L}^{al}$ | $\mathcal{L}^{cc}$ | $\mathcal{L}^{uur}$ | $\mathcal{L}^{cur}$ | $\mathcal{S}^1$ | $\mathcal{S}^2$ |
|---|---|---|---|---|---|---|---|
| **ResNet18 / Places365** | | | | | | | |
| UIBE (Doumanoglou et al., 2023) | ✓ | ✗ | ✗ | ✗ | ✗ | 60.93 | 28.39 |
| CBE (Doumanoglou et al., 2024) | ✓ | ✗ | ✓ | ✗ | ✗ | **69.43** | 31.53 |
| CBE /w $\mathcal{L}^{uur}$ | ✓ | ✗ | ✗ | ✓ | ✗ | 67.3 | 32.16 |
| EDDP-U | ✗ | ✓ | ✗ | ✓ | ✗ | 52.51 | **37.78** |
| EDDP-C | ✗ | ✓ | ✗ | ✗ | ✓ | 50.29 | 36.99 |

Table 26: Comparison and Ablation with respect to method enhancements that we propose in this work. $\mathcal{L}^{al}$ stands for Augmented Lagrangian Loss and $\mathcal{L}^{cc}$ refers to the CNN Classifier loss that was proposed in Doumanoglou et al. (2024). Methods below the dashed line correspond to variations that use the contributions we make in this work. Here the network is ResNet50 and $I = 2048$.

| Method | Ortho | $\mathcal{L}^{al}$ | $\mathcal{L}^{cc}$ | $\mathcal{L}^{uur}$ | $\mathcal{L}^{cur}$ | $\mathcal{S}^1$ | $\mathcal{S}^2$ |
|---|---|---|---|---|---|---|---|
| **ResNet50 / MiT** | | | | | | | |
| UIBE (Doumanoglou et al., 2023) | ✓ | ✗ | ✗ | ✗ | ✗ | 124.73 | 18.47 |
| CBE (Doumanoglou et al., 2024) | ✓ | ✗ | ✓ | ✗ | ✗ | 131.73 | 26.94 |
| CBE/w $\mathcal{L}^{uur}$ | ✓ | ✗ | ✗ | ✓ | ✗ | 158.76 | 33.02 |
| EDDP-U | ✗ | ✓ | ✗ | ✓ | ✗ | **209.2** | 33.04 |
| EDDP-C | ✗ | ✓ | ✗ | ✗ | ✓ | 126.46 | **35.98** |

Tables 25 and 26 briefly present comparisons of our work with the previous works that we extend, in terms of interpretability. First, with other aspects of Concept Basis Extraction (CBE) (Doumanoglou et al., 2024) being intact, we assess the efficacy of our Unconstrained Uncertainty Region Alignment loss in improving the interpretability of the clustering, compared to the CNN Classifier Loss $\mathcal{L}^{cc}$ that was proposed previously in Doumanoglou et al. (2024). This case is referred to in the tables as *CBE /w $\mathcal{L}^{uur}$*. We observe that in three out of the four cases, our Uncertainty Region Alignment loss leads to a notable relative improvement, by up to 22.56% in $\mathcal{S}^2$, which indicates a clustering with improved concept diversity. When we additionally take into account the rest of our contributions, and compare EDDP-U with CBE, we find that in the same three out of four cases, interpretability is further improved by up to 58.8% in terms of $\mathcal{S}^1$. Transitioning from CBE to EDDP-C, may lead to further improvements in $\mathcal{S}^2$ by up to 33.5%, yet with a decreased score in terms of $\mathcal{S}^1$.

### A.17 More Qualitative Segmentations and Statistics for Evaluating the Interpretability of the Concept Detectors

Figures 29, 30, 31, 32, 33, 34, 37, 38, 39 depict qualitative segmentation results obtained using the concept detectors that were learned with the proposed method for the architectures that we studied in this work. Visualizations are obtained using Bau et al. (2017). We can verify that concept detectors appear to be monosemantic. In some cases, there are more than one concept detector detecting the same concept. However, in most cases, the sets of positively classified samples across detectors are disjoint. Figures 35 and 36 depict qualitative segmentations obtained via NMF and PCA. In some of the depicted examples, it is evident that the identified concepts are less monosemantic. Tables 27, 28, 29 and 30 summarize the statistics obtained by Network Dissection regarding the interpretability of the clusterings considered in this work.

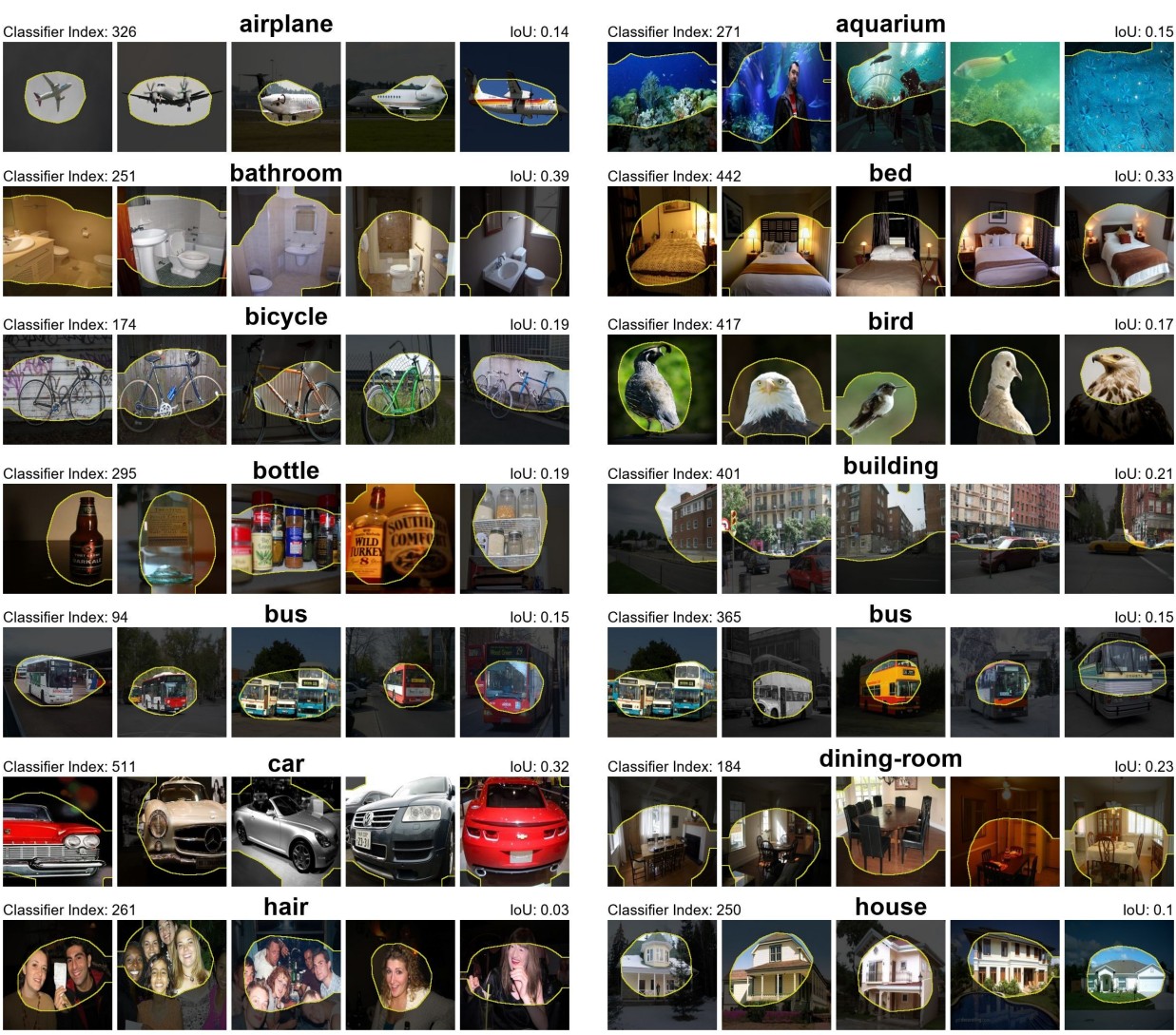

Figure 29: Qualitative segmentations using the concept detectors learned with our method. Here the network is ResNet18 trained on Places365, the method is using CFM, and $I = 512$.

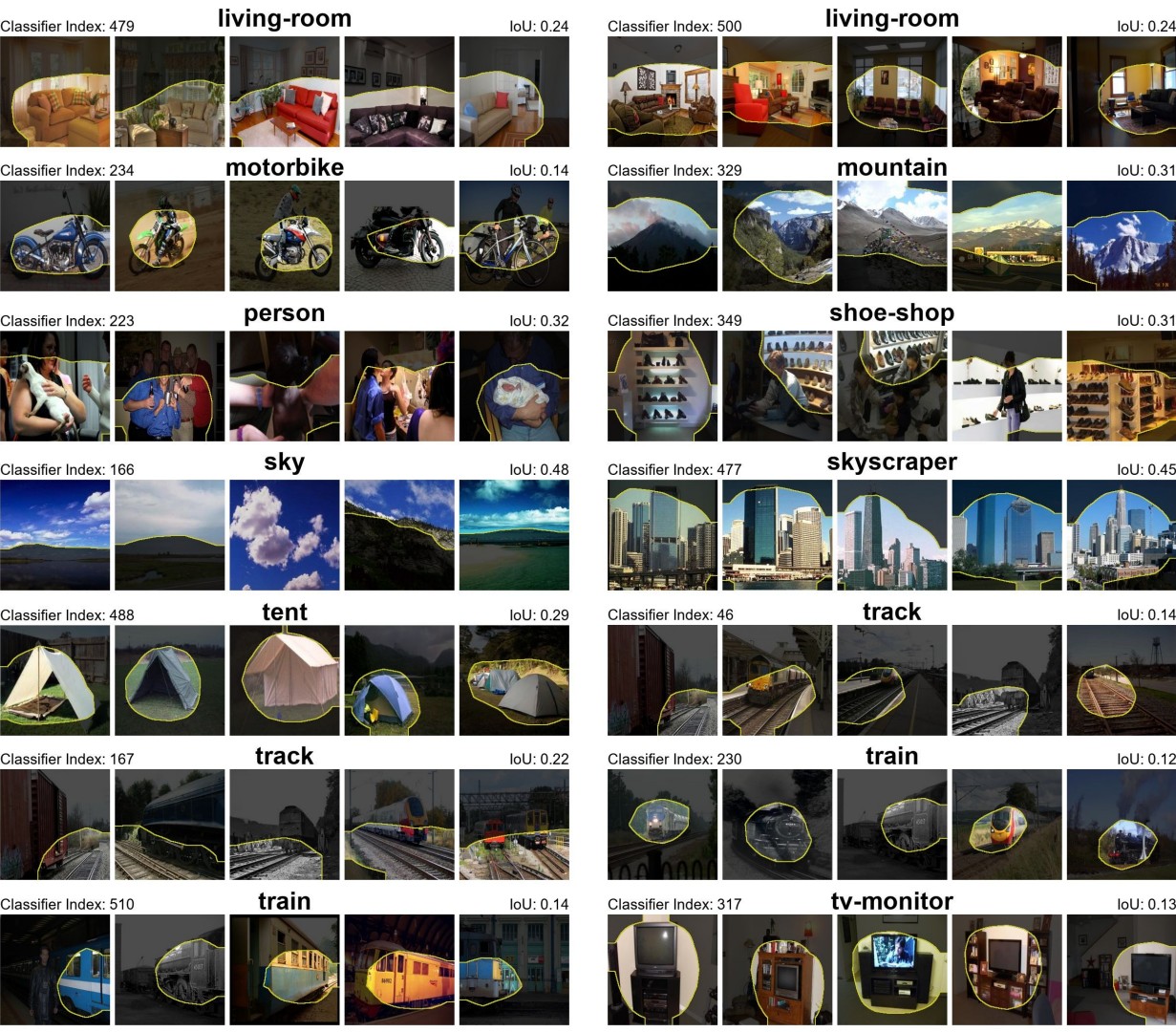

Figure 30: Qualitative segmentations using the concept detectors learned with our method. Here the network is ResNet18 trained on Places365, the method is using CFM, and $I = 512$.

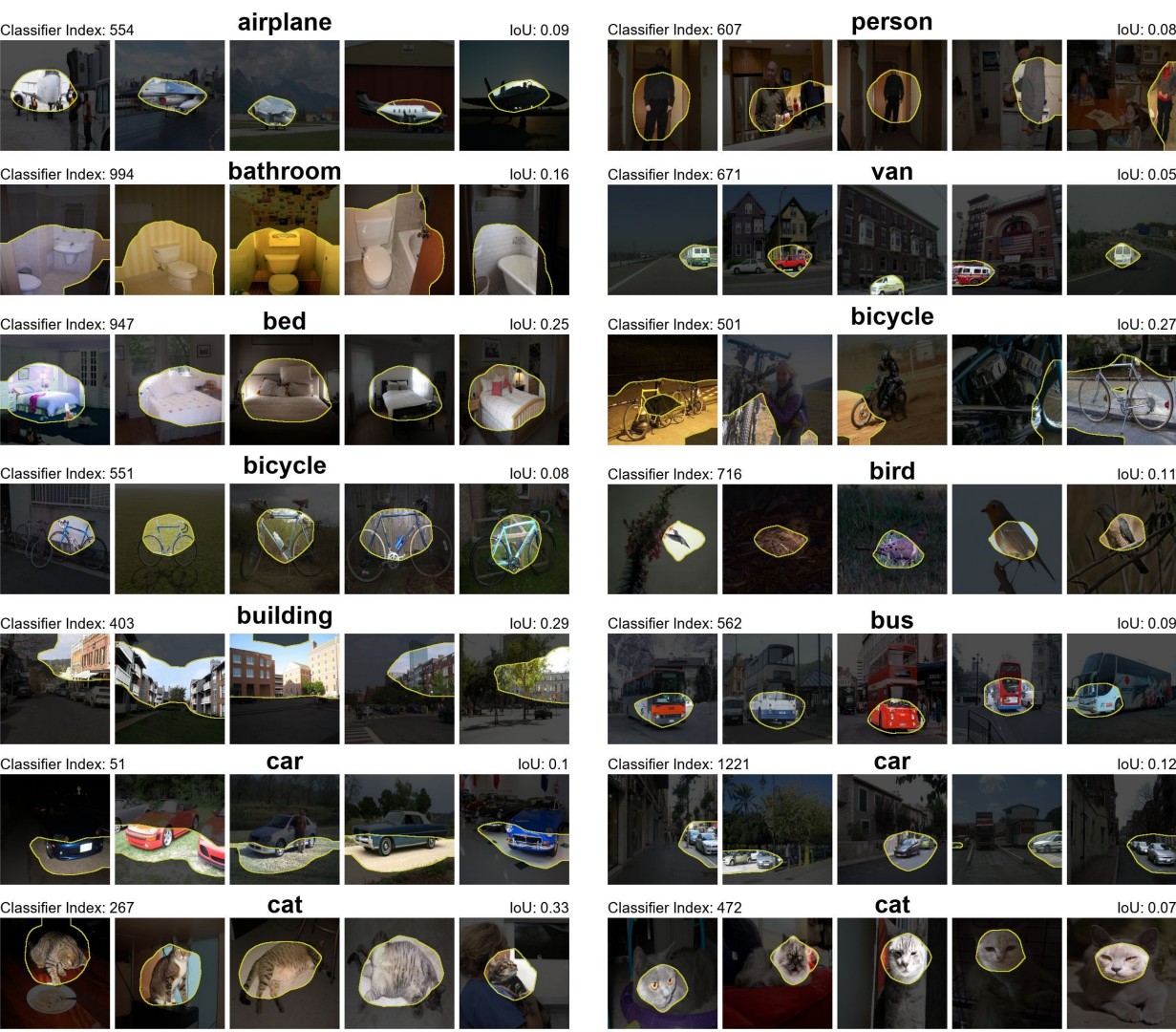

Figure 31: Qualitative segmentations using the concept detectors learned with our method. Here the network is EfficientNet trained on ImageNet, the method is using CFM, and $I = 1280$.

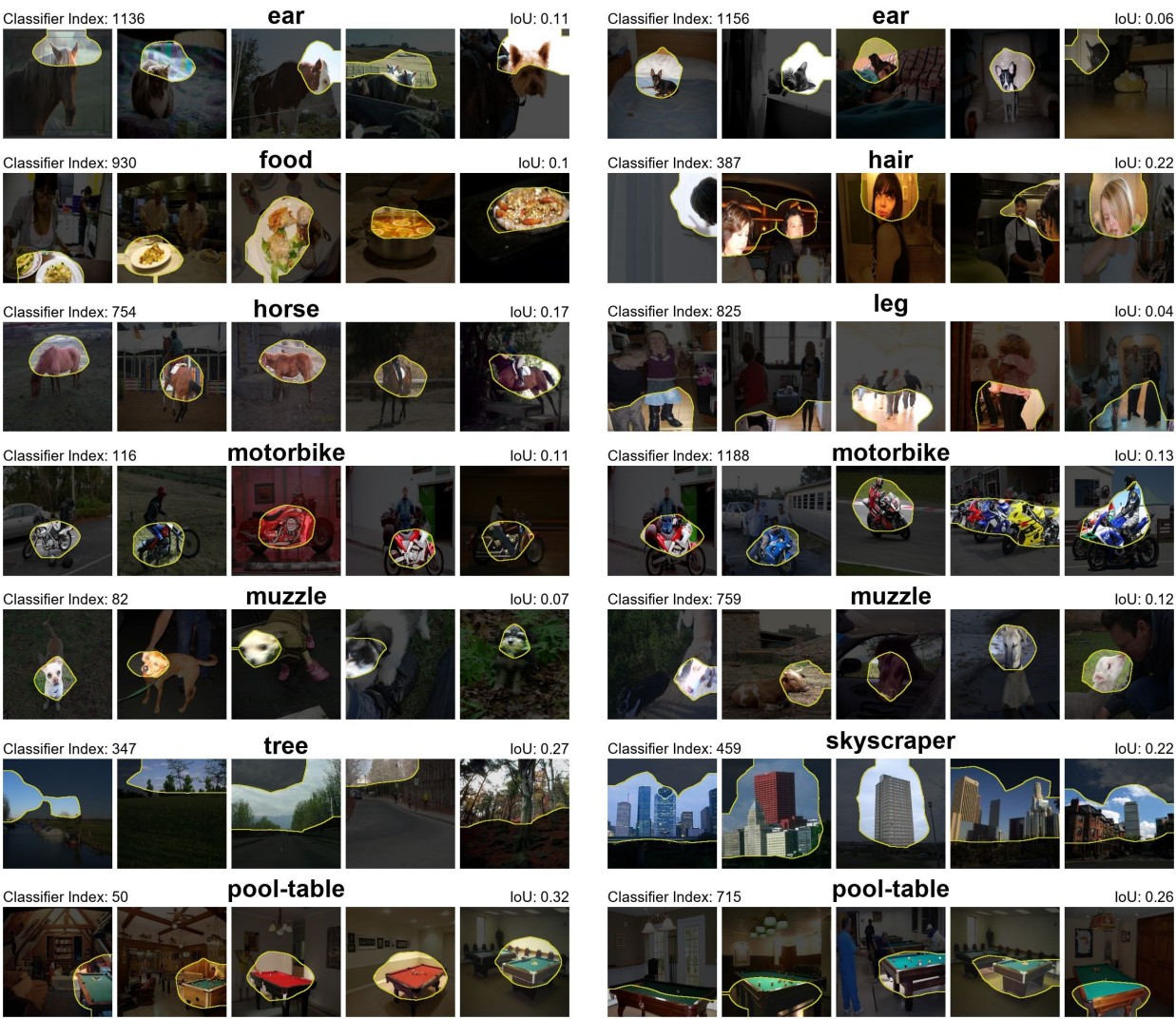

Figure 32: Qualitative segmentations using the concept detectors learned with our method. Here the network is EfficientNet trained on ImageNet, the method is using CFM, and $I = 1280$.

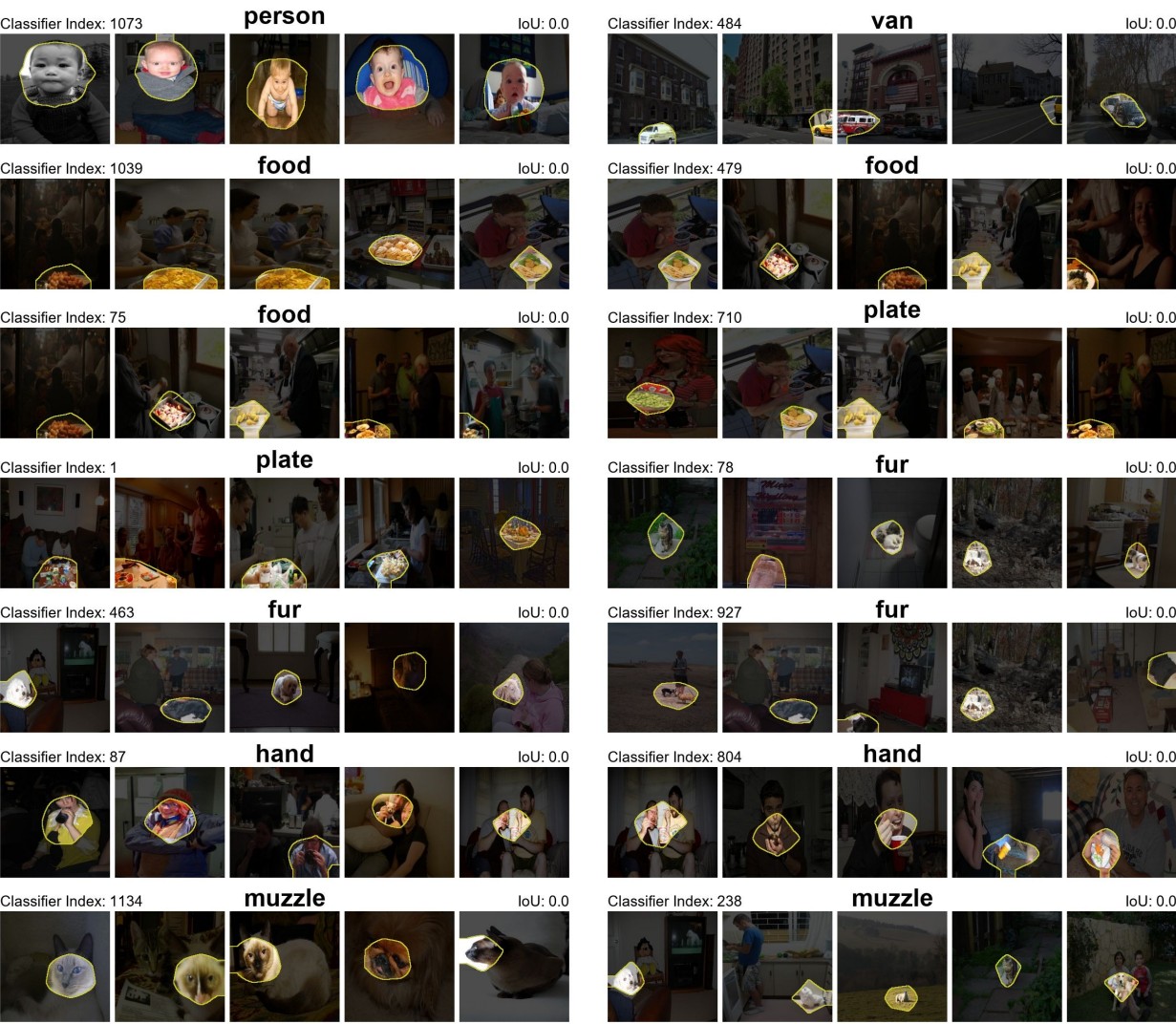

Figure 33: Qualitative segmentations using the concept detectors learned with our method. Here the network is EfficientNet trained on ImageNet, the method is using CFM, and $I = 1280$. All these concept detectors exhibit IoU scores less than 0.04 and Network Dissection does not count them as interpretable. Yet, in many cases these detectors still detect monosemantic concepts.

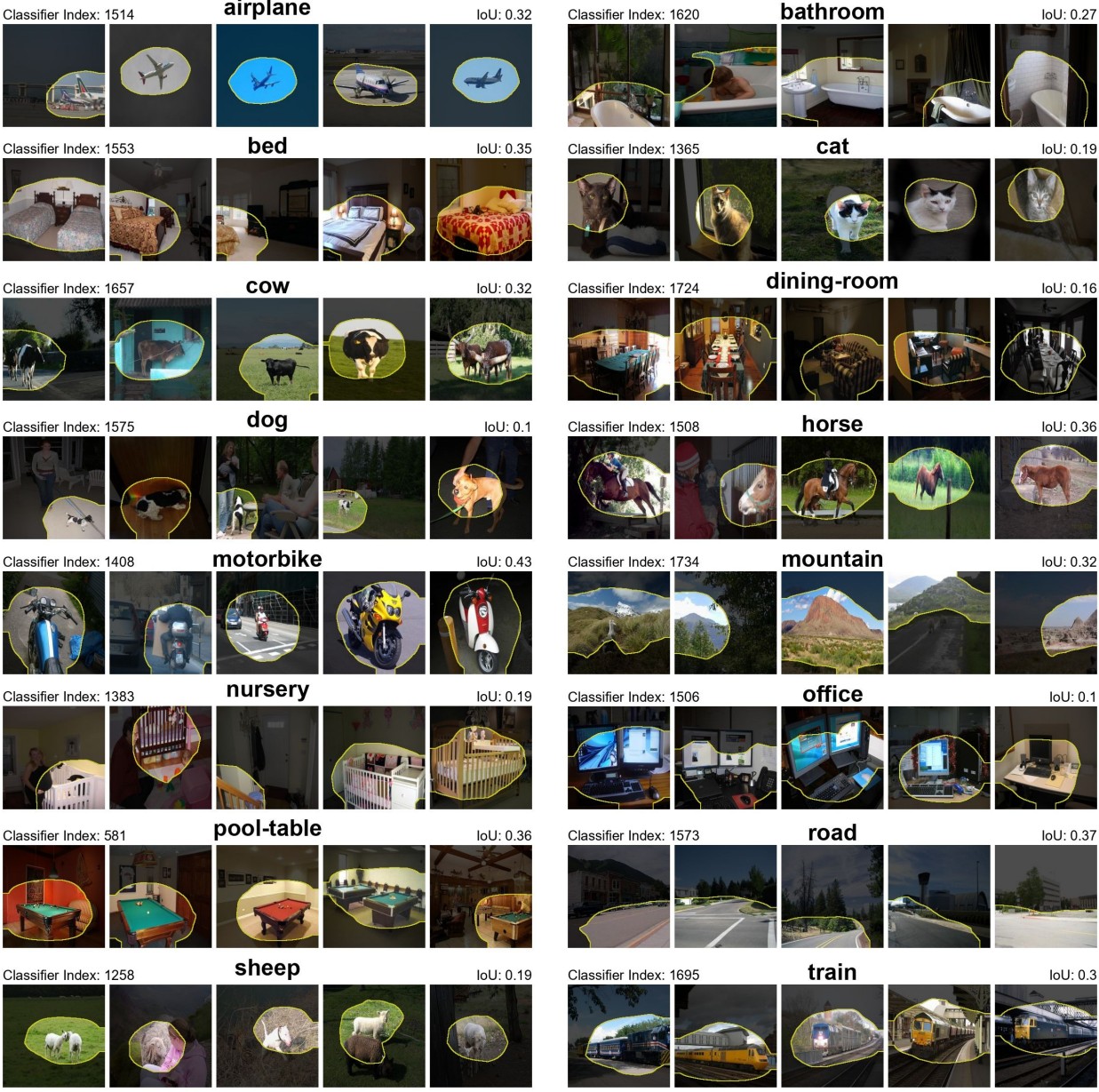

Figure 34: Qualitative segmentations using the concept detectors learned with our method. Here the network is Inception-v3 trained on ImageNet, the method is using UFM, and $I = 1792$.

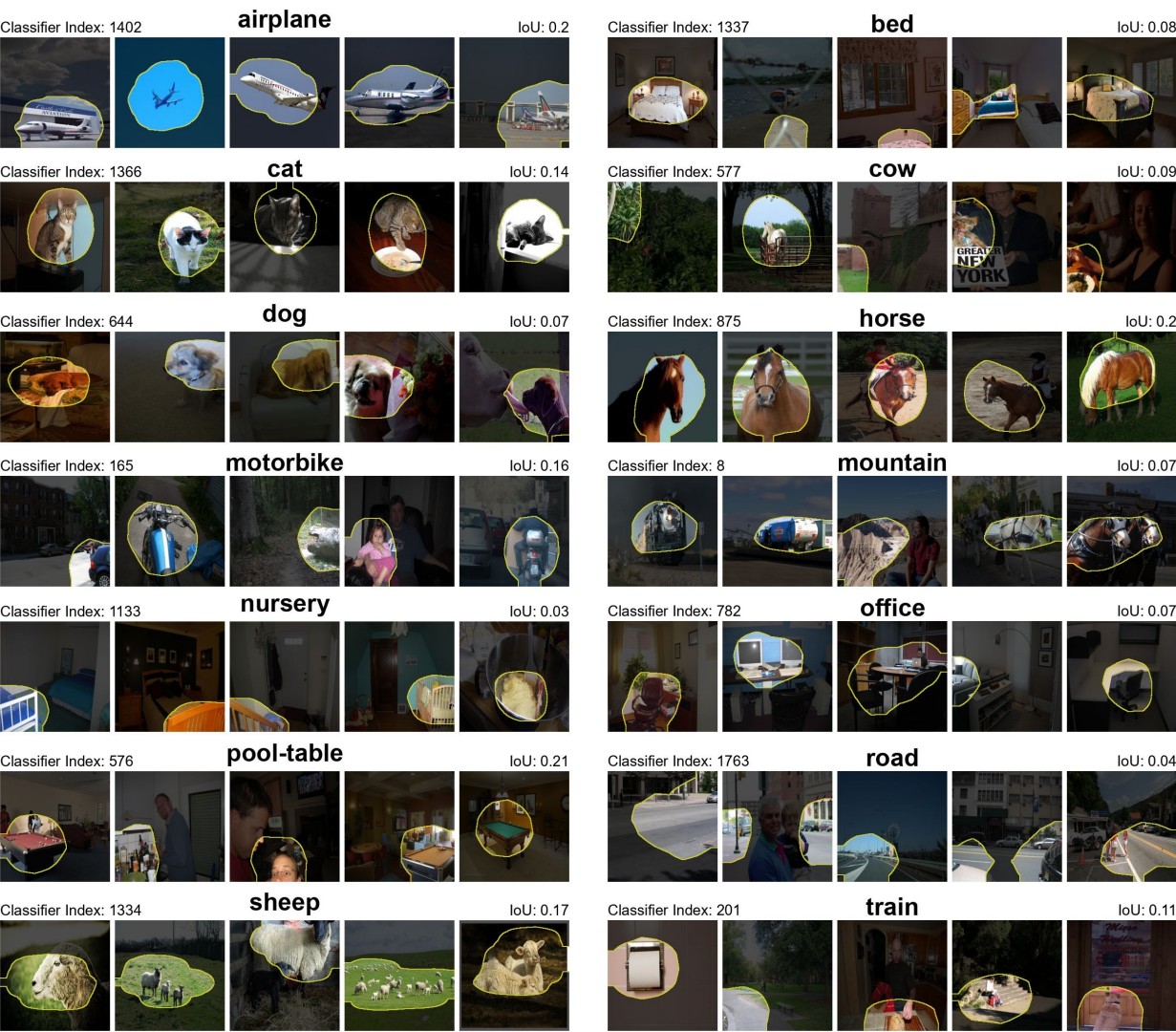

Figure 35: Qualitative segmentations using the factorization of NMF. Here the network is Inception-v3 trained on ImageNet and $I = 1792$.

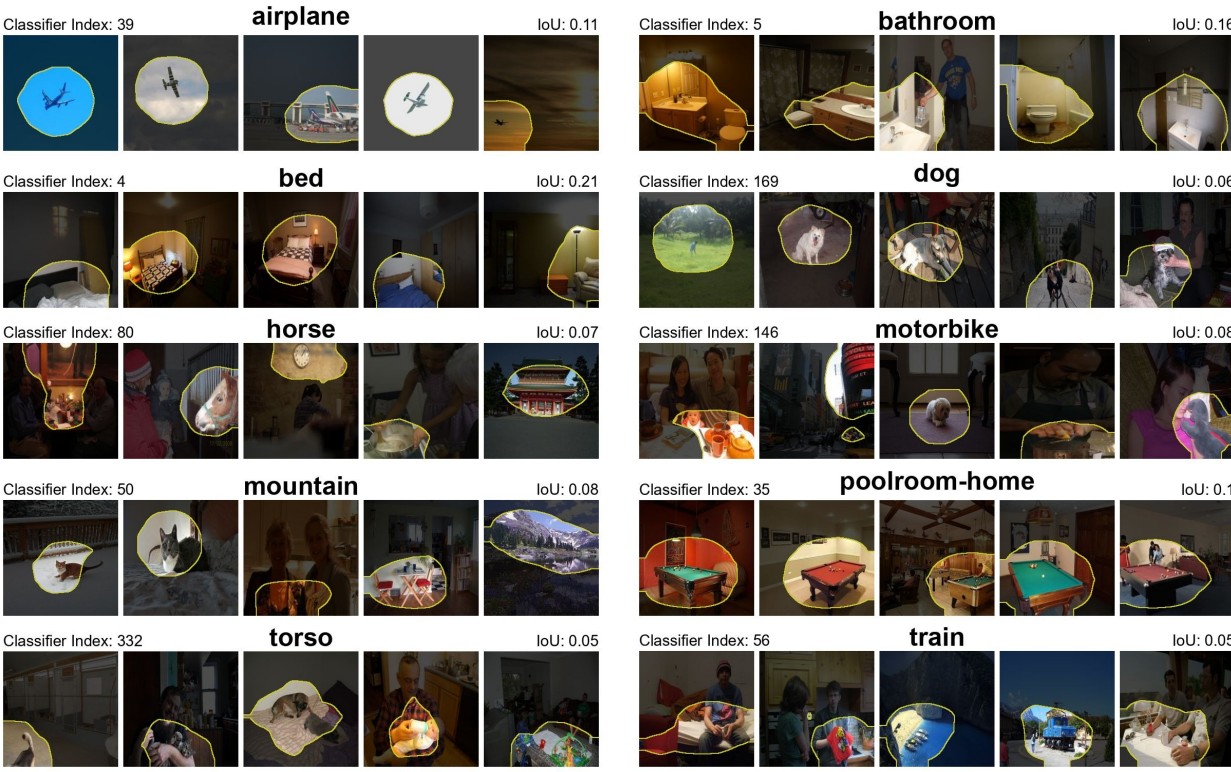

Figure 36: Qualitative segmentations using the concept detectors learned with PCA. Here the network is Inception-v3 trained on ImageNet, and $I = 1792$.

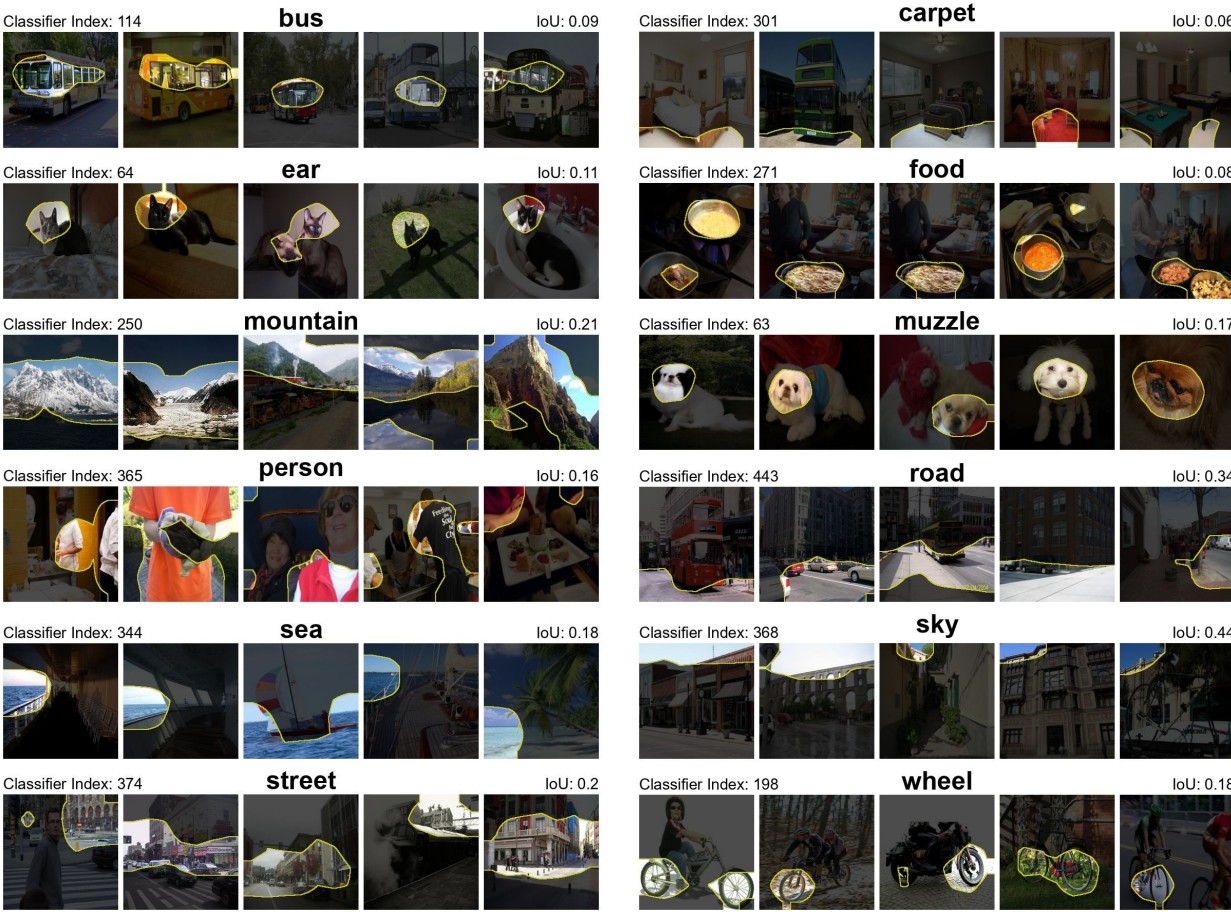

Figure 37: Qualitative segmentations using the concept detectors learned with our method. Here the network is VGG16 trained on ImageNet, the method is using UFM, and $I = 448$.

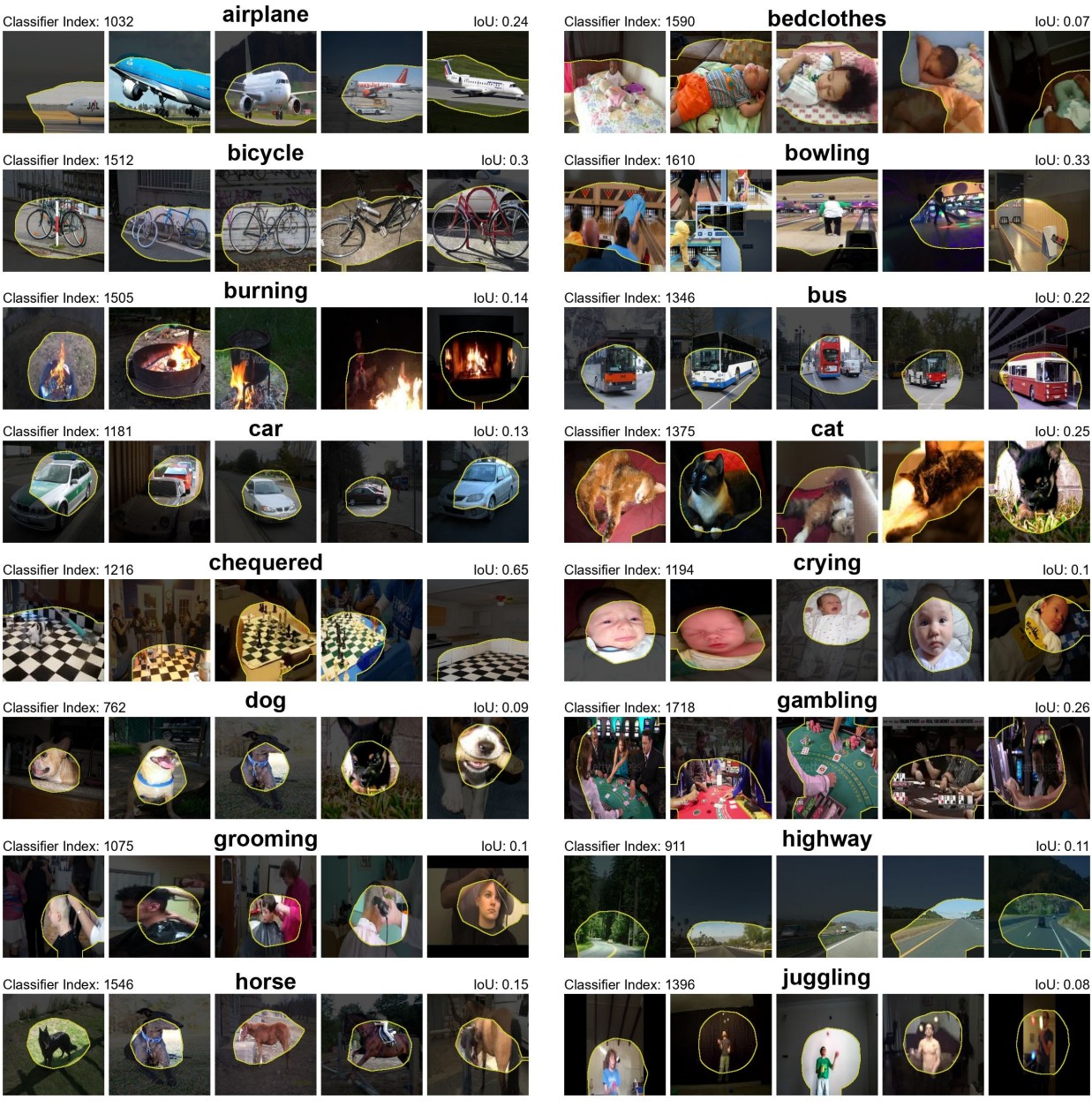

Figure 38: Qualitative segmentations using the concept detectors learned with our method. Here the network is ResNet50 trained on MiT, the method is using UFM, and $I = 1792$.

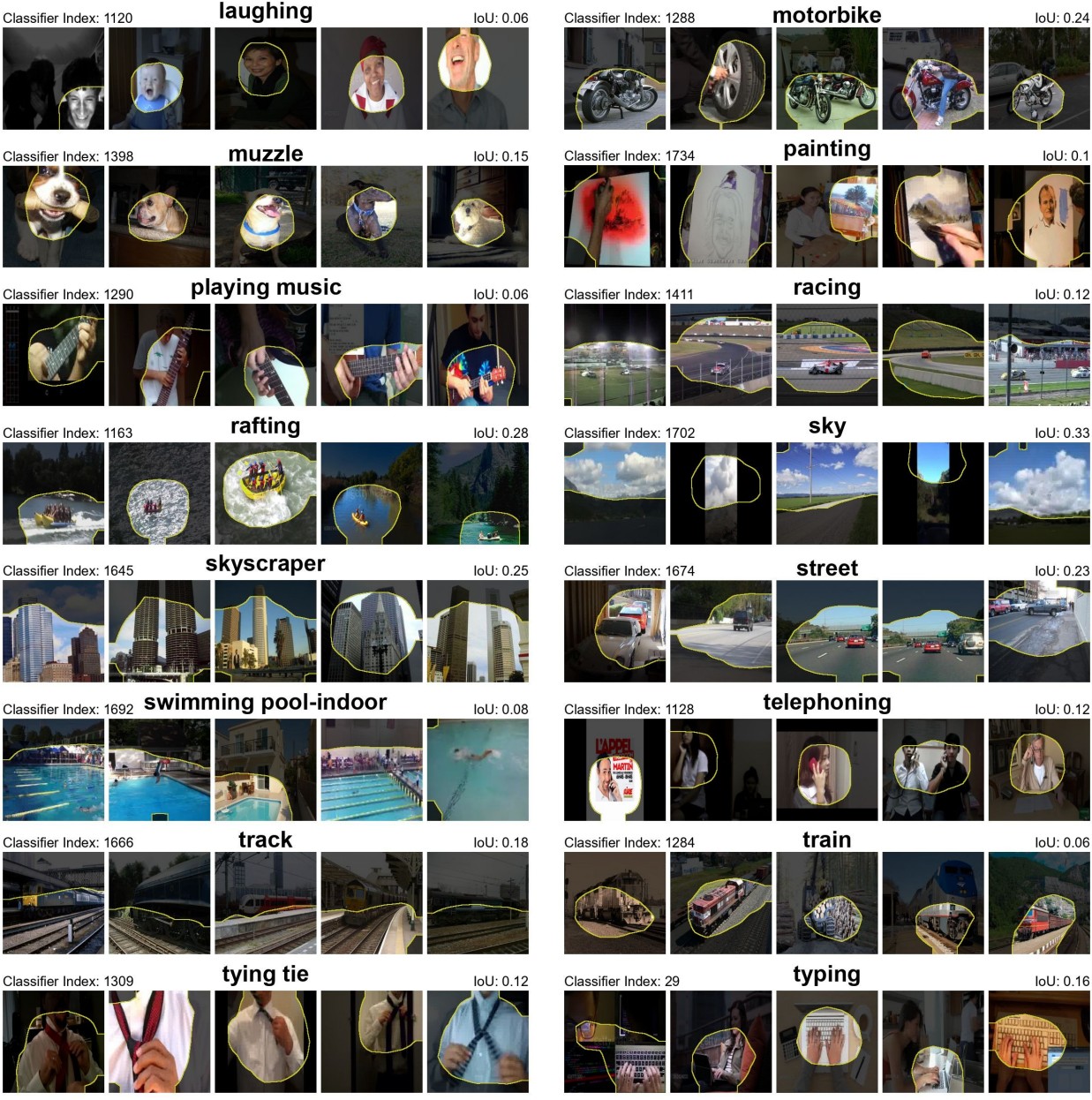

Figure 39: Qualitative segmentations using the concept detectors learned with our method. Here the network is ResNet50 trained on MiT, the method is using UFM, and $I = 1792$.

Table 27: Network Dissection statistics for EfficientNet trained on ImageNet. For each concept category in Broden, we report the two numbers: First, the number of concept detectors that were labeled with the name of a concept belonging to the category and second, the number of unique concept labels from the category that have been assigned to the set of the concept detectors.

| $I$ | Method | Color | Object | Part | Material | Scene | Texture | Total |
|---|---|---|---|---|---|---|---|---|
| | | | **EfficientNet / ImageNet** | | | | | |
| 960 | PCA | 0 / 0 | 58 / 15 | 58 / 4 | 0 / 0 | 14 / 5 | 11 / 4 | 141 / 28 |
| | EDDP-U | 0 / 0 | 134 / 51 | 14 / 9 | 4 / 4 | 34 / 26 | 21 / 17 | 207 / 107 |
| | EDDP-C | 0 / 0 | 120 / 50 | 13 / 8 | 5 / 4 | 35 / 26 | 21 / 17 | 194 / 105 |
| 1120 | PCA | 0 / 0 | 57 / 14 | 81 / 4 | 0 / 0 | 16 / 5 | 21 / 11 | 175 / 34 |
| | EDDP-U | 0 / 0 | 137 / 51 | 26 / 12 | 5 / 5 | 39 / 34 | 18 / 16 | 225 / 118 |
| | EDDP-C | 0 / 0 | 117 / 52 | 18 / 9 | 7 / 5 | 40 / 35 | 18 / 16 | 200 / 117 |
| 1280 | Natural | 0 / 0 | 155 / 29 | 28 / 11 | 1 / 1 | 29 / 20 | 10 / 9 | 223 / 70 |
| | PCA | 0 / 0 | 58 / 15 | 80 / 4 | 0 / 0 | 14 / 5 | 27 / 14 | 179 / 38 |
| | EDDP-U | 0 / 0 | 164 / 62 | 74 / 14 | 5 / 3 | 48 / 40 | 22 / 18 | 313 / 137 |
| | EDDP-C | 0 / 0 | 132 / 59 | 60 / 16 | 6 / 4 | 47 / 39 | 22 / 19 | 267 / 137 |

Table 28: Network Dissection statistics for Inception-v3 trained on ImageNet. For each concept category in Broden, we report the two numbers: First, the number of concept detectors that were labeled with the name of a concept belonging to the category and second, the number of unique concept labels from the category that have been assigned to the set of the concept detectors.

| $I$ | Method | Color | Object | Part | Material | Scene | Texture | Total |
|---|---|---|---|---|---|---|---|---|
| | | | **Inception-v3 / ImageNet** | | | | | |
| 1536 | PCA | 0 / 0 | 58 / 15 | 60 / 2 | 0 / 0 | 18 / 7 | 155 / 26 | 291 / 50 |
| | NMF | 0 / 0 | 325 / 44 | 14 / 5 | 4 / 2 | 210 / 58 | 174 / 42 | 727 / 151 |
| | EDDP-U | 0 / 0 | 763 / 73 | 16 / 11 | 8 / 7 | 111 / 81 | 55 / 36 | 953 / 208 |
| | EDDP-C | 0 / 0 | 593 / 76 | 16 / 12 | 7 / 5 | 104 / 78 | 47 / 31 | 767 / 202 |
| 1792 | PCA | 0 / 0 | 60 / 15 | 42 / 2 | 0 / 0 | 19 / 8 | 126 / 24 | 247 / 49 |
| | NMF | 0 / 0 | 340 / 47 | 14 / 6 | 2 / 1 | 207 / 64 | 137 / 38 | 700 / 156 |
| | EDDP-U | 0 / 0 | 992 / 65 | 12 / 7 | 6 / 5 | 103 / 76 | 33 / 27 | 1146 / 180 |
| | EDDP-C | 0 / 0 | 877 / 67 | 11 / 10 | 6 / 5 | 95 / 72 | 32 / 28 | 1021 / 182 |
| 2048 | Natural | 0 / 0 | 394 / 42 | 13 / 5 | 2 / 1 | 270 / 56 | 154 / 34 | 833 / 138 |
| | PCA | 0 / 0 | 57 / 15 | 58 / 2 | 0 / 0 | 19 / 8 | 153 / 25 | 287 / 5 |
| | EDDP-U | 0 / 0 | 701 / 85 | 23 / 15 | 6 / 5 | 197 / 113 | 60 / 38 | 987 / 256 |
| | EDDP-C | 0 / 0 | 488 / 75 | 22 / 13 | 7 / 4 | 263 / 113 | 73 / 38 | 853 / 243 |

Table 29: Network Dissection statistics for VGG16 trained on ImageNet. For each concept category in Broden, we report the two numbers: First, the number of concept detectors that were labeled with the name of a concept belonging to the category and second, the number of unique concept labels from the category that have been assigned to the set of the concept detectors.

| $I$ | Method | Color | Object | Part | Material | Scene | Texture | Total |
|---|---|---|---|---|---|---|---|---|
| | | | **VGG16 / ImageNet** | | | | | |
| 384 | PCA | 0 / 0 | 171 / 11 | 72 / 5 | 0 / 0 | 6 / 3 | 97 / 13 | 346 / 32 |
| | NMF | 0 / 0 | 109 / 31 | 33 / 15 | 4 / 2 | 24 / 15 | 45 / 18 | 215 / 81 |
| | EDDP-U | 1 / 1 | 51 / 36 | 203 / 13 | 1 / 1 | 4 / 3 | 20 / 19 | 280 / 73 |
| | EDDP-C | 2 / 2 | 45 / 33 | 199 / 12 | 1 / 1 | 4 / 3 | 17 / 16 | 268 / 67 |
| 448 | PCA | 0 / 0 | 211 / 11 | 82 / 5 | 0 / 0 | 5 / 3 | 113 / 15 | 411 / 34 |
| | NMF | 0 / 0 | 124 / 32 | 43 / 13 | 4 / 1 | 24 / 16 | 47 / 20 | 242 / 82 |
| | EDDP-U | 0 / 0 | 63 / 34 | 237 / 13 | 5 / 4 | 10 / 9 | 18 / 16 | 333 / 76 |
| | EDDP-C | 0 / 0 | 61 / 35 | 238 / 12 | 2 / 1 | 7 / 6 | 15 / 13 | 323 / 67 |
| 512 | Natural | 0 / 0 | 169 / 34 | 48 / 14 | 6 / 2 | 26 / 17 | 63 / 23 | 312 / 90 |
| | PCA | 0 / 0 | 263 / 11 | 84 / 5 | 0 / 0 | 6 / 4 | 117 / 18 | 470 / 38 |
| | EDDP-U | 1 / 1 | 71 / 40 | 267 / 14 | 2 / 2 | 11 / 10 | 25 / 19 | 377 / 86 |
| | EDDP-C | 1 / 1 | 68 / 40 | 260 / 14 | 2 / 2 | 10 / 9 | 26 / 19 | 367 / 85 |

Table 30: Network Dissection statistics for ResNet50 trained on Moments In Time. For each concept category in Broden, we report the two numbers: First, the number of concept detectors that were labeled with the name of a concept belonging to the category and second, the number of unique concept labels from the category that have been assigned to the set of the concept detectors.

| I | Method | Color | ResNet50 / MiT | | | | | | Total |
|---|--------|-------|--------|------|----------|--------|-------|---------|-------|
| | | | Object | Part | Material | Action | Scene | Texture | |
| | PCA | 0 / 0 | 15 / 9 | 5 / 2 | 0 / 0 | 1 / 1 | 14 / 5 | 657 / 29 | 692 / 46 |
| 1536 | NMF | 0 / 0 | 214 / 32 | 18 / 5 | 0 / 0 | 279 / 85 | 109 / 44 | 308 / 31 | 928 / 197 |
| | EDDP-U | 1 / 1 | 585 / 54 | 4 / 4 | 1 / 1 | 509 / 134 | 128 / 60 | 44 / 31 | 1272 / 285 |
| | EDDP-C | 1 / 1 | 459 / 54 | 22 / 5 | 1 / 1 | 386 / 143 | 73 / 58 | 46 / 30 | 988 / 292 |
| | PCA | 0 / 0 | 17 / 12 | 5 / 2 | 0 / 0 | 1 / 1 | 13 / 4 | 683 / 29 | 719 / 48 |
| 1792 | NMF | 0 / 0 | 199 / 35 | 23 / 5 | 1 / 1 | 468 / 111 | 99 / 46 | 144 / 22 | 934 / 220 |
| | EDDP-U | 0 / 0 | 673 / 52 | 4 / 4 | 2 / 2 | 680 / 135 | 161 / 67 | 35 / 23 | 1555 / 283 |
| | EDDP-C | 0 / 0 | 541 / 51 | 29 / 6 | 1 / 1 | 502 / 150 | 81 / 63 | 41 / 24 | 1195 / 295 |
| | Natural | 0 / 0 | 282 / 34 | 20 / 5 | 1 / 1 | 348 / 93 | 121 / 43 | 362 / 29 | 1134 / 205 |
| 2048 | PCA | 0 / 0 | 16 / 10 | 5 / 2 | 0 / 0 | 1 / 1 | 13 / 4 | 730 / 29 | 765 / 46 |
| | EDDP-U | 0 / 0 | 768 / 45 | 5 / 5 | 1 / 1 | 709 / 109 | 377 / 75 | 49 / 30 | 1909 / 265 |
| | EDDP-C | 0 / 0 | 542 / 49 | 112 / 5 | 2 / 2 | 648 / 136 | 94 / 66 | 51 / 31 | 1449 / 289 |

## A.18 More Global Model Explanations via Concept Sensitivity Testing

This Section complements Section 7.8. Figures 40, 41 and 42 depict Concept Influence Diagrams for classes of ResNet18 trained on Places365. while Figures 43 and 44 depict diagrams for ResNet50 trained on Moments in Time.

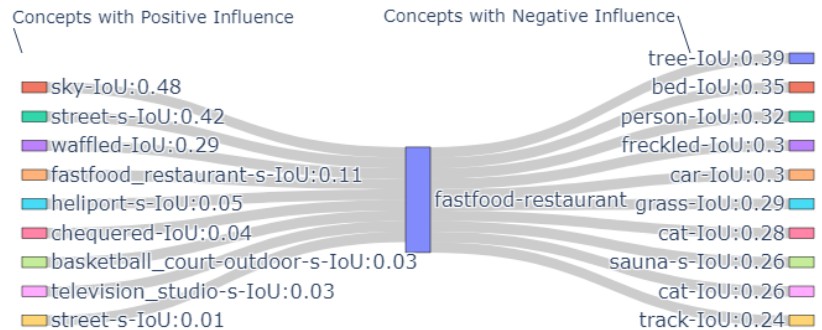

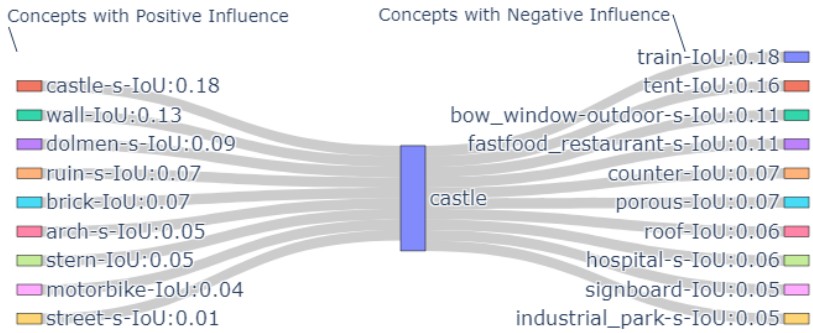

Figure 40: Concept Influence Diagram for ResNet18 trained on Places365. The model is sensitive to the depicted concepts with an absolute score above 0.99. (We use RCAV to quantify the sensitivity, and re-scale the score to $[-1, 1]$) Positive influencing and negative influencing concepts are provided. The number of concepts have been limited to 10. When concepts appear more than once, they correspond to different signal directions (as labeling the classifiers with NetDissect may assign the same concept name to more than one directions.). Here we report results for EDDP-C and $I = 512$.

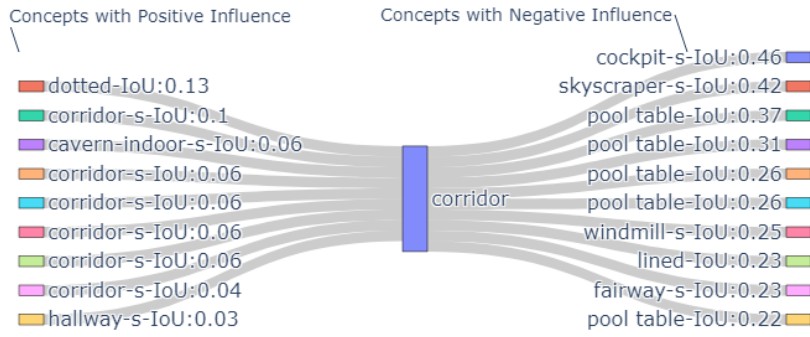

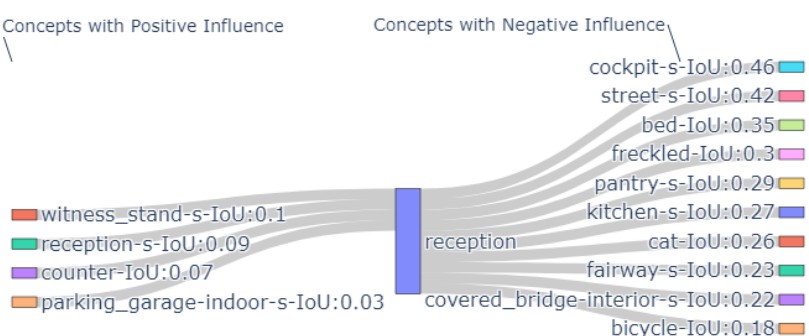

Figure 41: Concept Influence Diagram for ResNet18 trained on Places365. The model is sensitive to the depicted concepts with an absolute score above 0.99. (We use RCAV to quantify the sensitivity, and re-scale the score to $[-1, 1]$) Positive influencing and negative influencing concepts are provided. The number of concepts have been limited to 10. When concepts appear more than once, they correspond to different signal directions (as labeling the classifiers with NetDissect may assign the same concept name to more than one directions.) Here we report results for EDDP-C and $I = 512$.

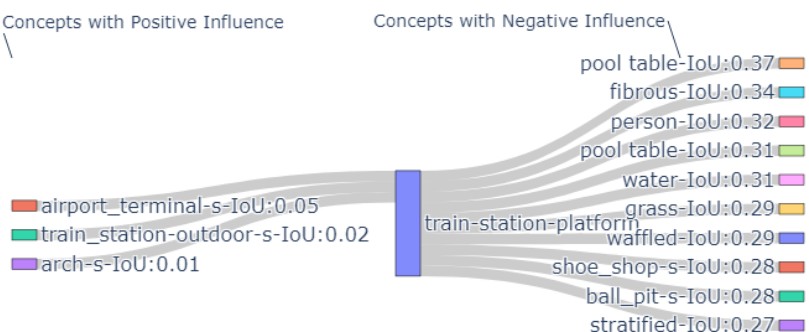

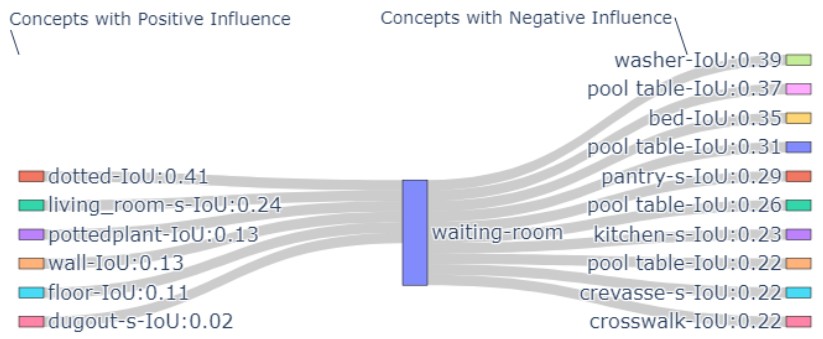

Figure 42: Concept Influence Diagram for ResNet18 trained on Places365. The model is sensitive to the depicted concepts with an absolute score above 0.99. (We use RCAV to quantify the sensitivity, and re-scale the score to $[-1, 1]$) Positive influencing and negative influencing concepts are provided. The number of concepts have been limited to 10. When concepts appear more than once, they correspond to different signal directions (as labeling the classifiers with NetDissect may assign the same concept name to more than one directions.) Here we report results for EDDP-C and $I = 512$.

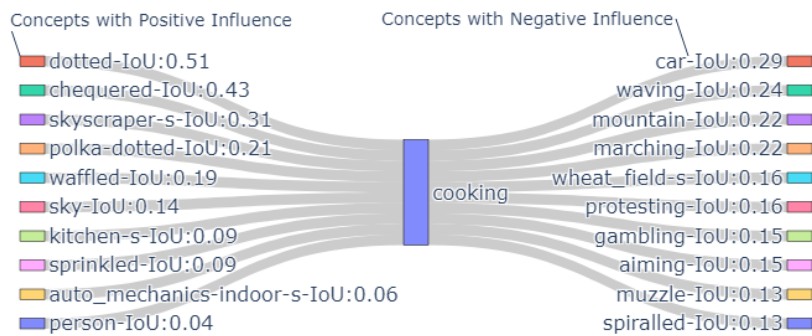

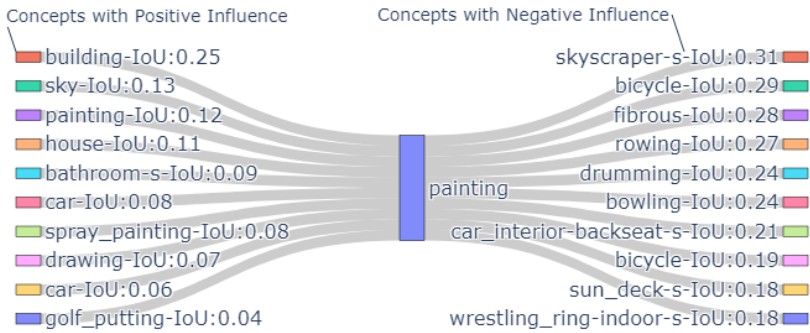

Figure 43: Concept Influence Diagram for ResNet50 trained on Moments in Time (MiT). The model is sensitive to the depicted concepts with an absolute score above 0.99. (We use RCAV to quantify the sensitivity, and re-scale the score to $[-1, 1]$) Positive influencing and negative influencing concepts are provided. The number of concepts have been limited to 10. When concepts appear more than once, they correspond to different signal directions (as labeling the classifiers with NetDissect may assign the same concept name to more than one directions.) Here we report results for EDDP-C and $I = 2048$.

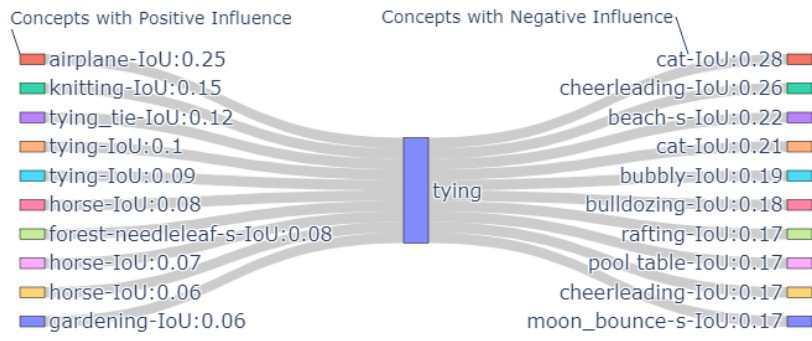

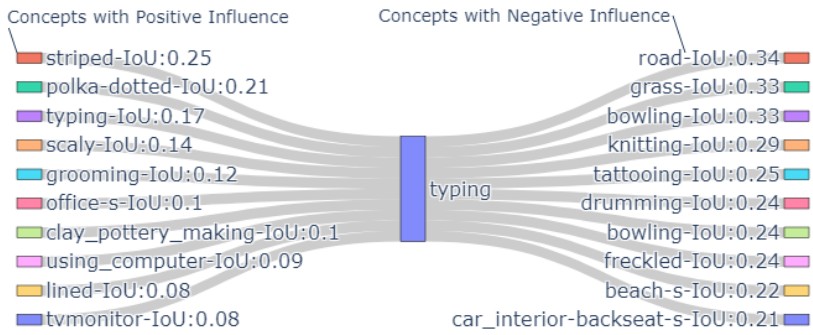

Figure 44: Concept Influence Diagram for ResNet50 trained on Moments in Time (MiT). The model is sensitive to the depicted concepts with an absolute score above 0.99. (We use RCAV to quantify the sensitivity, and re-scale the score to $[-1, 1]$) Positive influencing and negative influencing concepts are provided. The number of concepts have been limited to 10. When concepts appear more than once, they correspond to different signal directions (as labeling the classifiers with NetDissect may assign the same concept name to more than one directions.) Here we report results for EDDP-C and $I = 2048$.

### A.19 More Local Explanations with Concept Contribution Maps

This Section complements Section 7.9. Figures 45, 46 depict CCMs for the prediction of an image belonging to class *beach-house*. The respective image concept contribution scores are depicted in Figures 47 and 48. Figures 49, 50 depict CCMs for the prediction of an image belonging to class *bedchamber*. The respective image concept contribution scores are depicted in Figures 51 and 52.

### A.20 Details on Counterfactual Explanation Example

For intervention, we make use of (33). To compute the target projected values $t$, we use the training split of Broden. When we perform concept removal (or addition), we choose $t$ to be equal to the lowest (or topmost) 0.005% quantile of projected feature activations on the filter $\boldsymbol{w}_i$. This process ensures that, after the intervention, the signal value in the patch embedding for the concept matches the signal value of samples with or without the concept.

### A.21 Details on Toy Model Correction

**Network Architecture** After each convolutional layer, except the last, there is a `Dropout` layer with $p = 0.3$. All `Conv2d` layers have a kernel size of 3x3 and a stride of 2, except for the last one, which has a stride of 1. Furthermore, the latent space dimensionality is set to 16 for all convolutional units.

**Chess Dataset** The total number of images in the dataset is 210, 67 for *bishop*, 71 for *knight* and 72 for *rook*. The spatial resolution of those images is 85x85. We make a stratified train-test split with the training set ratio set to 0.7.

**Network Training** We train the network with cross-entropy loss and the Adam optimizer with a learning rate of 0.005 for 1000 epochs. In the (poisoned) training set, the model achieves 100% accuracy.

**Direction Learning** To learn the latent directions, we apply our method by following the learning process described in section A.7. Furthermore, for stable learning, we learn the directions with the Augmented Lagrangian loss scheme of Section 4.5. We optimize $\lambda^{fs}\mathcal{L}^{fs} + \lambda^{cur}\mathcal{L}^{cur}$ with the target constraints $\tau^{ma} = 0.8, \tau^{ic} = 0.01, \tau^{eac} = 0.01, \tau^{mm} = 8.0, \tau^{fso} = 0.1$, and weights $\lambda^{fs} = 2.6$ and $\lambda_{cur} = 0.25$.

### A.22 Details on Computational Resources

We utilized dedicated computing rigs for our experiments. Experiments for ResNet18 and EfficientNet were run on single-GPU configurations: one NVIDIA A10G (24 GB) and one NVIDIA RTX 4090 (24 GB), respectively. For InceptionV3, VGG16, and ResNet50, we employed a compute rig equipped with four NVIDIA A10G GPUs (4 × 24 GB). The runtime for each of the key processing steps (a), (b), and (d) detailed in Section A.7 varied by architecture: ResNet18 completed in under 2 hours; VGG16 in under 1 hour; ResNet50 in under 10 hours; InceptionV3 in under 13 hours; and EfficientNet in under 5 hours.

### A.23 Relation to Sparse Auto-Encoders

The superposition hypothesis Elhage et al. (2022) assumes that neural networks linearly represent more features than there are neurons in their hidden layers. In this line of work, a feature is defined to be an abstract property of the input, which may or may not align with human intuition, and is exploited by the network to make predictions. Sparse AutoEncoders (SAEs) Bricken et al. (2023) have been proposed as a tool to take features out of superposition; that is, given the activations of a network's intermediate layer, SAEs try to linearly decompose these activations in terms of latent feature components under a sparsity objective.

Let $\boldsymbol{x} \in \mathbb{R}^D$ denote the activation of a network in a hidden layer of study and $I \gg D$ denote the number of latent features that the network is assumed to represent. While several SAE variants have been proposed, such as Lim et al. (2024); Cunningham et al. (2024); Sharkey et al. (2022); Bussmann et al. (2025), in their

baseline form Bricken et al. (2023) they first extract latent feature components $\boldsymbol{v}$ by:

$$\boldsymbol{v} = \text{ReLU}(\boldsymbol{W}_{\text{enc}}^T(\boldsymbol{x} - \boldsymbol{b}_{\text{dec}}) + \boldsymbol{b}_{\text{enc}}) \tag{44}$$

and subsequently they aim to reconstruct the original activation by

$$\hat{\boldsymbol{x}} = \boldsymbol{W}_{\text{dec}}\boldsymbol{v} + \boldsymbol{b}_{\text{dec}} \tag{45}$$

with $\boldsymbol{W}_{\text{enc}} \in \mathbb{R}^{D \times I}, \boldsymbol{W}_{\text{dec}} \in \mathbb{R}^{D \times I}, \boldsymbol{b}_{\text{enc}} \in \mathbb{R}^I$ and $\boldsymbol{b}_{\text{dec}} \in \mathbb{R}^D$. The objective that drives SAE learning linearly combines an L2 reconstruction loss between $\boldsymbol{x}$ and $\hat{\boldsymbol{x}}$ and an penalization in the L1 norm of $\boldsymbol{v}$.

To relate SAEs with our approach, we first need to make a correspondence in terms of terminology. First, a SAE *feature*, which is an abstract property of the input, in our terminology is referred to as a *concept*. The *number of latent features* is related to our *concept cluster count*, and we refer to both of them as $I$. The **encoder** part of the AutoEncoder which is realized by the matrix $\boldsymbol{W}_{\text{enc}}$ is related to our **decoding** directions $\boldsymbol{W}$, while the AutoEncoder's **decoder** part $\boldsymbol{W}_{\text{dec}}$ is equivalent to our **encoding** directions (signal vectors) $\hat{\boldsymbol{S}}$. The SAE's $\boldsymbol{b}_{\text{dec}}$ is related to our multi-concept signal-distractor data model's latent space bias $\boldsymbol{c}$. Finally, a **feature latent $\boldsymbol{v}$** is somewhat related to the definition of our **signal values**. In particular, under our proposed multi-concept signal-distractor data model, $\boldsymbol{W}_{\text{enc}}^T(\boldsymbol{x} - \boldsymbol{b}_{\text{dec}})$ may be considered as a mechanism to extract signal values.

Despite the conceptual similarities between our method and SAEs, we also have several differences. First, (44) is conceptually acting as a classifier with positive predictions whenever the extracted signal value exceeds the threshold in $\boldsymbol{b}_{\text{enc}}$. From one perspective, the same equation acts like centering the signal value around the classification threshold for the presence of the concept in the representation. In SAEs, sparsity is enforced in the units of the latent features $\boldsymbol{v}$, while in our approach, the ReLU activation function is replaced with a sigmoid, and thus sparsity is enforced in the semantic space of concepts. Second, in SAEs, $\boldsymbol{W}_{\text{dec}}$ are learned via a reconstruction objective, while in our approach, signal vectors are learned under the properties of a probabilistic model constrained on the extracted signal values and the feature activations themselves. Third, we also consider constraining each decoding direction to be perpendicular to the signal vectors of other concepts for exact signal value extraction. Finally, we also consider uncertainty region alignment, which additionally exploits the use of the directions by the model.

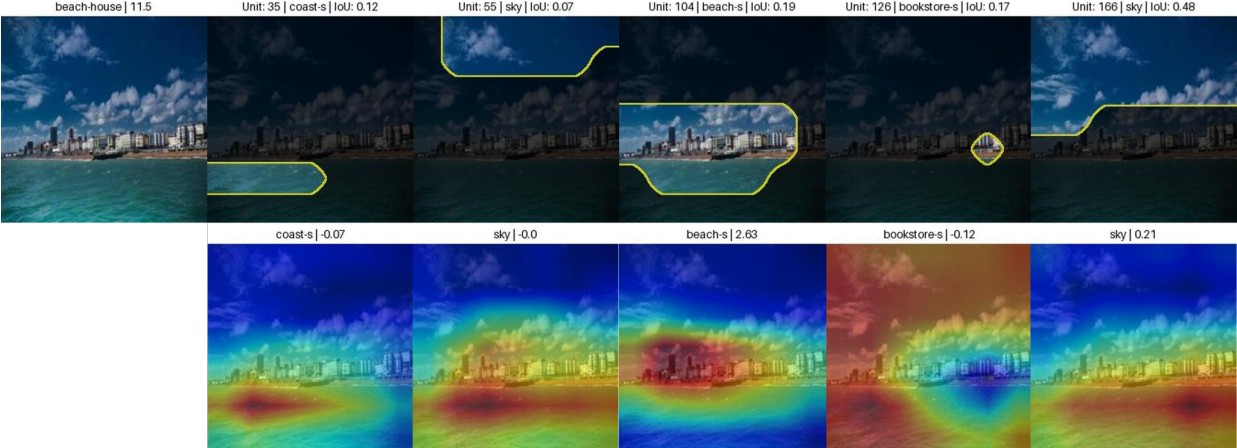

Figure 45: **Left:** Original image. The caption contains class prediction and output class logit. **Top Row:** Segmentation Maps obtained by the concept detectors. The caption contains classifier index (unit), concept-name and IoU score in the validation split of the dataset. **Bottom Row:** Concept Contribution Maps. The caption contains concept-name and contribution of the concept to the class logit.

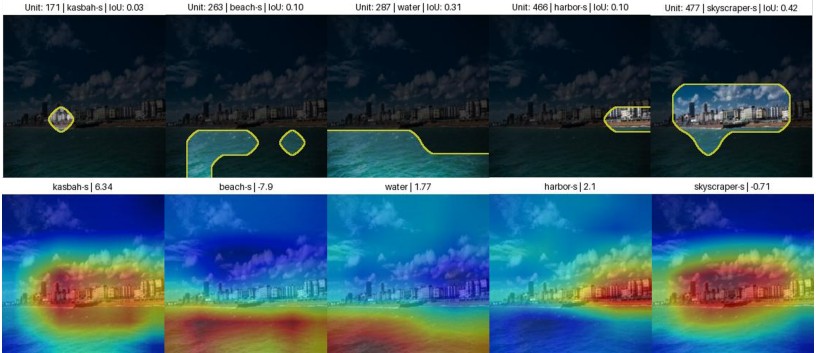

Figure 46: **Top Row:** Segmentation Maps obtained by the concept detectors. The caption contains classifier index (unit), concept-name and IoU score in the validation split of the dataset. **Bottom Row:** Concept Contribution Maps. The caption contains concept-name and contribution of the concept to the class logit.

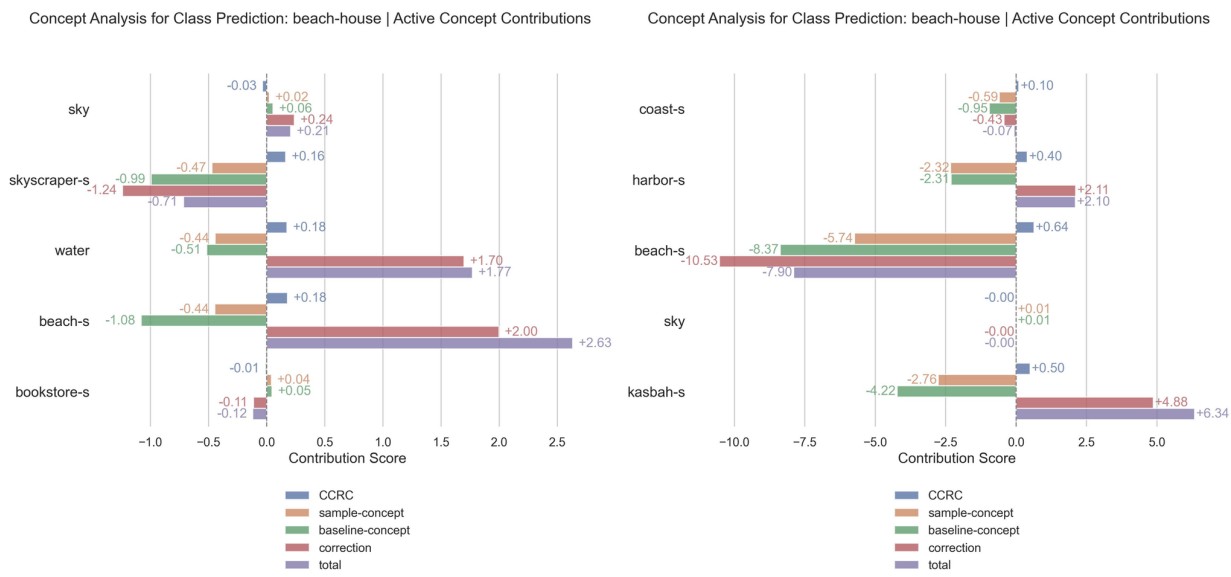

Figure 47: Concept Analysis for predicting an image of the *beach-house* class. The figure depicts concepts found in the image. Even though concepts may share the same name, they correspond to different direction pairs.

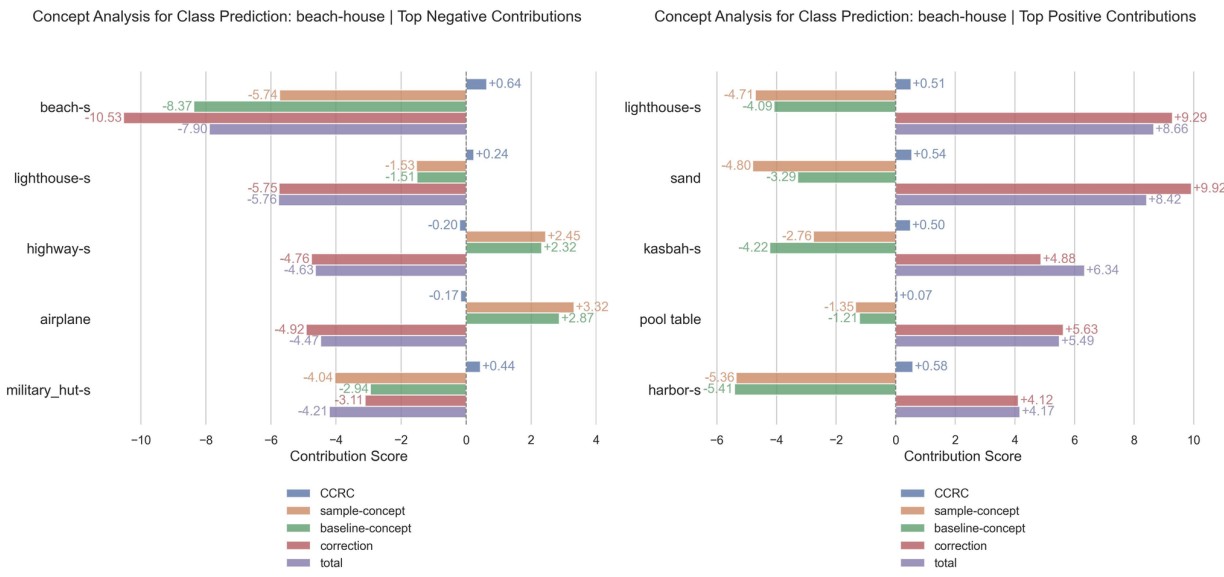

Figure 48: Concept Analysis for predicting an image of the *beach-house* class. The figure depicts top positive and top negative contributing concepts. Even though concepts may share the same name, they correspond to different direction pairs.

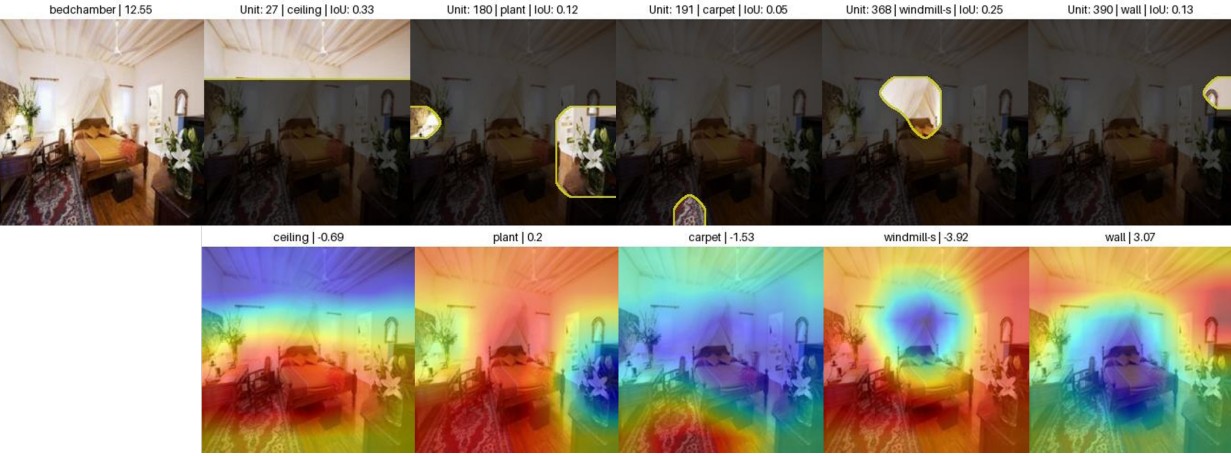

Figure 49: **Left:** Original image. The caption contains class prediction and output class logit. **Top Row:** Segmentation Maps obtained by the Concept Detectors. The caption contains classifier index (unit), concept-name and IoU score in the validation split of the dataset. **Bottom Row:** Concept Contribution Maps. The caption contains concept-name and contribution of the concept to the class logit.

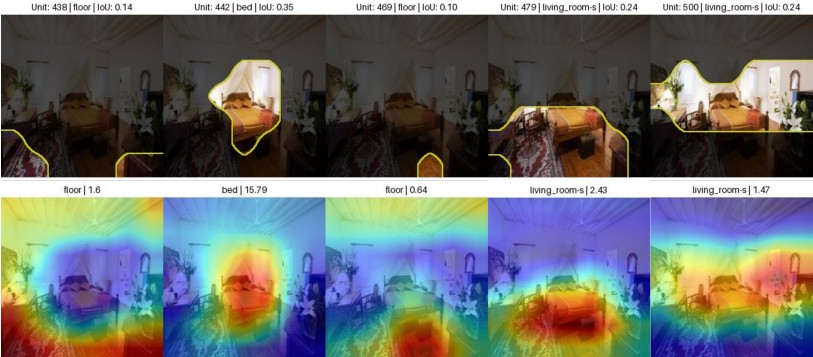

Figure 50: **Top Row:** Segmentation Maps obtained by the concept detectors. The caption contains classifier index (unit), concept-name and IoU score in the validation split of the dataset. **Bottom Row:** Concept Contribution Maps. The caption contains concept-name and contribution of the concept to the class logit.

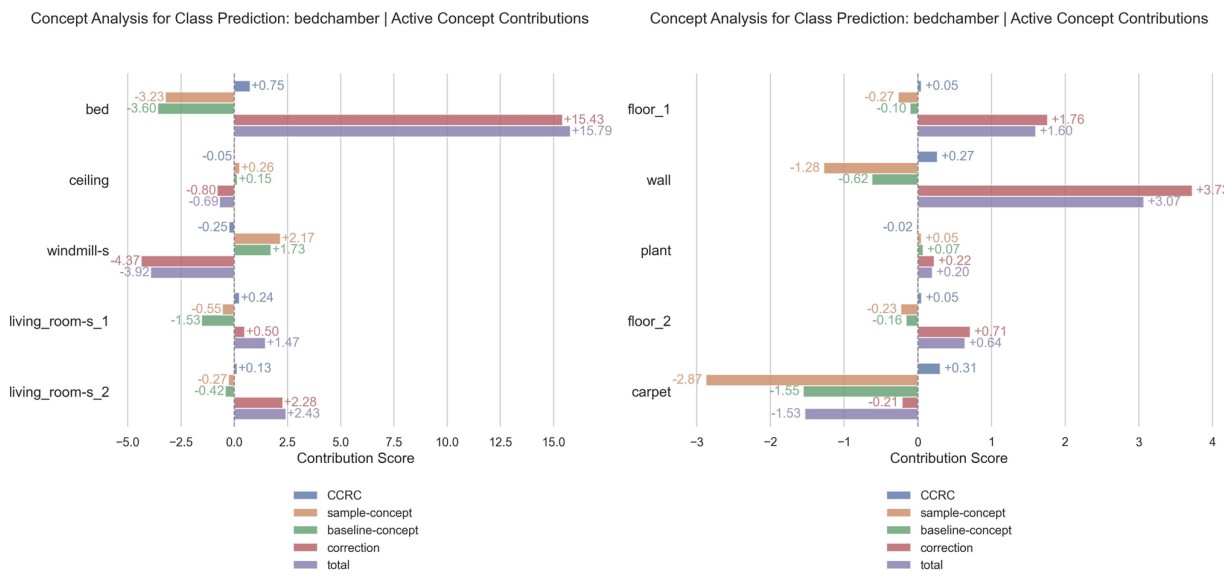

Figure 51: Concept Analysis for predicting an image of the *bedchamber* class. The figure depicts concepts found in the image. Even though concepts may share the same name, they correspond to different direction pairs.

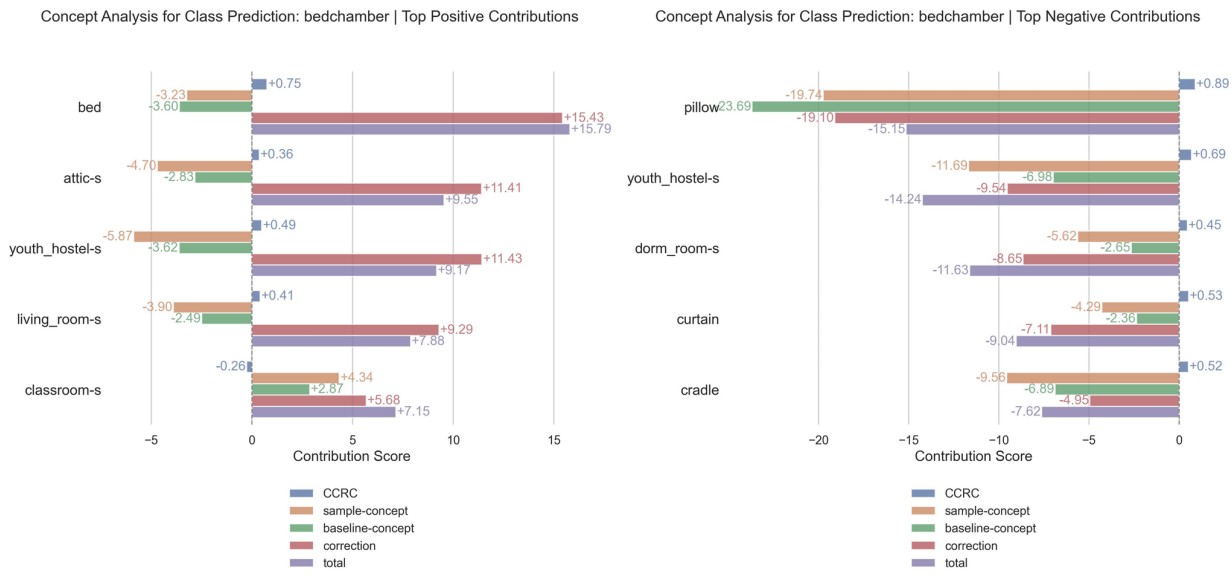

Figure 52: Concept Analysis for predicting an image of the *bedchamber* class. The figure depicts top positive and top negative contributing concepts. Even though concepts may share the same name, they correspond to different direction pairs.

