# OpenReview forum: "Learning Encoding-Decoding Direction Pairs to Unveil Concepts of Influence in Deep Vision Networks"
_TMLR — Accepted by TMLR_

### Review · Reviewer_twFM · 2025-11-23

**Summary Of Contributions:**

### Summary
- The authors introduce Encoding-Decoding Direction Pairs (EDDP), a novel framework designed to discover encoded concept signal directions within DNNs in an unsupervised manner.
- They propose a method for jointly learning decoding and encoding directions through the use of directional clustering. The approach leverages directional clustering to identify prominent directions in the latent representation of the data. Their methodology is grounded in works (Haufe et al, Kindermanns et al, Pahde et al) addressing the decomposition of signal and distractor directions.
- They present a variety of novel loss functions aimed at enhancing the interpretability and influence of identified concept directions, including the excessively active classifier loss, the focal sparsity loss, the filter-signal orthogonality loss, and a loss term that encourages uncertainty alignment.
- The concepts identified through EDDP are optimized to be both interpretable—aligned with human intuition—and influential, consistently affecting model predictions.


### Strengths
- The authors tackle a significant and relevant research question in the AI interpretability community, specifically the unsupervised detection of interpretable and influential concept directions.
- They introduce a range of novel, creative, and sensible loss terms that address various aspects of direction discovery, optimizing the interpretability and influence of found directions.
- The manuscript includes a comprehensive set of experiments, covering diverse applications and ablation studies that support the claims made in the paper.

### Weaknesses
- The authors focus solely on ConvNets. The work would be more impactful if it also addresses transformer-based architectures, as mentioned in Section 8.
- The apparent tradeoff between interpretability and influence is not sufficiently addressed. On page 9, the authors state that "uncertainty region alignment assists in improving interpretability or in aligning signal vectors with directions of interest." However, they later claim that interpretability and influence are negatively correlated (Section 7.5.6). A more detailed discussion would be valuable, including hypotheses or intuition for reasons for the tradeoff. Are there any thoughts on how more influential directions can be identified?
- The authors rely on the relatively outdated approach of Network Dissection, despite the existence of more recent methods like ClipDissect (Oikarinen et al., 2022). ClipDissect overcomes the need for a densely labeled dataset (Broden). This could be particularly relevant for specific applications, such as in the medical field.
- A tabular overview of all loss terms, their variations, and their respective roles would highly improve the readability of the paper. Although Figure 4 attempts to provide this information, it remains somewhat unclear.
- In Section 5.1 (*Intervening*), the concept of activation steering as a model intervention is not novel, and existing works (e.g., Anders et al. 2022, Neuhaus et al. 2023) should be cited.
- In Section 7.10 (*Concept-level intervention*), it would be helpful to clarify how each step impacts predictions, specifically the results (1) after concept removal and then (2) after concept insertion.
- In Section 7.11 (*Model correction*), how would the concept detector focused on the intended (bias) concept be identified in practice? In Figure 22, the bias is apparent from the example images, but this may not be straightforward in practical applications.
- What is the rationale for reporting the accuracy on the combined test set (clean & poisoned). Is this not just the average of the individual test sets?

### Questions
- Could the directional clustering approach be extended to semi-supervised concept discovery? For instance, if concept labels for one specific concept are known, could the cluster assignment for that set of samples be fixed while training the remainder?
- The authors mention the use of constrained optimization with Lagrangians to avoid tuning loss weights. Is the tuning of sensible target values $\tau$ not a similar challenge?
- Why does PCA perform so well across all experiments in Section 7.5 in terms of influence? Do the authors have any intuition for this observation?

**Additional Comments:**

Overall, I think the manuscript is too long and can be shortened to improve the accessibility of the paper. This could be achieved by a reduction of redundancy and by moving some experimental discussions (e.g., from Sec. 7.5) to the appendix.

**Audience:**

Yes

**Audience Explanation:**

Yes, the scalable (unsupervised) detection of concepts that (1) align with human mental representations (interpretability), (2) are influential on model decisions (influence), and (3) are useful for downstream tasks such as interventions, is an important research field in the interpretabiltiy community. The paper makes valuable contributions to the area.

**Broader Impact Concerns:**

No concerns. Improving the interpretability of DNNs has a positive impact on the AI community, as it fosters the verifiability and safety of AI models before deployment.

**Claims And Evidence:**

Yes

**Claims Explanation:**

(mostly)

- The authors conduct an extensive set of experiments to address each individual claim and component of their approach. This includes a simple toy experiment as well as a variety of applications that measure the faithfulness, interpretability, and influence of the identified concept directions. They further assess the applicability of their method to downstream tasks, covering both local and global explanations, as well as model correction and concept-level intervention. Additionally, the authors perform ablation studies to evaluate the impact of individual loss components, which is crucial given the compositional nature of their approach.
- Some metrics presented are not entirely convincing. In Section 7.2, the authors utilize deep dream to amplify inputs that trigger specific concept representations and fit what they call “dreaming directions”, which are then compared with the signal vectors. This metric appears to be error-prone due to the nature of the dreaming process, and the interpretation of results (sufficient faithfulness because scores are above 0.7) seems somewhat arbitrary. Moreover, the interpretation of the “influence” metric is unclear, particularly in the absence of ground truth information regarding the influence of specific concepts. If sensible concepts are not used by the model (because it employs other non-intuitive concepts), should we achieve high influence scores?

**Requested Changes:**

1. In my opinion, the authors should consider replacing the Network Dissection approach, which relies on the densely annotated Broden dataset for concept labeling, with the more recent CLIP Dissect method. This change would enable the adaptation of the framework to more specific use cases involving concepts that are not represented in the Broden dataset.
2. It would enhance the readability of the paper if the authors provided a tabular overview of all loss functions, including their variations and individual roles. This addition would improve the understanding of the contributions made by each loss term.

---

> ### Author Response · Authors · 2025-12-15
>
> We appreciate the thorough review and insightful comments.
> Below are some point-by-point responses:
>
> **RC1: Use CLIP-Dissect instead of NetDissect**
>
> We appreciate this suggestion but respectfully maintain the use of NetDissect in our evaluation pipeline, while clarifying that it is **not part of our core EDDP method**.
>
> 1. **NetDissect is used for Evaluation:** NetDissect serves strictly as an objective, widely accepted tool for **quantifying interpretability**. This **does not interfere** with the core of our method.
> 2. **CLIP-Dissect Lacks Focus:** Our evaluation heavily relies on quantifying the segmentation performance of concept detectors. CLIP-Dissect primarily lacks a mechanism to focus on specific image regions and often tends to assign more generic image labels (acknowledged in the respective paper)
> 3. **We Already use CLIP for Monosemanticity:** We address the limitations of relying on a finite annotated dataset by incorporating a CLIP-based metric for **monosemanticity** (Section 7.4.2). This metric, inspired by existing literature, crops the image region with the concept and directly leverages CLIP's embedding space to quantify concept coherence within clusters. Even though not perfect, cropping overcomes the focus problem described in (2), is simple, and has been previously used in a related work.
> 4. **CLIP-Dissect is Better Harnessed with Larger Un-Annotated Datasets**: The main point to move to CLIP-Dissect is the lack of annotations. When using Broden, NetDissect is probably more suitable for providing concept labels than CLIP-Dissect, since **labels are available**. Moving to a larger and un-annotated dataset may have advantages, but would also induce a larger computational cost to the subsequent processing steps, which we considered infeasible for our available resources.
> 5. **For the Studied Networks, Broden Should be Sufficient**: Following (4), the true dilemma is not whether CLIP-Dissect is better than NetDissect, but instead whether Broden is a sufficient concept dataset for our evaluations. We assert that Broden is. The large Broden and Broden-Action concept datasets, coverring up to 1,400 visual concepts with 86K images, are highly appropriate, as their domain closely matches the domain of the ConvNets studied.
>
> **RC2: Tabular overview of loss functions**
>
> We thank you for the idea and have included this in the revised version as Table 1
>
> **C1: Some Faithfulness metrics not convincing**
>
> We appreciate this constructive critique regarding the validation of direction pair fidelity.
> 1. **Dreaming:** The dreaming approach is a novel technique intended to address a critical gap: validating encoding/decoding fidelity in **real-world data** without resorting to synthetic or highly controlled training environments. We took several measures to mitigate error: a) we intitialize the pre-image with an image depicting the concept, b) we only do a few number of iterations c) we exclude zooming, mitigating distortions, d) we constrain the dreaming process to happen inside the masked region of the concept. In Figs. 9 & 10 we provide example images of the dreaming process, which show that the image content is not significantly altered during dreaming
> 2. **Fidelity Threshold:** We agree that the 0.7 threshold may appear arbitrary. We clarify that this choice is guided by existing literature: related supervised work (Pahde et al., 2024) found that the fidelity (in terms of cosine similarity) of supervised concept vectors, in experiments under **controlled** and **synthetic** settings, often falls within the range of 0.6–0.8. Our selected 0.7 threshold is thus grounded in the current state-of-the-art for validating concept directions
>
> **C2: Influence metric interpretation**
>
> In quantifying influence, there is no need for external ground-truth. Based on the definition of the concept (EDDP clusters), the sensitivity of the network to the concept **is what it is**, and the metric of influence (Eq. 32) is just an aggregation of concept sensitivity over concepts and output classes.
>
> **C3: Constrained optimization**
>
> We thank the reviewer for highlighting the nuance of lagrangians. We have revised Section 4.5 to clarify this rationale
>
> **C4: Interpretability-Influence tradeoff**
>
> Page 9 refers to ablating Uncertainty Region Alignment, while Section 7.5.6 compares methods
>
> **C5: Activation steering is not novel**
>
> We did cite Anders et al. in the Introduction. Section 5.1, is novel **under the lens of encoding-decoding direction pairs**, as most works do not make this distinction. In the revised version, we cite Neuhaus et al. as suggested
>
> **C6: Section 7.10 Concept-level intervention**
>
> We have updated Section 7.10 to provide the network predictions and confidence scores after each step
>
> **C7: ConvNets**
>
> We respectfully acknowledge this limitation in Section 8
>
> **C8: Semi-supervised learning**
>
> Yes
>
> **C9: Detecting bias**
>
> With human-in-the-loop. Our EDDP serves as central tool in the pipeline

---

### Review · Reviewer_PAQt · 2025-11-23

**Summary Of Contributions:**

The authors propose a novel unsupervised approach for recovering semantic concepts as *encoding–decoding direction pairs* in latent space. These directions can quantify concept presence.
They introduce Uncertainty Region Alignment, a method that leverages information in the model’s weights to guide the search for interpretable, prediction-relevant directions.
They include empirical validation by demonstrating accurate recovery of ground-truth direction pairs in controlled synthetic experiments. Further, they show that the learned decoding directions correspond to monosemantic, interpretable concepts and outperform unsupervised baselines on interpretability metrics.
Finally, they provide practical examples of applications illustrating how the learned directions can be used.

Strengths:

- The paper presents a method in the current topic of model interpretability. The method has some advantages over the popular methods such as SAEs and CAVs.
- The authors perform several experiments for empirical validation on real and synthetic data and they demonstrate convincing application use cases for the method.

Weaknesses:

- The writing: This paper is verbose and very hard to read. The narrative isn’t very structured or linear. It is at times repetitive, and I don’t think it needs to be 40 pages. Parts of the main paper could also be moved into the appendix, e.g., some details in section 5.
To give specific examples, it is difficult to understand what’s going on at first (e.g., in the abstract, new terms and details are being introduced before we even know what the problem broad scope of the paper), the method, while simple, is not stated formally or in a way simple to understand (it is given kind of in bursts). The introduction in part blurs into a literature review. Parsing what’s going on in the paper and what the authors are trying to communicate takes far too much effort for the reader.
Take the abstract, I would like to see a concise and structured half-page (max 3/4 page) paragraph (or two) consisting of something like a brief problem statement, introduction to your method, a key insight, and its significance. As it is, it is a one-page paragraph. It is hard to parse what you’re trying to communicate, like what this paper actually does, why it is important etc. I suggest trimming it to include only the essentials to understand the problem and then highlighting your contributions, while removing details that can be introduced much later in the paper. The same problem persists throughout the paper.
Another example is in the method (page 10), instead of formally introducing the method (perhaps giving a mathematical formulation of f+ etc), you detail the benefits of the method being unsupervised. This is a discussion that doesn’t belong in the method section, it is distracting to the reader.
- On a related note, the paper needs tidying up in general. E.g., the caption to Figure 1 is incorrectly titled “Linear Representation Hypothesis:” when it should say something like “overview of encoding-decoding”…
- Baseline methods need work: Firstly, further details on the baseline method is required. You give some details on how NMF is implemented, similar to Fel et al, but what about PCA? PCA is a very simple baseline, why did you choose it at all? Second, the only baselines compared are NMF and PCA. Some other methods are not feasible, as you mentioned. More baselines are required, specifically Graziani et al. [1], at a minimum. At several points in the paper, you cite Graziani et al. [1], but their comparable method (essentially SVD with gradient reranking) should be feasible to implement as a baseline. Please note that implementing PCA is not sufficient to compare against Graziani et al.! If you implemented it as in their method (it seems not), please clarify this, otherwise this comparison should be included. On a related note, Graziani et al. also implement ‘concept activation maps’ similar to your Concept Contribution Maps. A comparison would be interesting.
- I would like to see more discussion of the linear representation hypothesis, as this is a key assumption of the method (a discussion added to the appendix is okay). Many methods assume this largely empirical hypothesis, and it doesn’t have a theoretical basis. However, many works now suggest going beyond the linear representation hypothesis [2,3,4]. Please discuss methods that do and don’t assume this, its evidence and counterevidence/limitations.

[1] Graziani, M, et al. "Uncovering Unique Concept Vectors through Latent Space Decomposition." *Transactions on Machine Learning Research*.

[2] Park, Kiho, et al. "The Geometry of Categorical and Hierarchical Concepts in Large Language Models." *The Thirteenth International Conference on Learning Representations*.

[3] O'Mahony, L, et al. "Towards utilising a range of neural activations for comprehending representational associations." *2025 IEEE/CVF Winter Conference on Applications of Computer Vision.* IEEE, 2025.

[4] Diego Simon, Pablo J., et al. "A polar coordinate system represents syntax in large language models." *Advances in Neural Information Processing Systems* 37 (2024): 105375-105396.

**Additional Comments:**

None

**Audience:**

Yes

**Audience Explanation:**

Yes, I believe this work will interest researchers in this area. The paper needs significant refinement and changes, but with revisions, it presents an approach worth attention and further development.

Interpretability is a topic receiving significant attention, currently dominated by SAEs. Exploring alternative approaches to computationally expensive SAEs is valuable. The method could potentially extend to other modalities, such as application to VLMs and LLMs, in the future.

**Broader Impact Concerns:**

- Code: Provide minimal sample code or make a GitHub repository accessible once published.
- Computational complexity and scalability: The method involves multiple steps in encoding and decoding. Include details about the compute resources used for experiments to help readers assess its feasibility in resource-constrained environments.

**Claims And Evidence:**

Yes

**Claims Explanation:**

Yes, the authors validate their method on synthetic and real datasets. However, the comparative baselines need improvement, as detailed in the weaknesses.

**Requested Changes:**

Critical Changes Required for Acceptance:
- Writing and structure: Significant work is needed. Improve logical flow and eliminate repetition. Reduce verbosity throughout. Condense the abstract to 1/2-3/4 a page with a clear problem statement, method, and contributions. Fix formatting issues. I recommend targeting 25-35 pages instead of 40 (recommended but not a critical ask).
- Baseline comparisons: Add Graziani et al. SVD with gradient reranking method as a baseline. Provide more implementation details for existing baselines.

Recommended (To Strengthen the Work):
- Move detailed derivations, extended experiments, and additional related work to appendix to improve main paper accessibility.

---

> ### Author Response · Authors · 2025-12-15
>
> Thank you for the detailed review of our manuscript. We sincerely appreciate your direct feedback on the writing, structure, and baselines. We acknowledge the initial verbosity and have performed a significant revision to enhance clarity, conciseness, and the logical flow of our arguments.
>
> Below are some point-by-point responses:
>
> **RC1: Writing and structure**
>
> We sincerely thank the reviewer for the candid assessment of the manuscript's clarity and structure. We understand your comments regarding verbosity and non-linear narrative. We have performed a major revision focusing on the following:
>
> * **Abstract Revision:** The Abstract is now condensed, structured into three paragraphs that concisely introduce the problem, our novel methodology, and tangible applications.
> * **Method Contributions:** are clearly presented as bullets in Section 4. Introduced losses are presented in Table 1 and an Algorithm description is given in Section A.2.
> * **Trimming and Condensing:** We significantly reduced the overall length and eliminated repetitive or distracting text throughout the main body. The main paper length is now inside the suggested limit of 35 pages. Intro and related work were shortened, Section 5 was significantly reworked with math details moved to the appendix. Uncertainty Region Alignment derivations, parts of Sections 7.1, 7.4.3, 7.11 were moved to the appendix too.
> * **Structural Flow:** Narrative in Sections 4.2, 7.4.3, 7.9, 7.10, 7.11 has been improved, removing verbosity.
>
> **RC2: Baseline comparisons**
>
> We apologize for the confusion regarding baseline comparisons.
>
> To our knowledge, besides UIBE, CBE and SAEs, as well as PCA/NMF - based methods, no other approach has been introduced for **unsupervised** discovery of interpretable **directions**.
> Indeed, our PCA baseline is not an exact implementation of Graziani et al. However, while this may sound like acting dishonestly, below we provide some arguments to explain why such a comparison with the exact implementation is not feasible:
>
> 1. **Protocol Difference:** Our core protocol is **class-agnostic** (concepts discovered irrespective of image class), which differs fundamentally from the **class-conditioned** analysis (concepts discovered in relation to an output class)  used by Graziani et al. and Fel et al.. We prioritize the class-agnostic setting as it aligns with the purpose of using large, annotated datasets like Broden for quantitative evaluation.
> 2. **Infeasibility of Gradient Re-ranking:** The critical component of Graziani's work, gradient re-ranking, is **not feasible** under a class-agnostic setting where class labels are unavailable during the concept discovery and evaluation phases. We cannot revert to a class-conditioned setting because it would require annotations for the large training sets (like ImageNet) to **quantitatively evaluate interpretability**, which are currently unavailable. CLIP-Dissect, as was suggested by another reviewer, still does not solve the problem due to its limited ability to focus on image regions.
> 3. **Our Approach:** We adhere to a fair comparison by applying PCA and NMF as core **class-agnostic matrix decomposition baselines**.
> 4. **Concept Maps:** We clarify that our **Concept Contribution Maps** quantify **concept contribution to the output logit**, conveying a different message from the **Concept Activation Maps** of Graziani et al., which quantify only **concept presence**.
>
> A detailed explanation of this methodological rationale and the limitations of replicating Graziani's, Fel's and Zhang's full algorithm is added to Section A.12 of the revised manuscript, where we also acknowledge that, for these matrix decomposition algorithms, a class-agnostic analysis might be less effective than a class-conditioned. We also highlight the importance of our method in this under-explored protocol. We also made appropriate rephrasing in the name of Section 7.5 and Table/Figure captions to avoid reader's confusion.
>
> **C1: Linear representation hypothesis**
>
> In Section 8, we acknowledge that our method is based on the largely empirical **linear representation hypothesis**. While we recognize the existence of work exploring alternative, non-linear hypotheses, a comprehensive discussion comparing all theoretical bases and counter-evidence is beyond the scope of this work. Our primary focus is on providing **tangible, practical benefits** by demonstrating that exploiting the linear representations hypothesis, using our EDDP approach, is effective for achieving fine-grained model diagnostics and intervention.
>
> **C2: Code**
>
> We will make the source code publicly available upon publication.
>
> **C3: Computational complexity and scalability**
>
> We have added a dedicated section in Section A.21 detailing the computational complexity and the infrastructure used for training, allowing readers to accurately assess the feasibility of our method in various resource environments.

---

> > ### Comment · Reviewer_PAQt · 2025-12-24
> > **Response to authors rebuttal**
> >
> > I with to thank the authors for their response and the noticeable changes made to the manuscript.
> > Having considered the other reviews, I wish to note that I resonate with many of the points made (especially concerns around evaluation by both reviewers, and as Reviewer twFM mentioned, only ConvNet architectures were studied, which is quite a large limitation). Here is my response to the authors’ comment regarding my particular points:
> > - RC1: To be candid, the paper is still somewhat verbose and still leaves some liability of distilling the main points to the reader. Despite this, the clarity of the paper has improved significantly to above an acceptable standard. Therefore I am overall satisfied with the writing in the revised version.
> > - RC2: I remain unsatisfied by the lack of baselines. I am happy with the extra details added about the baseline PCA and NMF implementations, but these simple baselines are not enough to convince me of the merits of this method. On a related note, I strongly support Reviewer Tp88‘S point about including a human study. I would not feel comfortable recommending this paper for acceptance without a human evaluation for at least one model type. Despite the ‘subjectivity’ of human evaluation you mention, it is still an imperative form of evaluation in concept interpretability works.

---

> > > ### Author Response · Authors · 2025-12-26
> > >
> > > **Manuscript Writing**
> > >
> > > We sincerely thank the reviewer for their feedback regarding the manuscript's structure and for acknowledging that the clarity of the revised version has improved significantly to meet and exceed an acceptable standard.
> > >
> > > **Subjective Evaluations**
> > >
> > > We appreciate the reviewer’s perspective that human evaluation is often seen as a critical benchmark in interpretability research. However, we respectfully argue that our quantitative protocol already captures the necessary "human element" in a more objective and scalable manner:
> > >
> > > - **Implicit Human Evaluation via Annotated Data**: Our evaluation incorporates classification and segmentation metrics, such as Precision, Recall, AP and mIoU on the Broden and Broden-Action datasets. Because the ground-truth labels of those datasets were originally produced and verified by **human annotators**, **alignment** with these annotations serves as an implicit, large-scale human evaluation.
> > >
> > > - **CLIP as a Human Proxy**: We employ an automated monosemanticity metric ($\mathcal{M}$) leveraging the CLIP embedding space. As CLIP is trained to align visual features with human language, it acts as a reliable, high-level proxy for **semantic coherence**.
> > >
> > > Manual human studies, while valuable, their potential is capitalized in cases of limited availability of concept labels.
> > >
> > > Conclusively, we believe our evaluation approach already captures the human aspect in measuring interpretability, especially semantic coherence, which is reflected in the Precision and CLIP-based metrics. Especially for Precision, our EDDP achieves **high alignment** with the ground-truth annotations, scoring above 80% in the vast majority of the cases. This is mathematically equivalent to satisfying the constraints of the human annotators who defined those concepts.
> > >
> > > **Baselines**
> > >
> > > While we appreciate the reviewer’s request for additional implementation details regarding the baselines, we find the perceived **insufficiency** to be somewhat **abstract** in the **absence** of **specific**, actionable **suggestions** for more suitable alternatives.
> > >
> > > We respectfully maintain that any candidate baseline for this study must satisfy three rigorous criteria to ensure a fair and methodologically sound comparison:
> > >
> > > - a) Be unsupervised and direction-based
> > > - b) Be applicable in a class-angostic protocol
> > > - c) Not be restricted by the relation between the sizes of the concept space and the embedding space
> > >
> > > As we have discussed, existing methods often fail to meet these combined requirements. Consequently, we believe, no matter how simple they may seem, our current selection of PCA and NMF, which form the basis for more sophisticated approaches (Graziani, Zhang, Fel), and the equally sophisticated UIBE/CBE in the appendix, represents the most robust and theoretically appropriate baselines currently available for this specific research scope. What is more, even if all of (a)-(c) are satisfied, most other works do not make an encoding/decoding direction distinction.
> > >
> > > **Class-Conditioned Analysis**
> > >
> > > It is also worth to note, that our Uncertainty Region Alignment is not applicable in a class-condition protocol, similar to how Graziani's gradient re-ranking is not applicable to a class-agnostic one. Thus a fair direct comparison is also limited in this regard.

---

### Review · Reviewer_Tp88 · 2025-12-01

**Summary Of Contributions:**

The paper proposes a method (EDDP) to discover concept “encoding-decoding” direction pairs in deep vision models. First, an unsupervised clustering step finds decoding directions that behave like concept detectors. Second, a probabilistic signal–distractor model estimates corresponding encoding directions in feature space. A new Uncertainty Region Alignment loss encourages discovered concepts to matter for the classifier’s decisions by aligning regions where concept detectors and the classifier are uncertain. Experiments on standard CNNs report higher Network Dissection scores, more significant RCAV directions, and qualitative applications: concept segmentations, concept contribution maps for explanations, and a watermark-removal case study.

**Audience:**

Yes

**Audience Explanation:**

Unsupervised concept discovery and mechanistic interpretability of deep networks are relevant.
Joint learning of encoding and decoding directions, and tying them to classifier uncertainty, is conceptually interesting even if the current evidence is insufficient.

**Broader Impact Concerns:**

I see no unusual risks beyond those of standard interpretability and representation-editing work. The methods could help detect and correct spurious shortcuts, but might also be used to hide or tweak model behavior; these risks appear manageable with normal transparency and evaluation practices.

**Claims And Evidence:**

No

**Claims Explanation:**

Evidence for the key claims is limited:
1. Faithfulness of encoding directions is only demonstrated on a synthetic linear model and via “dreaming direction” similarity, which is indirect and assumes the method’s own modeling choices.
2. Improvements in interpretability rely almost entirely on Network Dissection metrics and cherry-picked qualitative figures, without human studies or causal evaluations of explanations.
3. RCAV statistics show more directions passing significance tests but say little about practical effect sizes or comparison to simpler baselines.
4. The watermark example is small-scale and lacks baselines, so it does not convincingly show that EDDP enables reliable correction of spurious correlations in realistic settings.

**Requested Changes:**

1. Clarify the algorithm, clearly separating prior components from novel ones and giving concise pseudocode for the whole pipeline.

2. Add experimental comparisons against other sparse autoencoders/dictionary-learning methods.

3. Analyze failure modes and sensitivity to hyperparameters and number of concepts.

4. Simplify and ablate loss terms and augmented-Lagrangian constraints, showing which are truly needed.

5. Quantify the quality of concept contribution maps and editing (e.g., explanation fidelity, success rates) beyond visual examples.

---

> ### Author Response · Authors · 2025-12-15
>
> We sincerely appreciate the detailed and rigorous critique of the reviewer.
> Below are some point-by-point responses:
>
> **RC1: Algorithm code and distiction of novel and previous losses**
>
> In addition to Figs. 4 & 5, in the revised version of the manuscript, we introduce Table 1, providing loss descriptions and novelty emphasized via font color. We also added algorithm pseudo code in Section A.2. Most importantly, we will make the code public upon publication.
>
> **RC2: Compare against SAEs/Dictionaries**
>
> We retain the position that comparing against Sparse Autoencoders (SAEs) or dictionary learning is an excellent avenue for future work. Our method operates in the regime where the concept space is less than the embedding dimension. In contrast, established SAE methods, typically assume an overcomplete representation. We explicitly acknowledge this constraint in Section 8 of the original manuscript.
>
> **RC3: Sensitivity to hyperparameters and concept count**
>
> * **Hyperparameter Sensitivity:** As we acknowledge in Section 8, the choice of hyper-parameters for the experiments rely mostly on previous works and preliminary empirical experimentation. While a detailed study of hyper-parameter sensitivity is beyond the scope of this work, in our experiments we minimized tuning, with most of our hyper-parameters being constant across architectures and training datasets (Section A.12). An exception is Section A.13 which analyzes sensitivity to the target margin loss $\tau^{mm}$.
> * **Concept Count:** Our core comparison (Section 7.5) already covers three distinct values for I (number of concepts) on all architectures.
>
> **RC4: Ablate loss terms and Augmented-Lagrangian constraints**
>
> We assert that the ablation studies in Sections 7.6 and A.14 sufficiently analyze the effect of our novel losses, demonstrating their necessity. Furthermore, we maintain the Augmented Lagrangian formulation because, it is essential for providing interpretable and direct control over the optimization targets (Section 4.5).
>
> **RC4: Quantify quality of CCMs and editing (e.g., explanation fidelity, success rates)**
>
> * **CCM Quantification:** Section 7.9 and Figures 17-20 provide a comprehensive, quantitative decomposition of the explanation logit ($\ell_e$) into its constituent *sample*, *baseline*, and *correction* terms for every concept. This breakdown explicitly quantifies the fidelity of the explanation in logit space.
> * **Editing Success Rates:** The model correction example quantifies the success rate of the intervention by reporting a robust +31\% accuracy improvement on the poisoned test set compared to the random and filter vector baselines.
>
> **C1: Faithfulness**
>
> We defend the "Dreaming Direction" analysis as a novel approach to validate fidelity **in the wild**. It specifically addresses a gap in the literature where prior state-of-the-art methods rely on synthetic or controlled training settings to validate their concepts. Our results demonstrate that the alignment achieved is comparable to the fidelity scores reported by supervised methods under controlled conditions (e.g., Pahde et al., 2024).
>
> **C2: Improvements in interpretability rely on Network Dissection metrics, qualitative figures, without human studies or causal evaluations of explanations.**
>
> We recognize the absence of human studies as a limitation of scope for this large-scale comparative work, as performing hundreds of comparisons across five architectures would be infeasible.
>
> However, the statement that improvements rely entirely on network dissection metrics needs more clarification from the reviewer's side. In our opinion, depending on the case, EDDP demonstrates interpretability improvements in any of the Classification (Precision/Recall/F1-Score/AP), Segmentation (mostly NetDissect-based), Monosemanticity (CLIP based) metrics.
> Moreover, more than 11 pages of the appendix depict qualitative figures to mitigate, at least to some extent, a reader's concern about our claims.
>
> Our Concept Contribution Maps (CCMs) serve as an initial causal explanation by quantifying the contribution of each concept to the predicted logit. We further validate explanations by the counterfactual interventions in Section 7.10.
>
> **C3: The watermark example is small-scale and lacks baselines, so it does not convincingly show that EDDP enables reliable correction of spurious correlations in realistic settings.**
>
> We acknowledge the toy nature of this example. However, the experiment *does* include baselines by explicitly comparing the correction efficacy of the learned **signal vector** against an intervention using a **random vector** and an intervention using the learned **filter vector** in the manipulation. This comparison validates that the full encoding-decoding pair recovered by EDDP is uniquely necessary for successful unlearning, achieving +31\% accuracy improvement over the baselines.

---

### Decision · Action_Editor_6vGf · 2026-01-19

**Recommendation:** Accept with minor revision

**Additional Comments:**

After the discussion, the reviewers did not converge to a recommended decision. Two Reviewers lean towards acceptance, while one leans towards rejection. The main concerns revolve around the limited baselines in the evaluation (see above).

Given that the Reviewers acknowledge the novelty of the approach and find the paper of interest for the community, I am suggesting acceptance of the paper, conditioned on softening the statements and claims throughout it in order to make it crystal clear what has been done and what hasn't. (See full comments above.)

**Audience:**

Yes

**Audience Explanation:**

Yes, the Reviewers agree that the paper is of interest for the AI interpretability research community. In particular, the research question is relevant, the method is novel, and the introduces loss fuctions are also novel and of interest.

**Claims And Evidence:**

Yes

**Claims Explanation:**

(Partially - See below)

This paper proposes an usupervised approach for recovering semantic concepts in deep vision models. The method jointly learns encoding–decoding direction pairs in latent space, using directional clustering to identify directions in the latent space. The paper also introduces various loss functions that improve interpretability and influence of the identified concept directions. The paper contains examples of practical applications that demonstrate the usefulness of the learned directions.

The introduced method is validated on synthetic and real datasets, which serves as a way to support the paper's claims. However, there are experimental limitations; in particular, the comparative baselines (just NMF and PCA) are insufficient. This is something that several Reviewers have pointed out. The authors have argued that no existing baseline is comparable (according to a number of properties), yet the Reviewers are not fully convinced, although no specific method was suggested after the discussion. At the very least, a human evaluation (which is standard in concept interpretability works) would be greatly useful. The authors' response to this was:
> "We recognize the absence of human studies as a limitation of scope for this large-scale comparative work, as performing hundreds of comparisons across five architectures would be infeasible."

I would argue that it is not necessary to compare against 5 architectures, but I do appreciate the essence of the comment regardless.

Given that the authors did not add any other baselines or human evaluation, I encourage them to instead soften the claims throughout the paper. For instance, the paper can make it explicit from the beginning (including abstract and introduction) that the "other unsupervised baselines" are PCA and NMF, so as to make the text more explicit and accurate.

Finally, I also encourage the authors to explicitly state that the comparison against SAEs is not done - this is currently mentioned in Section 8 but it would be good to mention early on (in the introduction), even if it's in passing.

Overall, the text should be faithful to its content and the statements should be factual and accurate, without any riks of misinterpreting or overselling the results.